# Corruption-Tolerant Asynchronous Q-learning with Near-Optimal Rates

**Sreejeet Maity** [1]   **Aritra Mitra** [1]

## Abstract

We study the problem of learning the optimal policy in a discounted, infinite-horizon reinforcement learning (RL) setting in the presence of adversarially corrupted rewards. To address this problem, we develop a novel robust variant of the $Q$-learning algorithm and analyze it under the challenging asynchronous sampling model with time-correlated data. Despite corruption, we prove that the finite-time guarantees of our approach match existing bounds, up to an additive term that scales with the fraction of corrupted samples. We also establish an information-theoretic lower bound, revealing that our guarantees are near-optimal. Notably, our algorithm is agnostic to the underlying reward distribution and provides the first finite-time robustness guarantees for asynchronous $Q$-learning. A key element of our analysis is a refined Azuma-Hoeffding inequality for almost-martingales, which may have broader applicability in the study of RL algorithms.

## 1. Introduction

In a typical reinforcement learning (RL) problem, a learning agent interacts sequentially with an environment modeled as a Markov Decision Process (MDP). Each interaction involves the agent playing an action and receiving feedback in the form of a reward for the action taken. Using such feedback, the agent gains a better understanding of the quality of the actions, allowing it to eventually learn an optimal decision-making policy. The formalism described above finds use in a variety of practical applications, spanning finance, medicine, recommendation systems, autonomous driving, robotics, and most recently, training large language models using human feedback. In each of these applications, *the effectiveness of the learned policy depends crucially on the quality of the feedback data (rewards)* used to train the

policy. In real-world applications, however, data can be noisy and can contain outliers: human feedback can be biased and have malicious intent, recommendation systems can be skewed by fake users, and sensor data in an autonomous vehicle can be prone to measurement errors and be corrupted by an adversary. If precautions are not taken to contend with "bad data", then the consequences can be dire, especially for safety-critical applications. Motivated by this concern, we revisit the classical RL problem from the perspective of *adversarial robustness* and study a scenario where a portion of the rewards observed by the learner can be corrupted *arbitrarily*. For this scenario, we wish to understand to what extent one can hope to still learn a (near)-optimal policy. Surprisingly, despite the popularity of RL, a complete theoretical understanding of this question is lacking in the current literature, especially for the scenario where data are collected in an online, sequential manner. Our work in this paper contributes to filling this gap.

We consider an infinite-horizon discounted RL problem, where an agent collects data from the environment based on a behavior/sampling policy, as is done with popular RL algorithms such as $Q$-learning (Watkins & Dayan, 1992). We depart from the standard RL observation model by allowing the rewards to be corrupted based on a fixed corruption probability $\varepsilon \in [0, 1/2)$: at each time-step, with probability $1 - \varepsilon$, the learner (agent) observes a reward sampled from the true reward distribution associated with the current state and action, and with probability $\varepsilon$, it observes a sample from an arbitrary adversarial distribution. Importantly, we put no restrictions at all on the adversarial distribution, allowing for potentially unbounded attack signals. Furthermore, we allow the true reward distributions to be *heavy-tailed*, requiring them to admit no more than a finite second moment. It should be noted here that our way of modeling corruption is inspired directly by the Huber model from robust statistics (Huber, 1992; 2004). Furthermore, similar corruption models have been extensively studied for the simpler bandits setting (Jun et al., 2018; Lykouris et al., 2018; Liu & Shroff, 2019; Gupta et al., 2019; Kapoor et al., 2019; Agrawal et al., 2024), and more recently in offline RL with human feedback (Mandal et al., 2024). However, when it comes to learning an optimal policy in the infinite-horizon discounted setting we consider here with online, sequential data, the effect of such an attack model remains completely unexplored.

---

[1] Electrical and Computer Engineering, North Carolina State University, Raleigh, NC, USA. Correspondence to: Sreejeet Maity <smaity2@ncsu.edu>, Aritra Mitra <amitra2@ncsu.edu>.

*Proceedings of the $43^{rd}$ International Conference on Machine Learning*, Seoul, South Korea. PMLR 306, 2026. Copyright 2026 by the author(s).

Since an optimal policy can be extracted by learning the optimal state-action value function (Sutton, 1988), we ask two concrete questions: Subject to our corruption model: (i) *Can one still reliably estimate the optimal state-action value function?* (ii) *What is a fundamental lower bound on estimation accuracy in this setting?* Our contributions described below comprehensively address these questions.

• **Novel Robust $Q$-learning Algorithm.** In Section 3, we start by considering a setting where bounds on the first and second moments of the true reward distributions are known to the learner. For this setting, we propose a new algorithm called Robust Async-Q that comprises two main ingredients. The first idea is to leverage the recent univariate trimmed mean estimator from Lugosi & Mendelson (2021) to maintain running estimates of the mean rewards for each state-action pair of the MDP, using historical data for such pairs. However, this idea is not enough on its own since the guarantees associated with robust mean estimation are probabilistic in nature, and, as such, may not hold on rare, extreme events. To control the errors introduced by adversarial contamination on such rare events, we employ a second layer of safety that involves keeping track of "typical" regions that contain the reward mean estimates; estimates that fall outside the typical regions are rejected. The size of these typical regions - as captured by an *adaptive threshold* - shrinks as the learner acquires more samples.

For the case where bounds on the reward statistics are *unknown* a priori, constructing the adaptive threshold accurately becomes much trickier. In Section 4, we propose a simple modification to Robust Async-Q that addresses this challenge by using a "slowly growing" function of time as a proxy for such bounds. *Overall, we prescribe a framework for constructing robust empirical estimates of the Bellman optimality operator using noisy, time-correlated, and corrupted data collected online.*

• **Finite-Time Rates under I.I.D. Sampling.** To build intuition, we start by analyzing Robust Async-Q under a simplified i.i.d. sampling model, commonly used in previous RL works (Korda & La, 2015; Dalal et al., 2018; Narayanan & Szepesvári, 2017; Lakshminarayanan & Szepesvári, 2017). In Theorems 2 and 4, we provide high-probability finite time rates for Robust Async-Q with known and unknown reward statistics, respectively. Given $T$ samples, in each case, our bounds match the known optimal rate (Wainwright, 2019b; Guannan Qu, 2020; Li et al., 2024) of $\widetilde{\mathcal{O}}(1/\sqrt{T})$, up to a small additive term on the order of $\mathcal{O}(\sqrt{\varepsilon})$, where $\varepsilon$ is the probability of corruption. Our bounds also reveal how the effect of asynchronous sampling can inflate the corruption-induced term. *To our knowledge, Theorems 2 and 4 provide the first formal guarantees of adversarial robustness for asynchronous Q-learning.*

• **Fundamental Lower Bound.** One might ask whether the

$\mathcal{O}(\sqrt{\varepsilon})$ term in our upper-bound is unavoidable. In Theorem 3, we settle this question by providing an information-theoretic fundamental lower bound, revealing that an $\Omega(\sqrt{\varepsilon})$ error in the estimation of the optimal state-action value function is *unavoidable*. **Collectively, our results are significant in that they reveal that** Robust Async-Q **achieves near-optimal finite-time guarantees for $Q$-learning under adversarial corruption.**

• **Finite-Time Rates under Markov Sampling.** In Section 4.1, we study our setting in full generality by considering the challenging single-trajectory Markovian sampling model with time-correlated data. In Theorem 6, we prove that one can nearly recover the same bounds as in the i.i.d. setting, up to an inflation in the $\widetilde{\mathcal{O}}(1/\sqrt{T})$ term caused by the mixing time of the underlying Markov chain; notably this inflation is consistent with prior bounds in the absence of corruption (Guannan Qu, 2020).

• **Novel Proof Techniques.** Arriving at our results involves several new proof ingredients. Even with i.i.d. sampling and known reward statistics, some work is needed to account for the fact that under the asynchronous sampling model, the number of times each state-action pair has been sampled (up to a given time-step) is a *random variable*, precluding the direct use of robust mean estimation bounds. To overcome this issue, we use Bernstein's inequality to control the number of visits to each state-action pair. A key new step in our analysis is to argue that after a certain burn-in time, no estimates will be rejected (due to thresholding) on a good event of sufficient measure. When the reward statistics are unknown a priori, the use of slowly growing functions of time as their proxies introduces significant new challenges. In particular, as we explain in Section 4, using the standard version of the Azuma-Hoeffding inequality - which is what is done in existing $Q$-learning analyses (Guannan Qu, 2020) - will unfortunately lead to vacuous bounds in our setting. Furthermore, relatively well-known variants of the Azuma-Hoeffding inequality for discrete probability spaces (Chung & Lu, 2006), and sub-Gaussian martingale differences (Shamir, 2011) also prove to be inadequate for our purposes. To overcome this challenge, we show how a refined variant of the Azuma-Hoeffding inequality from Shamir & Spencer (1987) can be carefully exploited to preserve near-optimal bounds; *we are unaware of the use of this new tool in any prior RL work*, and believe that it might be more broadly applicable. Finally, to handle the challenging single-trajectory Markov setting, we combine the above ideas with a coupling technique inspired by recent work (Dorfman & Levy, 2022; Nagaraj et al., 2020).

**Summary.** To sum up, we provide the first principled and comprehensive study of adversarial robustness in RL for the infinite-horizon, discounted setting with asynchronous Markovian data. Our new algorithms and analysis tech-

niques, complemented by nearly matching upper and lower-bounds, paint a fairly complete picture for this setting.

**Related Work.** We now discuss the most relevant works on corruption-robust RL here, and relegate a more detailed survey to Appendix A. The topic of reward corruption has been explored in several papers on bandits (Jun et al., 2018; Lykouris et al., 2018; Liu & Shroff, 2019; Gupta et al., 2019; Bogunovic et al., 2021; 2020; Garcelon et al., 2020; Kapoor et al., 2019; He et al., 2022; Agrawal et al., 2024). In the context of MDPs, data corruption in online, finite-horizon episodic RL problems is studied in Lykouris et al. (2021); Chen et al. (2021); Wei et al. (2022); Ye et al. (2023a), where performance is measured by cumulative regret and the algorithms are variants of either Upper-Confidence-Based (UCB) or Action-Elimination strategies. The infinite-horizon discounted setting we study here *differs fundamentally* in terms of the notion of performance (sample-complexity), and also in terms of the algorithm design principle, which is rooted in stochastic approximation theory. Corruption-robust algorithms in the offline setting or with access to a generative model/simulator are considered by Zhang et al. (2022); Ye et al. (2023b); Mandal et al. (2024); Maity & Mitra (2024), where batched data tuples are collected offline in an i.i.d. manner. In sharp contrast, we need to contend with a much more challenging observation model, where *heavy-tailed and corrupted* data arrives in an online, sequential manner as part of a *single trajectory*, and the state-action pairs are visited asynchronously, creating the problem of *partial observability*. Finally, we note that the issue of handling just heavy-tailed rewards (without adversarial corruption) has been studied in problem settings different from ours: for offline RL by Zhu et al. (2024), for episodic RL by Zhuang & Sui (2021), and for policy evaluation by Cayci & Eryilmaz (2024).

## 2. Background and Problem Formulation

We start by providing the basic background on RL, and then proceed to describe our problem of interest. We consider a $\gamma$-discounted infinite-horizon Markov Decision Process (MDP) $\mathcal{M} = (\mathcal{S}, \mathcal{A}, \mathcal{P}, R, \gamma)$, where $\mathcal{S}$ is a finite state space, $\mathcal{A}$ is a finite action space, $\mathcal{P}$ is a set of state transition kernels, $R$ is a reward function, and $\gamma \in (0, 1)$ is a discount factor. When in state $s \in \mathcal{S}$ the learner plays an action $a \in \mathcal{A}$, it observes a new state $s'$ drawn from $\mathcal{P}(\cdot|s, a)$, and a stochastic reward sample $r(s, a)$ drawn from a reward distribution $\mathcal{R}(s, a)$. The noisy reward $r(s, a)$ is unbiased with mean equal to the true expected reward $R(s, a)$ for state-action pair $(s, a)$, and variance $\sigma^2(s, a)$, i.e., $\mathbb{E}[r(s, a)] = R(s, a)$, and $\mathbb{E}[(r(s, a) - R(s, a))^2] = \sigma^2(s, a)$. We assume that the mean rewards and variances are uniformly bounded, i.e., there exist $\bar{R}, \bar{\sigma} \geq 1$ such that $|R(s, a)| \leq \bar{R}$ and $\sigma^2(s, a) \leq \bar{\sigma}^2, \forall (s, a) \in \mathcal{S} \times \mathcal{A}$. A policy $\mu : \mathcal{S} \to \Delta(\mathcal{A})$

is a mapping from the states to a space of probability distributions over actions, denoted by $\Delta(\mathcal{A})$. The quality of a policy $\mu$ is captured by an expected discounted infinite-horizon cumulative reward known as the value function $V^\mu$:

$$V^\mu(s) = \mathbb{E}\left[\sum_{t=0}^\infty \gamma^t R(s_t, a_t) \,\middle|\, s_0 = s, \mu\right], \quad (1)$$

where $s_t$ and $a_t$ are the state and action at time $t$, respectively, under the action of the policy $\mu$ on the MDP $\mathcal{M}$. The goal of the learner is to find a policy $\mu$ that maximizes the value function $V^\mu$ for all states, *without knowledge* of the transition kernels $\mathcal{P}$ and reward functions $R$ of the underlying MDP. To explain how this is done, we will need to introduce the notion of a state-action value function $Q^\mu$ for a policy $\mu$, defined as $Q^\mu(s, a) = \mathbb{E}\left[\sum_{t=0}^\infty \gamma^t R(s_t, a_t) \,\middle|\, (s_0, a_0) = (s, a), \mu\right]$. The celebrated $Q$-learning algorithm (Watkins & Dayan, 1992) uses data collected by a suitable behavior/sampling policy $\mu$ to iteratively maintain an estimate of the optimal state-action value function, denoted by $Q^*$. It turns out that $Q^*$ is the fixed point of a contractive operator known as the Bellman optimality operator (Sutton, 1988). Using this contraction property, classical asymptotic results (Tsitsiklis, 1994; Jaakkola et al., 1993) established that the sequence of iterates generated by $Q$-learning converges to $Q^*$ almost surely (under suitable assumptions on $\mu$). More recently, finite-time rates have been established (Wainwright, 2019b; Guannan Qu, 2020; Li et al., 2024), revealing that when run for $T$ iterations, the final iterate of $Q$-learning converges to $Q^*$ at a rate of $\widetilde{\mathcal{O}}(1/\sqrt{T})$, with high probability. Once $Q^*$ is known, an optimal policy can be determined by playing actions greedily with respect to $Q^*$ (Sutton & Barto, 2018).

**Adversarially Corrupted Reward Model.** Our formulation departs from the standard setting described above in two main ways. First, classical results on $Q$-learning either assume deterministic rewards or "light-tailed" noisy rewards with sub-Gaussian reward distributions. In contrast, our formulation requires the reward distributions $\mathcal{R}(s, a)$ to admit only up to a finite second moment, and nothing more. Thus, *the true reward distributions are allowed to be heavy-tailed.* More importantly, we allow a portion of the reward data to be corrupted *arbitrarily* by an adversary. To explain the corruption model precisely, suppose that data are collected based on a stochastic behavior policy $\mu$, such that $\mu(a|s) > 0, \forall s \in \mathcal{S}, \forall a \in \mathcal{A}$. Upon interacting with the MDP $\mathcal{M}$, the policy $\mu$ induces a Markov chain. Let $s_t$ be the state of this Markov chain at time $t$. Then, in the standard $Q$-learning setting, at each time-step $t$, the learner observes the data tuple $(s_t, a_t, s_{t+1})$, and noisy reward $r_t(s_t, a_t)$, where $a_t \sim \mu(\cdot|s_t)$, $s_{t+1} \sim \mathcal{P}(\cdot|s_t, a_t)$, and $r_t(s_t, a_t) \sim \mathcal{R}(s_t, a_t)$. Here, we assume that the noise process $\{n_t := r_t(s_t, a_t) - R(s_t, a_t)\}$ is independent over

time and of all other sources of randomness. In our setting, the learner still observes $(s_t, a_t, s_{t+1})$, but now receives a Huber-contaminated reward $y_t(s_t, a_t)$ generated as follows. At time $t$, a biased coin with probability of heads $1 - \varepsilon$ is tossed independently of the past, and all other sources of randomness in the problem; here $\varepsilon \in [0, 1/2)$ is a fixed probability that captures the fraction of corrupted samples. If the coin lands heads, $y_t(s_t, a_t)$ is drawn from the true reward distribution $\mathcal{R}(s_t, a_t)$. If it lands tails, $y_t(s_t, a_t)$ is drawn from an *unconstrained and arbitrary* adversarial distribution $\mathcal{Q}$ that can depend on history, and be time and state-action pair dependent. In other words, if $y_t(s_t, a_t)$ is drawn from $\mathcal{Q}$, it can be arbitrary (and hence, potentially unbounded). Concretely, we write $y_t(s_t, a_t) \sim (1 - \varepsilon)\mathcal{R}(s_t, a_t) + \varepsilon\mathcal{Q}$, where the notation $(1 - \varepsilon)\mathcal{P}_1 + \varepsilon\mathcal{P}_2$ is used to represent the mixture of two distributions $\mathcal{P}_1$ and $\mathcal{P}_2$. The corrupted observation $y_t(s_t, a_t)$ can be expressed as follows:

$$y_t(s_t, a_t) = (1 - Y_t)\, r_t(s_t, a_t) + Y_t\, z_t, \qquad (2)$$

where $\{Y_t\}_{t \geq 0}$ is an i.i.d. sequence of Bernoulli random variables with parameter $\varepsilon \in [0, 1/2)$, $r_t(s_t, a_t) = R(s_t, a_t) + n_t$ is the noisy reward and $z_t \sim \mathcal{Q}$ is the corruption signal, respectively, at time $t$.

**Problem 1.** *Given $T$ samples $(s_t, a_t, s_{t+1}, y_t(s_t, a_t)), t = 0, \ldots, T - 1$ from the corrupted reward model in (2), and a prescribed failure probability $\delta \in (0, 1)$, our goal is to generate a robust estimate $Q_T$ of the optimal value function $Q^*$, and quantify a bound on the $\ell_\infty$-error $\|Q_T - Q^*\|_\infty$ that holds with probability at least $1 - \delta$.*

Specifically, we ask: (i) Can one still hope to (nearly) preserve the optimal $\widetilde{\mathcal{O}}(1/\sqrt{T})$ rate of vanilla $Q$-learning? (ii) What are the fundamental limits on performance imposed by the reward-corrupted attack model? As far as we are aware, despite the popularity of $Q$-learning, answers to neither of these basic questions are available in the literature. The main contribution of our work is to close this gap by developing an algorithm that achieves near-optimal guarantees for the posed problem.

**Challenges.** There are several unique technical challenges in our problem. First, the heavy-tailed nature of the true reward distribution makes it harder for the learner to distinguish between true samples drawn from the tails of such distributions and adversarial outliers. This uncertainty is further exacerbated when the learner has no knowledge at all about the statistics of the reward distributions - a setting we analyze in Section 4. Second, data in our setting are collected in an online, asynchronous manner, where only a single state-action pair is visited at each time-step. Even in the absence of corruption, such a setting is non-trivial to analyze in the non-asymptotic regime. Third, the data is generated based on a time-correlated Markov chain, making it hard to directly apply standard results from robust statistics that deal with i.i.d. data collected offline. As we will

discuss throughout the paper, overcoming these challenges requires significant algorithmic and technical innovations.

Before we introduce our proposed approach, let us state an assumption that is standard in the analysis of RL algorithms (Tsitsiklis, 1994; Tsitsiklis & Van Roy, 1997; Bhandari et al., 2018; Guannan Qu, 2020; Li et al., 2024).

**Assumption 1.** *The Markov chain $\{s_t\}$ induced by the behavior policy $\mu$ is aperiodic and irreducible.*

If $\pi$ is the stationary distribution of the Markov chain induced by $\mu$, then Assumption 1 ensures that $\pi(s) > 0, \forall s \in \mathcal{S}$. At stationarity, note that the visitation probability of a particular state-action pair $(s, a)$ is given by $\lambda(s, a) := \pi(s)\mu(a|s)$, which is non-zero, based on our assumptions on the behavior policy. For later use, we further define the *minimum visitation probability* as $\lambda_{\mathrm{min}} = \min_{(s,a) \in \mathcal{S} \times \mathcal{A}} \lambda(s, a)$. To clearly explain our main ideas, we will assume in Sections 3 and 4 that at each time-step $t$, the state $s_t$ is sampled *independently* from its stationary distribution $\pi$. Later, in Section 4.1, we will relax this i.i.d. assumption, and consider single-trajectory Markov data.

## 3. Robust Asynchronous Q-learning

In this section, we develop a robust variant of the $Q$-learning algorithm that accounts for asynchronously sampled data, and adversarially corrupted rewards. Our algorithm, titled `Robust Async-Q`, is formally described in Algorithm 1. We start by providing an overview of `Robust Async-Q`, and then flesh out the details. Our approach has two core components: (i) **Robust Reward Estimation.** The first main idea is to use the history of reward observations for each state-action pair $(s, a)$ to generate a robust estimate of the mean reward $R(s, a)$; for this purpose, we exploit the univariate trimmed mean estimator from Lugosi & Mendelson (2021). (ii) **Adaptive Thresholding.** To account for rare events where robust estimation guarantees may not hold, we carefully design an adaptive thresholding mechanism to discard extreme estimates and ensure that the iterates of `Robust Async-Q` remain uniformly bounded. We will show later that by carefully stitching these ideas together, `Robust Async-Q` is able to achieve near-optimal convergence rates. We now supply the details.

● **Idea 1: Reward Filtering Mechanism.** We start by briefly describing the robust univariate trimmed mean estimator from Lugosi & Mendelson (2021) that we will employ to estimate reward functions. Consider a data set $\mathcal{D}$ comprising of $M$ i.i.d. samples of a scalar random variable $X$ with mean $\mu_X$ and variance $\sigma_X^2$. An adversary arbitrarily perturbs up to $\varepsilon M$ of the samples within $\mathcal{D}$ to produce a corrupted data set $\tilde{\mathcal{D}}$; here, $\varepsilon \in [0, 1/2)$ is the fraction of corrupted data. Using $\tilde{\mathcal{D}}$, the corruption fraction $\varepsilon$, and a confidence parameter $\delta$ as inputs, the trimmed mean esti-

mator from Lugosi & Mendelson (2021) produces a robust estimate $\hat{\mu}_X$ of the mean $\mu_X$ in the following way. The data set $\tilde{\mathcal{D}}$ is divided into two equal parts of $M/2$ samples each. The first part is used to compute empirical quantiles for filtering out extreme values. The estimate $\hat{\mu}_X$ is then simply an average of only those data samples in the second part that fall within the computed quantiles. To apply the estimator from Lugosi & Mendelson (2021) in our context, we need to make minor modifications to the algorithm and the analysis in Lugosi & Mendelson (2021) to account for the Huber contamination model introduced in Section 2. The details of these modifications, along with the manner in which the quantiles are computed, are provided in Appendix B. Let $\hat{\mu}_X = \text{TRIM}[\tilde{\mathcal{D}}, \varepsilon, \delta]$ be used to succinctly represent the output of the trimmed mean estimator described above. The following result, adapted from Lugosi & Mendelson (2021), will be of use to us in the sequel.

**Theorem 1.** *Let $\delta \in (0,1)$ be such that $\delta \geq 8e^{-M/2}$. The following then holds with probability at least $1 - \delta$:*

$$|\hat{\mu}_X - \mu_X| \leq \mathcal{C}\sigma_X \left( \sqrt{\varepsilon} + \sqrt{\frac{\log(8/\delta)}{M}} \right), \qquad (3)$$

*where $\mathcal{C} \geq 1$ is a universal constant.*

To make use of the estimator explained above, our algorithm maintains a reward history for each state-action pair $(s,a) \in \mathcal{S} \times \mathcal{A}$ via a dynamic array $\mathcal{D}_t(s,a)$ that is initialized from the empty set, i.e., $\mathcal{D}_0(s,a) = \emptyset, \forall(s,a)$. Now, under the asynchronous i.i.d. sampling model, at each time-step $t$, the learner observes a fresh state-action pair sampled as $s_t \sim \pi$ and $a_t \sim \mu(\cdot|s_t)$. If $(s,a) = (s_t, a_t)$, the observed reward $y_t(s_t, a_t)$ is appended to the corresponding array $\mathcal{D}_t(s_t, a_t)$. If $(s,a) \neq (s_t, a_t)$, then the corresponding array remains unchanged from before. Using the dynamic data set $\mathcal{D}_t(s_t, a_t)$, the corruption fraction $\varepsilon$, and a confidence level $\delta_1 = \delta/(4T)$, a robust estimate $\bar{r}_t(s_t, a_t)$ of the true expected reward $R(s_t, a_t)$ is computed as follows: $\bar{r}_t(s_t, a_t) = \text{TRIM}[\mathcal{D}_t(s_t, a_t), \varepsilon, \delta_1]$. Here, note that if we wish the overall output of Robust Async-Q to be accurate with a prescribed probability of at least $1 - \delta$, then the failure probability $\delta_1 = \delta/(4T)$ that needs to be fed to the trimmed mean estimator needs to be much finer. The operations above are described in lines 4-5 of Algorithm 1.

• **Idea 2: Adaptive Thresholding.** There are two main obstacles that prevent us from directly using $\bar{r}_t(s_t, a_t)$ (as estimated above) as a proxy for the true mean $R(s_t, a_t)$. First, during the initial phases of our algorithm, one may simply not have visited a particular state-action pair enough times for the robust estimation guarantee to be meaningful. *Thus, we need to wait long enough to acquire adequate observations for every state-action pair.* Second, even when each state-action pair has been visited several times, the guarantees associated with the mean estimator from Lugosi

& Mendelson (2021) only hold with *high-probability, not deterministically* (as is evident from Theorem 1). As a result, one cannot rule out extreme events, where the output of the trimmed mean estimator can deviate arbitrarily from the true mean. On such events, using $\bar{r}_t(s_t, a_t)$ directly can lead to uncontrolled errors. The above discussion suggests that *robust estimation is insufficient on its own.* To overcome the two issues described above, we introduce the idea of an **adaptive threshold** that dynamically keeps track of the *typical region* where we expect the output of the trimmed mean estimator to lie within. If the estimate $\bar{r}_t(s_t, a_t)$ falls outside this region, we deem it to be "extreme" and simply discard it by thresholding it to 0.

To formally introduce the adaptive threshold, we first define a burn-in time $\bar{T}$ as follows:

$$\bar{T} = \left\lceil \frac{104}{3\lambda_{\min}} \log \left( \frac{8|\mathcal{S}||\mathcal{A}|T}{\delta_1} \right) \right\rceil, \qquad (4)$$

where recall from Section 2 that $\lambda_{\min} > 0$ is the minimum state-action visitation probability. Our analysis will reveal that for $\forall t \geq \bar{T}$, the number of visits to each state-action pair $(s,a)$ up to time $t$ is well concentrated around its mean value $\lambda(s,a)t$ with high probability; this is needed to address the first issue of acquiring enough data. We now define our adaptive threshold $G_t$ as follows:

$$G_t = \begin{cases} 0, & \text{if } t \leq \bar{T}, \\ \mathcal{C}\bar{\sigma} \left( \sqrt{\dfrac{4\log(8/\delta_1)}{3\lambda_{\min}t}} + \sqrt{\varepsilon} \right) + \tilde{\sigma}, & \text{if } t > \bar{T}, \end{cases} \qquad (5)$$

where $\mathcal{C}$ is the universal constant from Theorem 1, and $\tilde{\sigma} = \max\{\bar{R}, \bar{\sigma}\}$; here, note that we implicitly assume $\tilde{\sigma}$ is known, an assumption we will relax in Section 4. With the threshold $G_t$ in hand, we account for extreme events as follows: if $|\bar{r}_t(s_t, a_t)| > G_t$, then we discard the estimate by thresholding it to 0. Else, we accept the output of the trimmed mean estimator as is. This operation is described in lines 6-10 of Algorithm 1, where the output of the thresholding scheme is denoted by $\tilde{r}_t(s_t, a_t)$. We emphasize here that the design of the adaptive threshold is the most innovative part of our algorithm **and needs to be done *just right* to achieve near-optimal guarantees**: *if the threshold is too tight, then we will reject estimates unnecessarily; if it is too loose, we might end up accepting extreme estimates. Either case can lead to vacuous bounds.*

• **Proposed Robust Q-Update.** We can now formally state the update rule of Robust Async-Q which uses $\tilde{r}_t(s_t, a_t)$ - as generated above - as a proxy for the true reward mean $R(s_t, a_t)$ in the $Q$-learning rule of Watkins (Watkins & Dayan, 1992), where $Q_{t+1}(s,a)$ is defined as follows:

$$\begin{cases} Q_t(s,a), & \text{if } (s,a) \neq (s_t, a_t), \\ (1-\alpha)Q_t(s,a) + \alpha \left[ \tilde{r}_t(s,a) + \gamma \max_{a' \in \mathcal{A}} Q_t(s_{t+1}, a') \right], & \text{otherwise.} \end{cases} \qquad (6)$$

**Algorithm 1** `Robust Async-Q`

---

1: **Input:** Step-size $\alpha$, corruption fraction $\varepsilon$, confidence level $\delta$, iteration count $T$.
2: Initialize datasets $\mathcal{D}_0(s,a) = \emptyset$, for all $(s,a) \in \mathcal{S} \times \mathcal{A}$, and Q-table $Q_0 = 0$.
3: **for** iteration $t = 0, \ldots, T-1$ **do**
4:     Observe data tuple $\{s_t, a_t, s_{t+1}\}$, and reward $y_t(s_t, a_t)$.
5:     Append $y_t(s_t, a_t)$ to $\mathcal{D}_t(s_t, a_t)$, and compute $\bar{r}_t(s_t, a_t) \leftarrow \text{TRIM}[\mathcal{D}_t(s_t, a_t), \varepsilon, \delta_1]$.
6:     **if** $|\bar{r}_t(s_t, a_t)| > G_t$ **then**
7:         Set $\tilde{r}_t(s_t, a_t) \leftarrow 0$
8:     **else**
9:         Set $\tilde{r}_t(s_t, a_t) \leftarrow \bar{r}_t(s_t, a_t)$
10:     **end if**
11:     Update $Q_{t+1}$ using Eq. (6).
12: **end for**

---

The update rule above ensures that only robust and bounded reward estimates influence the learning dynamics. In the next section, we will see that the combination of robust filtering and thresholding yields finite-time error bounds for `Robust Async-Q` that gracefully degrade with the corruption level $\varepsilon$, while matching the classical Q-learning rate in the absence of corruption.

**Remark 1.** Algorithm 1 uses the corruption fraction $\varepsilon$ and the minimum visitation probability $\lambda_{\min}$ to set design parameters. Using knowledge of $\varepsilon$ is standard in robust estimation, where $\varepsilon$ determines the filtering level (Lugosi & Mendelson, 2021; 2019), and also appears in related bandit and offline RL works (Kapoor et al., 2019; Agrawal et al., 2024; Zhang et al., 2022). Exact knowledge of $\varepsilon$ is not essential: any valid upper bound $\bar{\varepsilon} \geq \varepsilon$ can be used, with the resulting guarantee holding with $\varepsilon$ replaced by $\bar{\varepsilon}$. Similarly, knowledge of $\lambda_{\min}$ is routinely assumed even in recent analyses of asynchronous Q-learning without corruption, where it is used to tune the step-size (Guannan Qu, 2020; Li et al., 2024). Although $\lambda_{\min}$ can be consistently estimated from empirical state-action frequencies under Assumption 1, finite-time guarantees for plug-in tuning appear unavailable even for vanilla Q-learning. Developing fully adaptive robust Q-learning algorithms that require no prior knowledge of either $\varepsilon$ or $\lambda_{\min}$ remains an important future direction.

### 3.1. Main Results for Robust Async-Q

In this section, we provide our first set of results for `Robust Async-Q` with known bounds on reward means and variances. To that end, define $d_t := Q_t - Q^*, \forall t \geq 0$. We then have the following result.

**Theorem 2.** *Suppose Assumption 1 holds, and $T$ satisfies:* $T > \max\{\bar{T}, \log(T)/(\lambda_{min}(1-\gamma))\}$. *Given any $\delta \in (0,1)$,*

*the output of Algo. 1 with step-size $\alpha = \frac{\log T}{\lambda_{min}(1-\gamma)T}$ satisfies the following bound with probability at least $1 - \delta$:*

$$\|d_T\|_\infty \leq \frac{\|d_0\|_\infty}{T} + \mathcal{O}\left(\frac{\bar{\sigma}\log T}{\lambda_{\min}^{\frac{3}{2}}(1-\gamma)^{\frac{5}{2}}}\sqrt{\frac{\log\left(|\mathcal{S}||\mathcal{A}|T/\delta\right)}{T}} + \frac{\bar{\sigma}\sqrt{\varepsilon}}{\lambda_{\min}(1-\gamma)}\right).$$
(7)

**Discussion of Theorem 2.** To parse the result from Theorem 2, suppose for the moment that there is no corruption, i.e., $\varepsilon = 0$. The dominant convergence rate from Eq. (7) is then $\widetilde{\mathcal{O}}(1/((1-\gamma)^{2.5}\sqrt{T}))$, which matches the recent finite-time rates for Q-learning obtained in Wainwright (2019a); Guannan Qu (2020). Up to polynomial factors in $1/(1-\gamma)$, this rate is known to be minimax optimal (Li et al., 2024). When $\varepsilon \neq 0$, our bound features an additive $O(\sqrt{\varepsilon})$ term that depends only on the small corruption fraction $\varepsilon$, *but crucially is not affected by the magnitude of the injected attacks, highlighting the effectiveness of Algo. 1 in mitigating adversarial influences.* The corruption term is inflated by the noise variance (as one might expect), and by the inverse of the smallest visitation probability $\lambda_{\min}$. Intuitively, poisoning the data for the least-visited state-action pair can make it harder for the learner to reliably estimate the mean reward for this pair. This intuition is formalized by our upper-bound. The main takeaway from Theorem 2 is that despite corruption, `Robust Async-Q` is able to nearly recover the performance of vanilla Q-learning, up to a small $O(\sqrt{\varepsilon})$ term. **This is the first result on the adversarial robustness of $Q$-learning under asynchronous sampling.**

**Fundamental Lower Bound.** One might ask: Is the additive $O(\sqrt{\varepsilon})$ term in (7) unavoidable for our problem of interest? We now show that this is indeed the case by establishing an information-theoretic lower bound. To do so, it suffices to consider a simpler *synchronous* observation model (Kearns & Singh, 1998; Even-Dar et al., 2003; Sidford et al., 2018) for the learner, where it gets to observe data for **every** state-action pair $(s,a) \in \mathcal{S} \times \mathcal{A}$ at each time-step $t$. More precisely, in each iteration $t$, we toss a biased coin with probability of heads $1 - \varepsilon$, independently of the past. If the coin lands heads, for each $(s,a) \in \mathcal{S} \times \mathcal{A}$, the learner observes $y_t(s,a) \sim \mathcal{R}(s,a)$. If it lands tails, for each $(s,a) \in \mathcal{S} \times \mathcal{A}$, $y_t(s,a) \sim \mathcal{Q}$, where recall that $\mathcal{Q}$ is an *arbitrary* adversarial distribution. Let us use $\mathcal{H}(\varepsilon, \bar{\sigma}, \mathcal{Q})$ to collectively represent the set of all MDPs and observation models with finite state and action spaces, where the true underlying reward distributions have bounded mean rewards and variance at most $\bar{\sigma}^2$, and the observed rewards are generated based on the synchronous Huber contamination model described above. With a slight abuse of notation, we will use $Q^* \in \mathcal{H}(\varepsilon, \bar{\sigma}, \mathcal{Q})$ to imply that $Q^*$ is the optimal value function of an MDP consistent with the class of MDPs contained in $\mathcal{H}$. Now, suppose the learner is presented with $T$ independent data sets

$\tilde{D}_1, \ldots, \tilde{D}_T$, where $\tilde{D}_t = \{s_t(s,a), y_t(s,a)\}_{(s,a) \in \mathcal{S} \times \mathcal{A}}$, and $s_t(s,a) \sim \mathcal{P}(\cdot|s_t, a_t)$. An estimator $\hat{Q}_T$ of $Q^*$ is some measurable function of these $T$ sets. We then have the following *fundamental* lower bound.

**Theorem 3.** *(**Lower Bound**) There exists a universal constant $\tilde{c} > 0$ such that*

$$\inf_{\hat{Q}_T} \sup_{Q^* \in \mathcal{H}(\varepsilon, \bar{\sigma}, \mathcal{Q})} \mathbb{P}\left(\|\hat{Q}_T - Q^*\|_\infty \geq \frac{\tilde{c}\bar{\sigma}\sqrt{\varepsilon}}{(1-\gamma)}\right) \geq \frac{1}{4}.$$

**Main Takeaway.** From the above result, we infer that the additive corruption term in (7) is tight in its dependence on the corruption fraction $\varepsilon$, the discount factor $\gamma$, and the noise variance $\bar{\sigma}$. Interestingly, these dependencies persist even when the learner is presented with a more favorable observation model where it gets to observe rewards for all the state-action pairs simultaneously at each time-step. We note that similar additive corruption terms have been proven to be unavoidable in prior works on robust mean estimation (Chen et al., 2015; Lai et al., 2016; Cheng et al., 2019; Dalalyan & Minasyan, 2022), and multi-armed bandits with reward corruptions (Lykouris et al., 2018; Gupta et al., 2019; Kapoor et al., 2019). Our work is the first to show that such a term is also unavoidable for $Q$-learning. **Collectively, Theorems 2 and 3 establish the near-optimality of our proposed approach**, and paint a fairly complete picture for the theme of adversarial robustness in $Q$-learning.

Having established the near-optimality of our approach, the next two sections of the paper are devoted to further generalizing our results to scenarios where bounds on the reward means and variances are unknown (Section 4), and when the data is sampled in a Markovian manner (Section 4.1). Before jumping into these sections, we provide brief proof sketches for Theorems 2 and 3.

**Proof Sketch of Theorem 2.** We start by writing down a recursion for the error $d_t = Q_t - Q^*$ that features two main terms: a noise term that exhibits a martingale difference structure, and a term that captures the effect of adversarial corruption. *The main challenge in the analysis arises from the fact that these two terms are coupled*; notably, this difficulty does not arise when one analyzes the standard $Q$-learning algorithm. The coupling is a consequence of the fact that the noise term involves the iterate $Q_t$ which, in turn, is affected by the adversarially corrupted reward observations. Our proof strategy is to first control the effect of adversarial corruption via the following lemma, which is the key new tool in our overall analysis.

**Lemma 1.** *Suppose Assumption 1 holds. With probability at least $1 - \delta/2$, the following items are true for all $t > \bar{T}$: (i) $\tilde{r}_t(s_t, a_t) = \bar{r}_t(s_t, a_t)$, and (ii):*

$$|\tilde{r}_t(s_t, a_t) - R(s_t, a_t)| \leq \mathcal{O}\left(\bar{\sigma}\left(\sqrt{\frac{\log(8/\delta_1)}{\lambda_{\min} t}} + \sqrt{\varepsilon}\right)\right).$$

Lemma 1 tells us that after the burn-in time $\bar{T}$ is passed, with high-probability, no thresholding will take place, i.e., $\tilde{r}_t(s_t, a_t) = \bar{r}_t(s_t, a_t)$, and the reward proxies that we plug into our update rule (6) will be sufficiently accurate estimates of the true reward functions. The main difficulty in establishing Lemma 1 is that the number of times each state-action pair has been visited up to any time-step $t$ is a *random variable*. As such, we first use Bernstein's inequality to create a "good event" on which, after time $\bar{T}$, each state-action pair is sufficiently visited. We then carefully condition on this event to exploit the bound in (3). Lemma 1 helps us control the effect of adversarial corruption. To control the noise term, we first use the adaptive thresholding idea and an inductive argument to establish that the iterate sequence $\{Q_t\}$ generated by `Robust Aysnc-Q` is uniformly bounded, and then apply Azuma-Hoeffding. The complete details of the proof are deferred to Appendix C.

**Proof Sketch of Theorem 3.** The proof of this result relies on carefully constructing two different MDPs and associated attack distributions, such that (i) the optimal Q-functions in the two MDPs differ in magnitude by $\Omega(\bar{\sigma}\sqrt{\varepsilon}/(1-\gamma))$; and (ii) the distributions of the observed reward samples in the two MDPs are indistinguishable to a learner. The details are provided in Appendix D.

**On exact recovery under noise-free rewards.** In Section 2, motivated by practical considerations and to keep our developments general, we considered a noisy observation model where even in the absence of corruptions, when the learner visits a state-action pair $(s, a)$, it only gets to see a noisy version of the true mean reward $R(s, a)$. In what follows, we briefly explain that if the reward observation model is *deterministic*, that is, visiting $(s, a)$ causes the learner to observe $R(s, a)$ exactly (in the absence of corruption), then one can recover the exact same guarantees as vanilla Q-learning without corruption, using a simpler version of our algorithm. To see this, fix any state-action pair $(s, a)$, and let $N_t(s, a)$ represent the number of times $(s, a)$ has been visited up to time $t$ (including time $t$). On an average, $N_t(s, a)$ is $\lambda(s, a)t$. Furthermore, under our Huber contamination model, on an average, $\varepsilon\lambda(s, a)t$ of the observations for $(s, a)$ are corrupted. Crucially, (i) since $\varepsilon < 1/2$ by assumption, uncorrupted samples are in the majority, and (ii) every uncorrupted sample is precisely $R(s, a)$ (since there is no additional uncertainty caused by noise). As such, simply taking a median of the observations for each state-action pair $(s, a)$ enables the learner to *exactly* recover $R(s, a)$, i.e., there is no bias in the reward estimation. Once this is done, our algorithm evolves exactly as the standard Q-learning algorithm, and hence, does not incur the additional additive $\mathcal{O}(\sqrt{\varepsilon})$ term that shows up in (7). To make the above argument precise, we need to account for the concentration of $N_t(s, a)$ around its mean value, and also for the concentration of the number of corrupted samples around

its mean value, both of which can be done via an application of Bernstein's inequality. Such an analysis would reveal that after a suitably long burn-in time after which concentration kicks in, uncorrupted samples for each state-action pair would be in the majority, and a simple median would suffice to recover the true reward means. We should note here that our discussion above does not contradict the lower bound in Theorem 3 since the bound scales with the noise variance which is zero under deterministic rewards.

# 4. Reward-Agnostic Robust Q-learning

In the previous section, we developed a robust variant of the asynchronous $Q$-learning algorithm that achieves near-optimal guarantees under reward corruption, while assuming access to upper bounds on just the first two moments of the true reward distributions. These assumptions enabled us to precisely design the adaptive threshold $G_t$ in Eq. (5) to safeguard against adversarial outliers. We now ask: *Is it possible to preserve the same rates as before while assuming no prior knowledge at all about the reward statistics?* This is a challenging question motivated by real-world applications where precise bounds on the moments of the reward distributions may not be available to the learner. The lack of knowledge of the parameter $\tilde{\sigma} = \max\{\bar{R}, \bar{\sigma}\}$, which previously played a central role in designing the threshold function $G_t$, now creates more uncertainty for the learner to contend with. Nonetheless, we establish that one can continue to enjoy the same bounds as before with two simple modifications to Algorithm 1 that we now describe.

`Modification 1.` Our key idea is to use a polynomial function of time, denoted by $m(t) = t^p$, as a proxy for the *unknown* upper-bound $\tilde{\sigma}$. Any positive integer $p \geq 1$ will suffice for our purpose; we will comment on the choice of $p$ shortly. The new threshold is

$$
\tilde{G}_t = \begin{cases} 0, & \text{if } t \leq \bar{T}, \\ \mathcal{C}\, m(t)\left(\sqrt{\frac{4\log(8/\delta_1)}{3\lambda_{\min}t}} + \sqrt{\varepsilon}\right) + m(t), & \text{if } t > \bar{T}, \end{cases}
$$
(8)

where the universal constant $\mathcal{C}$ and the burn-in time $\bar{T}$ are defined as in Section 3. The intuition for this proxy is quite simple: since $\tilde{\sigma}$ is a constant, any growing function of time will eventually dominate $\tilde{\sigma}$, after which point, the new threshold $\tilde{G}_t$ will serve as an upper-bound for the threshold $G_t$ that we designed in (5). Lemma 1 will then kick in.

`Modification 2.` To make the analysis go through, we will require the failure probability parameter $\delta_1$, that is fed as input to the `TRIM` function, to be finer than before: we set $\delta_1 = \delta^2 / \left(512\,|\mathcal{S}|^2|\mathcal{A}|^2T^{2p+3}\right)$, where $p$ is the same parameter that appears in $m(t)$. Thus, the overall change to Algorithm 1 involves the new choice of $\delta_1$ in line 5, and the replacement of $G_t$ by $\tilde{G}_t$ in line 6. We call this new reward-agnostic variant `Robust Async-RAQ`. Our main

result for this variant is as follows.

**Theorem 4.** *Suppose the conditions in Theorem 2 hold. Then, given any $\delta \in (0,1)$,* `Robust Async-RAQ` *satisfies the following with probability (w.p.) at least $1-\delta$:*

$$
\|d_T\|_\infty \leq \frac{\|d_0\|_\infty}{T} + \mathcal{O}\left(\frac{\tilde{\sigma}^{1+1/2p}\log T}{\lambda_{\min}^{\frac{3}{2}}(1-\gamma)^{\frac{5}{2}}}\sqrt{\frac{\log\left(|\mathcal{S}||\mathcal{A}|T/\delta\right)}{T}} + \frac{\bar{\sigma}\sqrt{\varepsilon}}{\lambda_{\min}(1-\gamma)}\right).
$$
(9)

**Main Takeaway.** Comparing equations (9) and (7), we note that even with no prior knowledge of the reward statistics, `Robust Async-RAQ` is able to remarkably preserve the same near-optimal rates we established before, up to a slight inflation in the dependence on $\tilde{\sigma}$ in the dominant term. This goes on to show the flexibility of our overall framework in accommodating asynchronous sampling, adversarial corruptions, and completely unknown reward statistics. Now, let us comment on the choice of $p$ in the function $m(t)$. Making $p$ larger would lead to a shorter wait time before the modified threshold $\tilde{G}_t$ dominates the true threshold $G_t$, and an improvement in dependence on $\tilde{\sigma}$ in (9). However, a larger $p$ would also imply a smaller failure probability $\delta_1$, which will eventually cause our overall bound to get scaled linearly by $p$, since $\delta_1$ fortunately appears inside a logarithm. Due to the latter fact, up to constant factors, making $p$ large does not degrade our final bound.

**Challenges and Novelty in the Proof of Theorem 4.** In addition to the proof challenges for Theorem 2 we discussed earlier, the modified threshold $\tilde{G}_t$ introduces various new subtleties and technical challenges in the proof, which precludes the use of standard concentration tools used typically in the analysis of RL algorithms. Like before, to exploit the martingale structure of the noise term that shows up in our analysis, we need a uniform bound on $\|Q_t\|_\infty$. While this bound was $\mathcal{O}(1)$ previously, in light of the new threshold, it now becomes on the order of $\mathcal{O}(T^p)$. Using this new upper bound with the standard Azuma-Hoeffding inequality will lead to a vacuously large rate that does not reflect the "typical" behavior of the algorithm. Thus, we need a much more intricate analysis than before. Our key observation is that the iterate sequence $\{Q_t\}$ generated by `Robust Async-RAQ` exhibits an interesting structure: they are bounded by a crude $\mathcal{O}(T^p)$ term deterministically, and a finer $\mathcal{O}(1)$ term with high-probability. This observation does not immediately resolve our problem since we now need a finer version of Azuma-Hoeffding that can exploit the structure identified above. In this regard, some common variants of Azuma-Hoeffding for discrete probability spaces (Chung & Lu, 2006) and martingale differences with sub-Gaussian tails (Shamir, 2011) are inadequate for our purpose, since the martingale difference in our setting neither belongs to a discrete space nor is sub-Gaussian. Fortunately, we are able to leverage an elegant result from Shamir & Spencer (1987) on martingale differences that admit a

coarse bound deterministically, and a finer bound with high-probability. We record this result below.

**Theorem 5.** *(Shamir & Spencer, 1987) Let $X_0, \ldots, X_n$ be a martingale with $X_0$ constant, such that:*

*(i)* $\mathbb{P}(|X_{i+1} - X_i| \leq c_i) \geq 1 - r, \text{ for } 0 \leq i < n.$

*(ii)* $|X_{i+1} - X_i| \leq b_i$, *deterministically.*

*Assume $b_i \cdot r^{\frac{1}{2}} \leq c_i$. Then, the following bound holds with probability at least $1 - \delta - 2nr^{1/2}$:*

$$|X_n - X_0| \leq \sqrt{\left(32 \sum_{i=1}^{n} c_i^2\right) \log\left(\frac{2}{\delta}\right) + \sum_{i=0}^{n-1} b_i \cdot r^{1/2}}. \quad (10)$$

This refined variant of Azuma-Hoeffding is the key new tool in our analysis, *and, as far as we are aware, has not appeared before in prior finite-time analysis of RL algorithms.* Thus, proving `Robust Async-RAQ` requires substantial new ideas beyond prior work; we relegate the full argument to Appendix E.

### 4.1. Extension to Markovian Sampling

We now explain how our developments can be extended to account for single-trajectory Markov data. Previously, we assumed that at each time-step $t$, $s_t$ is sampled in an i.i.d. manner from the stationary distribution $\pi$ of the Markov chain induced by the behavior policy $\mu$. We now relax this assumption, and let $s_t$ be the state of this Markov chain at time $t$. It is easy to verify that $Z_t = (s_t, a_t, s_{t+1})$ is also a Markov chain, and that this chain is ergodic based on Assumption 1 (Chen et al., 2019). Using this, we now propose a simple modification to `Robust Async-RAQ` that ignores certain data points. To explain this modification, let $\Omega$ represent the state space for the Markov chain $\{Z_t\}$, and let $\rho$ be its stationary distribution. Following Dorfman & Levy (2022), define $d_{mix}(t) := \sup_{Z \in \Omega} D_{TV}(\mathbb{P}(Z_t \in \cdot | Z_0 = Z), \rho)$, where $D_{TV}$ is used to represent the total variation distance between probability measures. We now define the mixing time as $\bar{\tau} := \inf\{t | d_{mix}(t) \leq 1/4\}$. Finally, we define a "block" parameter $\tau := \lceil \ell\bar{\tau} \rceil$, where $\ell = \lceil \log(2T/\delta)/\log 2 \rceil$. The only modification to `Robust Async-RAQ` is that the agent now uses every $\tau$-th sample, and drops the rest. For this variant (described in Appendix F), we have the following result.

**Theorem 6.** *Suppose Assumption 1 holds, and $Z_0 \sim \rho$. Then, given any $\delta \in (0, 1)$, for suitably chosen $\alpha$ and large enough $T$, the following holds w.p. at least $1 - \delta$:*

$$\|d_T\|_\infty \leq \frac{\|d_0\|_\infty}{T} + \mathcal{O}\left(\frac{\tilde{\sigma}^{1+1/2p} \log T}{\lambda_{\min}^{\frac{3}{2}}(1-\gamma)^{\frac{5}{2}}} \sqrt{\frac{\tau \log(|\mathcal{S}||\mathcal{A}|T/\delta)}{T}} + \frac{\bar{\sigma}\sqrt{\varepsilon}}{\lambda_{\min}(1-\gamma)}\right). \quad (11)$$

**Main Takeaway.** Comparing Theorems 4 and 6, we note that despite Markov sampling, we are able to essentially preserve the same bounds as in the i.i.d. case up to an inflation by a factor of $\sqrt{\tau}$, where $\tau$ captures the mixing time of the Markov chain (up to logarithmic factors). Such an inflation by the mixing time shows up for vanilla $Q$-learning as well (Guannan Qu, 2020). The assumption that $Z_0 \sim \rho$ is only made to simplify some of the algebra as in prior RL work (Bhandari et al., 2018; Dorfman & Levy, 2022). Overall, **Theorem 6 establishes the first robustness guarantees for $Q$-learning with single-trajectory Markovian data.**

**Remark 2.** The sub-sampling procedure used in this section requires a tuning parameter that depends on the mixing time $\tau$. Such knowledge is commonly assumed in the design and analysis of RL and stochastic-approximation algorithms with Markovian data, including asynchronous $Q$-learning (Guannan Qu, 2020; Li et al., 2024) and more general Markovian stochastic approximation schemes (Srikant & Ying, 2019; Chen et al., 2022; Patil et al., 2023; Mitra, 2024). When $\tau$ is not known exactly, it can in principle be estimated from trajectory data using existing methods for estimating the mixing time of ergodic Markov chains (Wolfer & Kontorovich, 2019; Hsu et al., 2015).

**Simulations.** Although our contribution is theoretical, in Appendix G, we provide several simulations on different environments such as `Grid-world`, `FrozenLake`, `CliffWalking`, and `Taxi` to corroborate our theory. Across these environments, we consistently observe that vanilla $Q$-learning incurs significant errors, whereas our proposed robust variants converge to a neighborhood of $Q^*$.

## 5. Conclusion and Future Work

We studied the problem of learning an optimal policy in RL subject to heavy-tailed and adversarially corrupted rewards. To achieve this goal, we proposed a novel robust variant of the classical $Q$-learning algorithm that accounts for asynchronous, single-trajectory data, and requires no prior knowledge of the statistics of the true reward distributions. We established that the finite-time guarantees of our proposed algorithm match that of vanilla $Q$-learning (under no attacks), up to an additive term proportional to the corruption fraction. To complement this upper bound, we established an information-theoretic lower bound, showing that the corruption-dependent term is fundamental and cannot be avoided. Overall, our work takes a significant step toward advancing the current theoretical understanding of RL in harsh, adversarial environments. Future work includes extending the framework to corruptions in the full feedback stream, including both transitions and rewards, as well as to function approximation settings. Another interesting direction is to reduce the memory of our algorithm using online robust estimators (Yao & Sundaram, 2022).

## Acknowledgments

This work is supported by the following grant from the National Science Foundation: NSF CAREER award 2542396.

## Impact Statement

This paper presents work whose goal is to advance the field of Machine Learning. We cannot think of any potential societal consequences of our work that must be specifically highlighted here.

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

# A. Additional Literature Survey and Standard Results

In this section, we provide a more detailed discussion of the relevant threads of literature.

1. $Q$**-learning.** The $Q$-learning algorithm was first introduced by Watkins and Dayan in Watkins & Dayan (1992). There is a long line of work that has explored the asymptotic performance of $Q$-learning algorithms in the limit of infinite samples; see, for instance, Borkar (2009); Tsitsiklis (1994); Jaakkola et al. (1993), using ideas from stochastic approximation theory (Borkar, 2009; Borkar & Meyn, 2000). A more recent strand of literature has focused on the non-asymptotic analysis of $Q$-learning and its variants (Shah & Xie, 2018; Wainwright, 2019a; Guannan Qu, 2020; Li et al., 2024), accounting also for function approximation (Chen et al., 2019). While we build on some of the techniques in these papers, our work departs from RL line of literature by considering the robustness of $Q$-learning to adversarial perturbations - a topic that has not been explored in the papers mentioned above. For a detailed literature review on $Q$-learning, we refer the reader to Li et al. (2024).

2. **Stochastic Approximation.** Our work is broadly related to the area of stochastic approximation algorithms in reinforcement learning, which includes $Q$-learning (Watkins & Dayan, 1992) and TD learning (Sutton, 1988) as special cases. As mentioned earlier, the asymptotic theory of such algorithms is rich (Tsitsiklis & Van Roy, 1997). Finite-time results, however, are much more recent. Initial finite-time results under the i.i.d. sampling model (that we also consider in this work) were provided in Korda & La (2015); Lakshminarayanan & Szepesvári (2017); Dalal et al. (2018); Narayanan & Szepesvári (2017); Prashanth et al. (2021). The extension to the Markov setting was first derived in (Bhandari et al., 2018) for a projected TD learning algorithm. The assumption of the projection step was later removed in Srikant & Ying (2019) and Mitra (2024). Some other relevant recent works on the finite-time theory of TD learning include (Liu & Olshevsky, 2021; Patil et al., 2023; Khamaru et al., 2020; Xie et al., 2025; Lee & Orabona, 2025). Each of the papers mentioned above studies the basic versions of the concerned algorithms, where updates are made using noisy versions of some true underlying operator. Our work analyzes the robustness of these algorithms to adversarial perturbations. On a related note, we mention here that other types of perturbations resulting from communication-induced challenges (e.g., delays and compression) have been explored recently in (Mitra et al., 2023; Adibi et al., 2024; Dal Fabbro et al., 2024).

3. **Reward Contamination in Multi-Armed Bandits.** A large body of work has explored the effects of reward contamination on the performance of stochastic bandit problems, both for the unstructured multi-armed bandit (MAB) setting (Jun et al., 2018; Liu & Shroff, 2019; Kapoor et al., 2019; Lykouris et al., 2018; Gupta et al., 2019), and also for structured linear bandits (Bogunovic et al., 2020; Garcelon et al., 2020; Bogunovic et al., 2021; He et al., 2022). The basic premise in these papers is that an adversary can modify the true stochastic reward/feedback on certain rounds; a corruption budget $C$ captures the total corruption injected by the adversary over the horizon $T$. In particular, the authors in (Kapoor et al., 2019) study a Huber-contaminated reward model like us, where in each round, with probability $\eta$ (independently of the other rounds), the attacker can bias the reward seen by the learner. A fundamental lower bound of $\Omega(\eta T)$ on the regret is also established in (Kapoor et al., 2019). While our reward contamination model is directly inspired by the above line of work, we emphasize that the stochastic approximation setting we study here fundamentally differs from the bandit problem. As such, our algorithms and proof techniques are also different from the bandit literature.

4. **Robust Statistics.** The study of computing different statistics (e.g., mean, variance, etc.) of a data set in the presence of outliers was pioneered by Huber (Huber, 1992; 2004). Since then, the field of robust statistics has significantly advanced, with more recent work focusing on computationally tractable algorithms in the high-dimensional setting (Lai et al., 2016; Chen et al., 2015; Minsker, 2018; Cheng et al., 2019; Lugosi & Mendelson, 2021; Dalalyan & Minasyan, 2022). Our paper builds on this rich line of work and uses it in the context of RL.

5. **Relation to Closely Related Robust RL Results.** The present paper substantially extends Maity & Mitra (2024), which considers synchronous i.i.d. sampling and assumes prior bounds on the first two moments of the reward distributions. In contrast, the present analysis addresses asynchronous single-trajectory sampling with temporal correlations, requiring new blocking and coupling arguments; even without corruption, this transition is technically non-trivial (Guannan Qu, 2020; Bhandari et al., 2018). Our paper also studies a reward-agnostic regime in which the learner has no prior knowledge of the reward distribution, while the adversary has full knowledge of the MDP; the analysis in this regime relies on a refined Azuma–Hoeffding argument. Moreover, unlike Maity & Mitra (2024), this paper establishes an information-theoretic lower bound in Theorem 3.

The results presented in this paper are also related to the recent work of Maity & Mitra (2025), which investigates adversarial robustness in the context of temporal difference (TD) learning with linear function approximation. That said, there are considerable differences in the problem formulation, assumptions, algorithm design, and analysis techniques, as we explain next. First, Maity & Mitra (2025) addresses the *policy evaluation* problem in RL, whereas our focus in this paper is on the more challenging *control problem*. Second, the algorithm design and analysis in Maity & Mitra (2025) exploit the linearity of the operator associated with TD learning under linear function approximation; in contrast, the Bellman optimality operator for our problem is non-linear. Third, the performance guarantees in Maity & Mitra (2025) are expressed in terms of the expected mean-squared $\ell_2$ error, while the results in this paper are established under the $\ell_\infty$ error metric. The $\ell_2$ norm, being induced by an inner product, is particularly well suited for gradient-based optimization-style analyses that do not readily carry over to the $\ell_\infty$ metric. Lastly, and most importantly, the robustness guarantees in Maity & Mitra (2025) hinge on prior knowledge of the reward statistics, namely, an upper bound on both the reward means and variances. In sharp contrast, the latter part of this paper establishes that robustness guarantees are attainable *even in the complete absence of such statistical knowledge*, as rigorously formalized in Theorems 4 and 6.

### A.1. Useful Facts and Results

In this section, we compile a few useful results that will be used by us throughout the proofs. We start by listing some properties of the Bellman optimality operator $\mathcal{T} : \mathbb{R}^{|\mathcal{S}| \times |\mathcal{A}|} \to \mathbb{R}^{|\mathcal{S}| \times |\mathcal{A}|}$ given by:

$$(\mathcal{T}Q)(s,a) = R(s,a) + \gamma \mathbb{E}_{s' \sim \mathcal{P}(\cdot|s,a)} \left[ \max_{a' \in \mathcal{A}} Q(s',a') \right]. \tag{12}$$

It turns out that the optimal state-action value function $Q^*$ is a fixed point of $\mathcal{T}$, i.e., $\mathcal{T}Q^* = Q^*$. Furthermore, $\mathcal{T}$ is contractive in the $\infty$-norm, a fact that we will exploit in all our main convergence proofs. Formally, the Bellman optimality operator satisfies the following contraction property $\forall Q_1, Q_2 \in \mathbb{R}^{|\mathcal{S}| \times |\mathcal{A}|}$:

$$\|\mathcal{T}Q_1 - \mathcal{T}Q_2\|_\infty \leq \gamma \|Q_1 - Q_2\|_\infty. \tag{13}$$

The above facts can be found in Sutton & Barto (2018). Next, we record all the basic probabilistic machinery that will be needed throughout the paper.

**Lemma 2.** *(**Bernstein's Inequality** (Chung & Lu, 2006)) If $X_1, X_2, \ldots, X_N$ are independent random variables with $\mathbb{P}(|X_i| \leq c) = 1$ and common mean $\mu$, then for any $\varepsilon > 0$:*

$$\mathbb{P}(|\bar{X}_N - \mu| > \varepsilon) \leq 2 \exp \left\{ -\frac{N\varepsilon^2}{2\sigma^2 + \frac{2c\varepsilon}{3}} \right\}, \tag{14}$$

*where $\bar{X}_N = \frac{1}{N}\sum_{i=1}^N X_i$ and $\sigma^2 = \frac{1}{N}\sum_{i=1}^N Var(X_i)$.*

**Lemma 3.** *(**Azuma-Hoeffding** (Chung & Lu, 2006)) Let $Z_1, Z_2, Z_3, \ldots$ be a martingale difference sequence with $|Z_i| \leq c_i$ for all $i \in \mathbb{N}$, where each $c_i$ is a positive real. Then, for all $\lambda \geq 0$:*

$$\mathbb{P}\left( \left| \sum_{i=1}^n Z_i \right| \geq \lambda \right) \leq 2e^{-\frac{\lambda^2}{2\sum_{i=1}^n c_i^2}}.$$

**Lemma 4.** *(**Tower property for nested $\sigma$-algebras**) (Durrett, 2019) Let $(\Omega, \mathcal{F}, \mathbb{P})$ be a probability space, let $\mathcal{B}_1 \subseteq \mathcal{B}_2 \subseteq \mathcal{F}$ be $\sigma$-algebras, and let $X$ be integrable. Then, the following holds almost surely:*

$$\mathbb{E}\big[\, \mathbb{E}[X \mid \mathcal{B}_2] \mid \mathcal{B}_1 \,\big] = \mathbb{E}[X \mid \mathcal{B}_1].$$

**Lemma 5.** *(**Bretagnolle–Huber Inequality** (Lattimore & Szepesvári, 2020, Theorem 14.2)) Let $P$ and $Q$ be two probability measures on the same measurable space $(\Omega, \mathcal{F})$, and let $\mathcal{A} \in \mathcal{F}$ be any arbitrary event. Then,*

$$P(\mathcal{A}) + Q(\mathcal{A}^c) \geq \frac{1}{2} \exp\big( -KL(P \,\|\, Q)\big),$$

*where $\mathcal{A}^c$ is the complement of the event $\mathcal{A}$, and $KL(P \,\|\, Q)$ is the Kullback–Leibler distance between $P$ and $Q$.*

## A.2. Comments on Assumptions and Algorithmic Implementation

• **On the choice of the behavior policy.** All our main convergence results rely on Assumption 1 which requires the Markov chain induced by the behavior policy $\mu$ to be aperiodic and irreducible. Although this assumption is extremely standard, for the sake of completeness, we now provide some details on how such an assumption can be satisfied. In particular, as we explain next, given *any* stochastic policy $\mu$ that places non-zero mass on each action, i.e., $\mu(a|s) > 0, \forall(s, a)$, Assumption 1 will hold under fairly mild conditions on the underlying MDP.

To see this, note that under policy $\mu$, the probability of going from state $s$ to state $s'$ is given by

$$P_\mu(s, s') = \sum_{a \in \mathcal{A}} \mu(a|s) P(s'|s, a).$$

Now, consider the standard one-to-one correspondence between a Markov transition matrix $P_\mu$ and a directed graph $\mathcal{G}_\mu := \mathcal{G}(\mathcal{S}, E_\mu)$, where the nodes of the graph are the states of the Markov chain, and an edge $(s, s') \in E_\mu$ if and only if $P_\mu(s, s') > 0$. For each action $a$, let the graph $\mathcal{G}_a$ be defined accordingly. Since the behavior policy places non-zero mass on every action at every state, we make the key observation that the graph corresponding to $\mu$ is simply the union of the graphs corresponding to the individual actions, i.e.,

$$\mathcal{G}_\mu = \bigcup_{a \in \mathcal{A}} \mathcal{G}_a.$$

Given the equivalence between irreducibility and strong-connectivity, it follows that the chain induced by $\mu$ is irreducible if and only if the union-graph $\bigcup_{a \in \mathcal{A}} \mathcal{G}_a$ is strongly-connected.

For aperiodicity, a standard sufficient condition is for $\mathcal{G}_\mu$ to have self-loops, i.e., we want $P_\mu(s, s) > 0, \forall s \in \mathcal{S}$. Following a similar reasoning as above, this can be ensured if for each state $s \in \mathcal{S}$, there exists an action $a(s)$ such that $P(s|s, a(s)) > 0$. In fact, since $\mathcal{G}_\mu$ is assumed to be strongly connected (for irreducibility), the existence of even one node with a self-loop is enough to guarantee aperiodicity.

• **On the choice of the parameter** $\mathcal{C}$ **in** (5)**.** The constant $\mathcal{C}$ in Theorem 1, which also appears in our threshold $G_t$ in (5), is a universal constant from the trimmed mean guarantee in Lugosi & Mendelson (2021). A value of $\mathcal{C} = 20\sqrt{3}$ suffices in Lugosi & Mendelson (2021). After the extra steps in our analysis (see Appendix B), a value of $\mathcal{C} \approx 100$ suffices for our case, which we use in our experiments.

• **On the choice of the parameter** $p$ **in** (8)**.** The choice of $p$ in the modified threshold $\tilde{G}_t$ in (8) not only affects the burn-in time, but also the main convergence bound in Theorem 4. From the dependence of $\delta_1$ on $p$, the burn-in time trade-off is reflected in a term of the form $\max\{p, \tilde{\sigma}^{1/p}\}$, while the trade-off in the final convergence bound shows up as $\sqrt{p}\tilde{\sigma}^{1/p}$. One can choose a theoretically optimal $p$ to balance either of these metrics by using the above explicit expressions. However, such an optimal $p$ would depend on the *unknown* $\tilde{\sigma}$. Fortunately, *any* $p \geq 1$ works in our analysis; a $p$ in the range $5 - 10$ typically suffices for practical purposes as we explain below.

Recall that $\bar{R}$ and $\bar{\sigma}$ denote upper bounds on the reward means and the noise standard deviation, respectively, and we define $\tilde{\sigma} = \max(\bar{R}, \bar{\sigma})$. In Section 3, when designing the threshold $G_t$, we assumed $\tilde{\sigma}$ was known; in the agnostic setting of Section 4, we instead used $m(t) = t^p$ as a proxy for $\tilde{\sigma}$ in the threshold $\tilde{G}_t$. Comparing Eq. (8) with Eq. (5), once $t^p > \tilde{\sigma}$, the proxy threshold $\tilde{G}_t$ overestimates $G_t$, effectively reducing the problem to the known $\tilde{\sigma}$ case. For instance, if $\tilde{\sigma} = 1000$ and $p = 5$, this condition is met in only four steps – typically far fewer than the burn-in period $\bar{T}$, before which no updates occur. Thus, values of $p$ below 10 are sufficient in most practical scenarios. From a theoretical standpoint, in Theorem 4 the effect of $p$ appears through $\tilde{\sigma}^{1+1/(2p)}$ and a $\sqrt{p}$ factor, the latter arising from the choice of $\delta_1$ in `Modification 2` and is absorbed into the $\mathcal{O}$ notation since $p$ is treated as a constant. As $p$ increases, $\tilde{\sigma}^{1+1/(2p)}$ approaches $\tilde{\sigma}$ as in Eq. (7), and the extra cost from $\sqrt{p}$ remains modest.

## B. Analysis of the Trimmed Mean Estimator under Huber Contamination

---

**Algorithm 2** Univariate Trimmed-Mean Estimator from (Lugosi & Mendelson, 2021) (`trimSC`)

---

**Require:** Corrupted Dataset $\tilde{\mathcal{D}} = \{X_1, X_2, \ldots, X_M\} = \mathcal{D}_1 \oplus \mathcal{D}_2$, such that $|\mathcal{D}_i|_{i \in \{1,2\}} = M/2$; corruption fraction $\varepsilon$; confidence level $\delta$.

1: Set $\zeta = 8\varepsilon + 24\frac{\log(4/\delta)}{M}$.
2: Let $X_1^* \leq X_2^* \leq \cdots \leq X_{M/2}^*$ represent a non-decreasing arrangement of $\mathcal{D}_1$.
3: Compute `quantiles`: $\alpha = X_{\zeta M/2}^*, \quad \beta = X_{(1-\zeta)M/2}^*$.
4: Define the function $\phi_{\alpha,\beta}(x)$ as

5:
$$\phi_{\alpha,\beta}(x) = \begin{cases} \beta & \text{if } x > \beta \\ x & \text{if } x \in [\alpha, \beta] \\ \alpha & \text{if } x < \alpha \end{cases}$$

6: Compute the `trimmed mean`: $\hat{\mu}_X = (2/M) \sum_{X_i \in \mathcal{D}_2} \phi_{\alpha,\beta}(X_i)$.

---

We start by briefly recalling the strong-contamination data model studied in Lugosi & Mendelson (2021). Consider a data set $\mathcal{D}$ comprising of $M$ i.i.d. samples of a scalar random variable $X$ with mean $\mu_X$ and variance $\sigma_X^2$. An adversary arbitrarily perturbs up to $\varepsilon M$ of the samples within $\mathcal{D}$ to produce a corrupted data set $\tilde{\mathcal{D}}$; here, $\varepsilon \in [0, 1/2)$ is the fraction of corrupted data. Using $\tilde{\mathcal{D}}$, the corruption fraction $\varepsilon$, and a confidence parameter $\delta$ as inputs, the trimmed mean estimator from Lugosi & Mendelson (2021) produces a robust estimate $\hat{\mu}_X$ of the mean $\mu_X$ in the following way. The data set $\tilde{\mathcal{D}}$ is divided into two equal parts of $M/2$ samples each. The first part is used to compute empirical quantiles for filtering out extreme values. The estimate $\hat{\mu}_X$ is then simply an average of only those data samples in the second part that fall within the computed quantiles. Let $\hat{\mu}_X = \text{trimSC}[\tilde{\mathcal{D}}, \varepsilon, \delta]$ be used to succinctly represent the output of the trimmed mean estimator described above, and outlined in Algorithm 2; here, the subscript 'SC' is used to represent the strong contamination attack model considered in Lugosi & Mendelson (2021). For this setting, we have the following guarantee from Lugosi & Mendelson (2021).

**Theorem 7.** *(Lugosi & Mendelson, 2021, Theorem 1) Let $\delta \in (0,1)$ be such that $\delta \geq 4e^{-M/2}$, and suppose $\hat{\mu}_X = \text{trimSC}[\tilde{\mathcal{D}}, \varepsilon, \delta]$. Then, there exists an universal constant c, such that with probability at least $1 - \delta$,*

$$|\hat{\mu}_X - \mu_X| \leq c\sigma_X \left( \sqrt{\varepsilon} + \sqrt{\frac{\log(4/\delta)}{M}} \right). \tag{15}$$

Our goal in this section is to show how the same result can be extended to account for the Huber contamination model of interest to us, where each data sample in $\mathcal{D}$ is arbitrarily corrupted with probability $\varepsilon$. For future reference, we will call the Huber-contaminated data set $\mathcal{D}'$. As we will show, all that needs to happen is that Algorithm 2 needs to be invoked with a slightly larger corruption fraction that will follow from our subsequent analysis.

**Step 1. Bounding the number of corrupted samples.** We begin with a dataset $\mathcal{D}$ consisting of $M$ samples, where each sample is independently corrupted with probability $\varepsilon$, as specified in the corruption model described in Section 2. Our first objective is to bound the total number of corrupted samples in this dataset (with high probability). To this end, we define an event $\mathcal{W}$, where the number of corrupted samples does not exceed $3\varepsilon'M/2$, where $\varepsilon'$ is chosen as follows:

$$\varepsilon' = \varepsilon + \frac{32}{3M} \log\left(\frac{4}{\delta}\right). \tag{16}$$

Our goal is to provide an upper bound on the probability of the complementary event $\mathcal{W}^{\text{C}}$. We start by choosing $Y_i$ as an indicator random variable such that $Y_i = 1$ if the $i^{\text{th}}$ sample is corrupted, and $Y_i = 0$ otherwise. Under the Huber contamination model, we have $\mathbb{E}[Y_i] = \varepsilon$ for all $i \in [M]$. Furthermore, the average variance satisfies $\sum_{i=1}^{M} \text{Var}(Y_i)/M \leq \varepsilon$.

Now observe:

$$\mathcal{W}^{\mathsf{C}} := \left\{ \sum_{i=1}^{M} Y_i \geq \frac{3\varepsilon' M}{2} \right\}$$

$$= \left\{ \frac{1}{M} \sum_{i=1}^{M} Y_i - \varepsilon \geq \frac{3\varepsilon'}{2} - \varepsilon \right\} \tag{17}$$

$$\implies \left\{ \frac{1}{M} \sum_{i=1}^{M} Y_i - \varepsilon \geq \frac{\varepsilon'}{2} \right\},$$

where in the last step, we used the fact that $\varepsilon' > \varepsilon$. Applying Bernstein's inequality outlined in Lemma 2 then yields the following high-probability bound on the event $\mathcal{W}^{\mathsf{C}}$:

$$\mathbb{P}\left(\mathcal{W}^{\mathsf{C}}\right) \leq 2e^{-\frac{3\varepsilon' M}{32}} \leq \frac{\delta}{2}, \tag{18}$$

where the last inequality follows from the definition of the inflated corruption fraction $\varepsilon'$ in (16).

**Step 2. Proof of Theorem 1.** To repurpose Algorithm 2 to account for the Huber contamination model, we simply invoke Algorithm 2 with an inflated corruption fraction and a deflated failure probability. Specifically, let $\hat{\mu}_X = \texttt{TRIM}[\mathcal{D}', \varepsilon, \delta] := \texttt{trimSC}[\mathcal{D}', \bar{\varepsilon}, \delta/2]$, where $\bar{\varepsilon} := \frac{3}{2}\varepsilon'$. In simple words, our modified estimation algorithm for the Huber contaminated setting, denoted by $\texttt{TRIM}$, takes as input the Huber-contaminated data set $\mathcal{D}'$, the contamination probability $\varepsilon$, and failure probability $\delta$. It then invokes Algorithm 2 with the same data set, but with an inflated corruption fraction $\bar{\varepsilon}$, and a deflated failure probability $\delta/2$. To analyze the performance of $\hat{\mu}_X$, let us define an event $\mathcal{V}$ as follows:

$$\mathcal{V} := \left\{ |\hat{\mu}_X - \mu_X| > c\sigma_X \left( \sqrt{\bar{\varepsilon}} + \sqrt{\frac{\log\left(\frac{8}{\delta}\right)}{M}} \right) \right\}, \tag{19}$$

where $c$ is the universal constant in Theorem 7. We now decompose the event $\mathcal{V}$ as $\mathcal{V} = \{\mathcal{V} \cap \mathcal{W}\} \cup \{\mathcal{V} \cap \mathcal{W}^{\mathsf{C}}\}$, which immediately implies the following:

$$\mathbb{P}(\mathcal{V}) = \mathbb{P}(\mathcal{V} \cap \mathcal{W}) + \mathbb{P}(\mathcal{V} \cap \mathcal{W}^{\mathsf{C}}) \leq \mathbb{P}(\mathcal{V} \cap \mathcal{W}) + \mathbb{P}(\mathcal{W}^{\mathsf{C}})$$
$$\leq \mathbb{P}(\mathcal{V}|\mathcal{W}) \cdot \mathbb{P}(\mathcal{W}) + \mathbb{P}(\mathcal{W}^{\mathsf{C}})$$
$$\leq \underbrace{\mathbb{P}(\mathcal{V}|\mathcal{W})}_{(*)} + \underbrace{\mathbb{P}(\mathcal{W}^{\mathsf{C}})}_{(**)}. \tag{20}$$

From (18), we already know that $(**) \leq \delta/2$. Furthermore, conditioned on the event $\mathcal{W}$, we know that there are at most $\bar{\varepsilon}M$ corrupted samples in the data set $\mathcal{D}'$. Thus, invoking Theorem 7 immediately yields that $(*) \leq \delta/2$. We conclude that with probability at least $1 - \delta$,

$$|\hat{\mu}_X - \mu_X| \leq c\sigma_X \left( \sqrt{\bar{\varepsilon}} + \sqrt{\frac{\log\left(\frac{8}{\delta}\right)}{M}} \right) \stackrel{(\bullet)}{\leq} c\sigma_X \left( \sqrt{\frac{3}{2}\varepsilon'} + \sqrt{\frac{\log\left(\frac{8}{\delta}\right)}{M}} \right)$$

$$\stackrel{(\bullet\bullet)}{\leq} \mathcal{C}\sigma_X \left( \sqrt{\varepsilon} + \sqrt{\frac{\log\left(\frac{8}{\delta}\right)}{M}} \right), \tag{21}$$

where $\mathcal{C} > c$ is some suitably large universal constant. In $(\bullet)$, we substituted the value of $\bar{\varepsilon}$, while in $(\bullet\bullet)$, we substituted $\varepsilon'$ from Eq. (16), and applied the elementary inequality $\sqrt{a+b} \leq \sqrt{a} + \sqrt{b}$, that holds for all positive scalars $a, b$. The rest follows from simple algebra. We have thus provided a proof for Theorem 1.

# C. Proof of Theorem 2

The proof of Theorem 2 follows a careful sequence of arguments that we proceed to outline next. We begin by decomposing the proposed update rule to isolate the key sources of error arising from both adversarial and non-adversarial components. This is followed by establishing $\ell_\infty$-error bounds for the non-adversarial noise in Lemmas [6,7], and for the adversarial corruption in Lemmas [8, 9]. Finally, we complete the proof of Theorem 2 by assembling these results through an inductive argument.

**Error Decomposition Step.** First, using the Bellman optimality operator in Eq. (12), the proposed robust $Q$-learning update in Eq. (6) is decomposed as follows:

$$Q_{t+1}(s_t, a_t) = (1 - \alpha)Q_t(s_t, a_t) + \alpha \mathcal{T} Q_t(s_t, a_t) + \alpha \eta_t(s_t, a_t). \tag{22}$$

Here, $\eta_t(s_t, a_t)$ is a perturbation that captures the combined effect of noise and adversarial corruption. Specifically, $\eta_t(s_t, a_t)$ is as follows:

$$\eta_t(s_t, a_t) \triangleq \gamma \max_{a' \in \mathcal{A}} Q_t(s_{t+1}, a') - \gamma \mathbb{E}_{s' \sim \mathcal{P}(.|s_t, a_t)} \left[ \max_{a' \in \mathcal{A}} Q_t(s', a') \right] + \tilde{r}_t(s_t, a_t) - R(s_t, a_t). \tag{23}$$

To aid the analysis, we further re-define the following two terms which add up to $\eta_t(s_t, a_t)$ in Eq. (23):

$$\eta_{t,1}(s_t, a_t) = \gamma \max_{a' \in \mathcal{A}} Q_t(s_{t+1}, a') - \gamma \mathbb{E}_{s' \sim \mathcal{P}(.|s_t, a_t)} \left[ \max_{a' \in \mathcal{A}} Q_t(s', a') \right],$$

$$\eta_{t,2}(s_t, a_t) = \tilde{r}_t(s_t, a_t) - R(s_t, a_t). \tag{24}$$

**Discussion on the Error Terms.** The term $\eta_t(s_t, a_t)$ defined in Equation (23) captures the deviation between the actual and ideal updates for the sampled state-action pair $(s_t, a_t)$ at the $t^{th}$ time step. Under adversarial reward corruption, this deviation naturally decomposes into two components. The first term $\eta_{t,1}(s_t, a_t)$ captures the gap between the noisy Bellman update and the true Bellman update in (12), excluding the reward term. The second term $\eta_{t,2}(s_t, a_t)$ accounts for the difference between the proposed reward proxy and the expected reward. Note that in the absence of corruption, $\tilde{r}_t(s_t, a_t) = r_t(s_t, a_t)$, such that $\mathbb{E}[r_t(s_t, a_t)] = R(s_t, a_t)$. In this case, the entire term $\eta_t(s_t, a_t)$ reduces to the difference between the noisy Bellman update and the true Bellman update.

**Final Error Decomposition and Matrix Formulation.** For aiding our analysis, we now write Eq. (22) in a compact matrix form, by introducing a time-dependent sparse, diagonal matrix $[D_t]_{|\mathcal{S}|^2 \cdot |\mathcal{A}|^2} \triangleq D_t$, whose only non-zero entry corresponds to the sampled state-action pair $(s, a) = (s_t, a_t)$ at the $t^{th}$ iteration, and equals 1. This allows us to represent the $Q$-value update for the current state-action pair using matrix notation:

$$Q_{t+1} = (I - \alpha D_t)Q_t + \alpha D_t(\mathcal{T} Q_t) + \alpha \eta_t(s_t, a_t) \mathbb{1}_t, \tag{25}$$

where $\mathbb{1}_t$ is a $|\mathcal{S}| \cdot |\mathcal{A}|$ dimensional indicator vector, which has the value 1 at the position corresponding to $(s_t, a_t)$ and 0 elsewhere. Since we are concerned with the asynchronous sampling scheme, $D_t$ is a random matrix. As a result, we introduce a new collective error term to account for this randomness, defined as follows:

$$\zeta_t \triangleq \eta_t(s_t, a_t) \mathbb{1}_t - (D_t - D)(Q_t - \mathcal{T} Q_t), \tag{26}$$

where

$$\mathbb{E}_{s_t \sim \pi, a_t \sim \mu(\cdot|s_t)}[D_t] = D, \quad \text{and} \tag{27}$$

$$D = \begin{bmatrix} \lambda(s_1, a_1) & 0 & 0 & \cdots & 0 \\ 0 & \ddots & 0 & \cdots & 0 \\ 0 & 0 & \lambda(s_i, a_i) = \pi(s_i) \cdot \mu(a_i|s_i) & \cdots & 0 \\ \vdots & \vdots & \vdots & \ddots & \vdots \\ 0 & 0 & 0 & \cdots & \lambda(s_{|\mathcal{S}|}, a_{|\mathcal{A}|}) \end{bmatrix}. \tag{28}$$

The definition of $\zeta_t$ in Eq. (26) accounts for the collective vectorized error, which includes the discrepancy described in Eq. (23) as well as the error arising from the asynchronous sampling nature of the algorithm, captured by the difference $(D_t - D)$. With the introduction of the collective error term in Eq. (26), Eq. (25) can be rewritten as follows:

$$Q_{t+1} = (I - \alpha D)Q_t + \alpha D(\mathcal{T} Q_t) + \alpha \zeta_t. \tag{29}$$

Now, $Q^*$ is the fixed point of the Bellman optimality operator $\mathcal{T}$, as defined in Equation (12), i.e., $\mathcal{T}Q^* = Q^*$. We can leverage this property to construct the error iterates $(Q_t - Q^*)$ as follows:

$$Q_{t+1} - Q^* = (I - \alpha D)(Q_t - Q^*) + \alpha D(\mathcal{T}Q_t - \mathcal{T}Q^*) + \alpha\zeta_t. \tag{30}$$

Unrolling the above recursion over $t + 1$ iterations, we get:

$$Q_{t+1} - Q^* = (I - \alpha D)^{t+1}(Q_0 - Q^*) + \alpha D \sum_{k=0}^{t} (I - \alpha D)^{t-k}(\mathcal{T}Q_k - \mathcal{T}Q^*) + \Delta_t, \tag{31}$$

where $\Delta_t$ is defined as follows:

$$\Delta_t \triangleq \alpha \sum_{k=0}^{t} (I - \alpha D)^{t-k}\zeta_k. \tag{32}$$

Notably, in the presence of adversaries, $\Delta_t$ is not a standard Martingale sequence candidate, since adversarial corruptions introduce a new bias term. To isolate the contributions of stochastic noise and adversarial perturbations, we further decompose $\Delta_t$ into two components, $\Delta_{t,1}$ and $\Delta_{t,2}$, such that:

$$\Delta_{t,1} = \alpha \sum_{k=0}^{t} (I - \alpha D)^{t-k}\zeta_{k,1}, \quad \Delta_{t,2} = \alpha \sum_{k=0}^{t} (I - \alpha D)^{t-k}\zeta_{k,2}, \quad \text{where} \tag{33}$$

the noisy $\zeta_{t,1}$ and adversarial $\zeta_{t,2}$ components which contribute to $\zeta_t$ are defined as follows:

$$\zeta_{t,1} \triangleq \eta_{t,1}(s_t, a_t)\mathbb{1}_t - (D_t - D)(Q_t - \mathcal{T}Q_t), \quad \zeta_{t,2} \triangleq \eta_{t,2}(s_t, a_t)\mathbb{1}_t. \tag{34}$$

Also, the $(s, a) - th$ component of the drift parameters in Eq. (33) is denoted as:

$$\Delta_{t,1}(s,a) \triangleq \alpha \sum_{k=0}^{t} (1 - \alpha\lambda(s,a))^{t-k}\zeta_{k,1}(s,a), \quad \Delta_{t,2}(s,a) \triangleq \alpha \sum_{k=0}^{t} (1 - \alpha\lambda(s,a))^{t-k}\zeta_{k,2}(s,a). \tag{35}$$

**Step 1: Bounding the Non-Adversarial Noisy Error $\Delta_{t,1}$.** To begin analyzing the overall error, we first consider the contribution from the cumulative non-adversarial noise term $\Delta_{t,1}$, described in Eq. (33). We first argue that $\{\zeta_{k,1}\}_{k\in[t]}$ is a standard martingale difference sequence (M.D.S). We show this by proving two key properties: uniform boundedness, established in Lemma 6, and the fact that it has a zero conditional expectation, as shown in first part of Lemma 7. In the latter part of Lemma 7, we use the standard Azuma-Hoeffding inequality from Lemma 3 to bound the cumulative error term $\Delta_{t,1}$ arising from the non-adversarial noise. We now proceed to prove the uniform boundedness property in the next result.

**Lemma 6.** *(Bounding Iterates for Robust Async-Q) The following bounds hold deterministically for all $t \in [T]$:*

$$|\eta_{t,1}(s_t, a_t)| \le \frac{6\mathcal{C}\tilde{\sigma}}{1 - \gamma}, \quad \|\zeta_{t,1}\|_\infty \le \frac{12\mathcal{C}\tilde{\sigma}}{1 - \gamma}, \tag{36}$$

*where $\mathcal{C}$ is the universal constant that appears in (5).*

*Proof.* To establish the claimed bounds, our first step is to argue that the iterates generated by Robust Async-Q remain uniformly bounded. We will prove the fact via induction. In particular, we claim that for all $s \in \mathcal{S}, a \in \mathcal{A}$, and $t \in [T]$, the following is true:

$$|Q_t(s,a)| \le \frac{3\mathcal{C}\tilde{\sigma}}{1 - \gamma}, \tag{37}$$

where $\mathcal{C}$ is the universal constant in Eq. (5). The base case of induction at $t = 0$ holds trivially since $Q_0(s,a) = 0$ for all $(s,a)$. Now suppose the bound in (37) holds up to time $t$. To show that it also applies to time $t + 1$, notice that for a state-action pair $(s,a) \ne (s_t, a_t)$, $Q_{t+1}(s,a)$ remains unchanged from time $t$ to time $t + 1$, and thus, the induction claim trivially applies to all state-action pairs that are not sampled at time $t$. Next, for the sampled state-action pair $(s_t, a_t)$ at time $t$, applying the triangle inequality to the asynchronous $Q$-learning update equation in Eq. (6) yields:

$$|Q_{t+1}(s_t, a_t)| \le (1 - \alpha)|Q_t(s_t, a_t)| + \alpha \left| \tilde{r}_t(s_t, a_t) + \gamma \max_{a'\in\mathcal{A}} Q_t(s_{t+1}, a') \right|,$$

$$\le (1 - \alpha)|Q_t(s_t, a_t)| + \alpha \left( |\tilde{r}_t(s_t, a_t)| + \gamma \max_{a'\in\mathcal{A}} |Q_t(s_{t+1}, a')| \right). \tag{38}$$

To proceed, we note from the thresholding operation in lines [6-9] of Algorithm 1 that: $|\tilde{r}_t(s_t, a_t)| \leq G_t, \forall t \geq 0$. Moreover, from the definition of $G_t$ in Eq. (5), we observe that $G_t = 0$ for all $t \leq \bar{T}$. Also, for all $t > \bar{T}$, we further have that $G_t \leq 2\mathcal{C}\tilde{\sigma} + \tilde{\sigma} \leq 3\mathcal{C}\tilde{\sigma}$, where we used the fact that $\mathcal{C} \geq 1$, and the definition of $\bar{T}$ in Eq. (4). We thus conclude that in light of the thresholding step in Algorithm 1, the following holds deterministically at all time-steps: $|\tilde{r}_t(s_t, a_t)| \leq 3\mathcal{C}\tilde{\sigma}$. Plugging this bound into Eq. (38), and using the induction hypothesis, we obtain the following for the sampled state-action pair $(s_t, a_t)$ at the $t^{th}$ instant:

$$|Q_{t+1}(s_t, a_t)| \leq (1 - \alpha) \cdot \frac{3\mathcal{C}\tilde{\sigma}}{1 - \gamma} + \alpha \left(3\mathcal{C}\tilde{\sigma} + \gamma \cdot \frac{3\mathcal{C}\tilde{\sigma}}{1 - \gamma}\right),$$
$$= \left(\frac{1 - \alpha}{1 - \gamma} + \frac{\alpha}{1 - \gamma}\right) 3\mathcal{C}\tilde{\sigma} \leq \frac{3\mathcal{C}\tilde{\sigma}}{1 - \gamma}.$$

We have thus shown that the induction claim in Eq. (37) holds for all state-action pairs $(s, a) \in \mathcal{S} \times \mathcal{A}$, and $\forall t \in [T]$. With a deterministic bound on the iterates, we now proceed to bound the non-adversarial deviation term defined in Eq. (24):

$$|\eta_{t,1}(s_t, a_t)| = \left|\gamma \max_{a' \in \mathcal{A}} Q_t(s_{t+1}, a') - \gamma \mathbb{E}_{s' \sim \mathbb{P}(\cdot|s_t, a_t)} \left[\max_{a' \in \mathcal{A}} Q_t(s', a')\right]\right|,$$
$$\leq \gamma \left|\max_{a' \in \mathcal{A}} Q_t(s_{t+1}, a')\right| + \gamma \mathbb{E}_{s' \sim \mathbb{P}(\cdot|s_t, a_t)} \left|\left[\max_{a' \in \mathcal{A}} Q_t(s', a')\right]\right|,$$
$$\leq \gamma \max_{a' \in \mathcal{A}} |Q_t(s_{t+1}, a')| + \gamma \mathbb{E}_{s' \sim \mathbb{P}(\cdot|s_t, a_t)} \left[\max_{a' \in \mathcal{A}} |Q_t(s', a')|\right],$$
$$\leq \gamma \frac{6\mathcal{C}\tilde{\sigma}}{1 - \gamma} \leq \frac{6\mathcal{C}\tilde{\sigma}}{1 - \gamma},$$

where the final inequality uses the bound in Eq. (37). Finally, consider the combined deviation term in Eq. (34). For this term, we have

$$\|\zeta_{t,1}\|_\infty \leq |\eta_{t,1}(s_t, a_t)| + \|D_t - D\|_\infty (\|Q_t\|_\infty + \|\mathcal{T}Q_t\|_\infty)$$
$$\overset{(a)}{\leq} \frac{6\mathcal{C}\tilde{\sigma}}{1 - \gamma} + (\|Q_t\|_\infty + \|\mathcal{T}Q_t\|_\infty)$$
$$\overset{(b)}{\leq} \frac{12\mathcal{C}\tilde{\sigma}}{1 - \gamma} \triangleq \bar{\Gamma}.$$

In the above steps, for (a), we used the previously established bound on $|\eta_{t,1}(s_t, a_t)|$, along with the fact that $\|D_t - D\|_\infty \leq 1$. For (b), we used (37) to deduce that $\|Q_t\|_\infty$ and $\|\mathcal{T}Q_t\|_\infty$ are both upper-bounded by $\frac{3\mathcal{C}\tilde{\sigma}}{1 - \gamma}$. In particular, the bound on $\|\mathcal{T}Q_t\|_\infty$ also uses the fact that $|R(s, a)| \leq \bar{R} \leq \tilde{\sigma}$. This completes the proof of Lemma 6, establishing deterministic uniform bounds on the non-adversarial noisy sequences $\{\eta_{t,1}\}$, and $\{\zeta_{t,1}\}$. $\square$

With the above result in hand, we now proceed to prove Lemma 7, which provides an $\ell_\infty$-norm bound on $\Delta_{t,1}$.

**Lemma 7.** *(**Bounding the Noise effect in Robust Async-Q**) With probability at least $1 - \frac{\delta}{2}$, the following bound holds simultaneously $\forall t \in [T]$:*

$$\left\|\sum_{k=0}^{t} \alpha(I - \alpha D)^{t-k} \zeta_{k,1}\right\|_\infty \leq \frac{12\mathcal{C}\tilde{\sigma}}{1 - \gamma} \sqrt{\frac{2\alpha}{\lambda_{min}} \log\left(\frac{4|\mathcal{S}||\mathcal{A}|T}{\delta}\right)}, \tag{39}$$

*where $\zeta_{k,1}$ is as defined in Eq. (34).*

*Proof.* For a fixed state-action pair $(s, a) \in \mathcal{S} \times \mathcal{A}$, we claim that the process $\{\alpha(1 - \alpha\lambda(s, a))^{t-k} \zeta_{k,1}(s, a)\}_{k \in [t]}$ is a martingale difference sequence (M.D.S) with respect to an appropriate filtration. To formally verify this property, we choose a filtration $\mathcal{F}_{k-1}$ denoted by the $\sigma$-algebra generated by the observation history up to time $k - 1$, that is, $\mathcal{F}_{k-1} := \sigma(\mathcal{O}_i : 0 \leq i \leq k - 1)$, where $\mathcal{O}_i := \{s_i, a_i, s_{i+1}, y_i(s_i, a_i)\}$. Let us also define an augmented $\sigma$-algebra $\mathcal{G}_k := \sigma(\mathcal{O}_i : 0 \leq i \leq k - 1, (s_k, a_k))$, such that $\mathcal{F}_{k-1} \subseteq \mathcal{G}_k$. In Lemma 6, we have established the uniform boundedness of $\zeta_{k,1}(s, a)$ for all $(s, a) \in \mathcal{S} \times \mathcal{A}$, and for all $k \in [t]$. To conclude that $\zeta_{k,1}(s, a)$ is indeed a M.D.S, it remains to show that $\mathbb{E}[\zeta_{k,1}(s, a)|\mathcal{F}_{k-1}] = 0$.

**Conditional Zero-Expectation Property for M.D.S.** To proceed, we start evaluating $\mathbb{E}[\zeta_{k,1}|\mathcal{F}_{k-1}]$ as follows:

$$
\begin{aligned}
\mathbb{E}[\zeta_{k,1}|\mathcal{F}_{k-1}] &= \mathbb{E}\Big[\Big(\eta_{k,1}(s_k, a_k)\mathbb{1}_k - (D_k - D)(Q_k - \mathcal{T}Q_k)\Big)\Big|\mathcal{F}_{k-1}\Big] \\
&\overset{(\bullet)}{=} \mathbb{E}\Big[\eta_{k,1}(s_k, a_k)\mathbb{1}_k\Big|\mathcal{F}_{k-1}\Big] - \mathbb{E}\Big[(D_k - D)(Q_k - \mathcal{T}Q_k)\Big|\mathcal{F}_{k-1}\Big] \\
&\overset{(\bullet\bullet)}{=} \mathbb{E}\Big[\mathbb{E}[\eta_{k,1}(s_k, a_k)\mathbb{1}_k|\mathcal{G}_k]\Big|\mathcal{F}_{k-1}\Big] = [\mathbf{0}]_{|\mathcal{S}|\times|\mathcal{A}|}.
\end{aligned}
\tag{40}
$$

In $(\bullet)$, we invoke the linearity property of conditional expectation: for integrable random variables $A$ and $B$, and a filtration $\mathcal{F}$, the following $\mathbb{E}[A + B|\mathcal{F}] = \mathbb{E}[A|\mathcal{F}] + \mathbb{E}[B|\mathcal{F}]$ holds almost surely. In $(\bullet\bullet)$, we observe that $Q_k$ is $\mathcal{F}_{k-1}$-adapted and that the sampling at time $k$ is independent of the past, under the i.i.d. sampling model. Also, $\mathbb{E}[D_k] = D$, as explained in Equation (27), it follows that $\mathbb{E}\big[(D_k - D)(Q_k - \mathcal{T}Q_k)|\mathcal{F}_{k-1}\big] = 0$. We also apply the *tower property* of conditional expectation (Lemma 5), which states that for nested $\sigma$-algebras $\mathcal{B}_1 \subseteq \mathcal{B}_2$, we have $\mathbb{E}[\mathbb{E}[X|\mathcal{B}_2]|\mathcal{B}_1] = \mathbb{E}[X|\mathcal{B}_1]$ almost surely. Using this property, we note $\mathbb{E}[\eta_{k,1}(s_k, a_k)\mathbb{1}_k|\mathcal{G}_k] = 0$. Hence, we conclude that $\mathbb{E}[\zeta_{k,1}|\mathcal{F}_{k-1}] = [\mathbf{0}]_{|\mathcal{S}|\times|\mathcal{A}|}$. Consequently, it follows that $\mathbb{E}[\zeta_{k,1}(s, a)|\mathcal{F}_{k-1}] = 0$ for all $(s, a) \in \mathcal{S} \times \mathcal{A}$. Combined with the uniform boundedness of $\zeta_{k,1}(s, a)$ established in Lemma 6, we conclude that $\{\zeta_{k,1}(s, a)\}_{k\in[t]}$ is indeed a uniformly bounded martingale difference sequence.

**Establishing the Final Bound on $\Delta_{t,1}$.** The boundedness and zero conditional expectation of the noise sequence $\{\zeta_{k,1}\}_{k\in[t]}$, as established in Lemma 6 and Eq. (40), respectively, allow us to invoke the Azuma–Hoeffding inequality described in Lemma 3 to control the deviation of the accumulated noise term. Specifically, we aim to bound $\|\Delta_{t,1}\|_\infty$ described in Eq. (33) with high probability. To achieve this, we analyze each component $\Delta_{t,1}(s, a)$ of the vector $\Delta_{t,1}$ and notice that based on Azuma-Hoeffding, for a fixed $(s, a) \in \mathcal{S} \times \mathcal{A}$ and time-step $t \in [T]$, the following high-probability concentration bound holds with probability at least $1 - \bar{\delta}_1$:

$$
\begin{aligned}
|\Delta_{t,1}(s, a)| = \Big|\sum_{k=0}^{t} \alpha(1 - \alpha\lambda(s,a))^{t-k}\zeta_{k,1}(s,a)\Big| &\overset{(a)}{\leq} \bar{\Gamma}\sqrt{2\alpha^2 \log\Big(\frac{2}{\bar{\delta}_1}\Big)\sum_{k=0}^{t}(1 - \alpha\lambda(s,a))^{2(t-k)}}, \\
&\overset{(b)}{\leq} \bar{\Gamma}\sqrt{2\alpha^2 \log\Big(\frac{2}{\bar{\delta}_1}\Big)\sum_{r=0}^{\infty}(1 - \alpha\lambda(s,a))^{r}}, \\
&\overset{(c)}{\leq} \bar{\Gamma}\sqrt{\frac{2\alpha}{\lambda_{\min}} \log\Big(\frac{2}{\bar{\delta}_1}\Big)},
\end{aligned}
\tag{41}
$$

where $\bar{\Gamma}$ is as defined in Lemma 6. We use the standard Azuma-Hoeffding inequality in $(a)$. In $(b)$, we substituted the sum of even powers by a dominating infinite sum of natural powers. In $(c)$, we have used the fact that $\lambda(s, a) \geq \lambda_{\min}$ for all $(s, a) \in \mathcal{S} \times \mathcal{A}$. Now, union bounding over all such good events for all state-action pairs $(s, a) \in \mathcal{S} \times \mathcal{A}$, and time-steps $t \in [T]$, we note that the bound derived above *holds simultaneously* for all state-action pairs and all time-steps with probability at least $1 - \bar{\delta}_1|\mathcal{S}||\mathcal{A}|T$.

Next, in order to simplify, we substitute $\bar{\delta}_1 = \delta/(2|\mathcal{S}||\mathcal{A}|T)$, and $\bar{\Gamma} = 12\mathcal{C}\tilde{\sigma}/(1 - \gamma)$. We then obtain that the following also holds for all $t \in [T]$ with probability at least $1 - \frac{\delta}{2}$:

$$
\begin{aligned}
\Big\|\sum_{k=0}^{t} \alpha(I - \alpha D)^{t-k}\zeta_{k,1}\Big\|_\infty &= \max_{(s,a)\in\mathcal{S}\times\mathcal{A}}\Big|\sum_{k=0}^{t} \alpha(1 - \alpha\lambda(s,a))^{t-k}\zeta_{k,1}(s,a)\Big| \\
&\leq \frac{12\mathcal{C}\tilde{\sigma}}{1 - \gamma}\sqrt{\frac{2\alpha}{\lambda_{\min}} \log\Big(\frac{4|\mathcal{S}||\mathcal{A}|T}{\delta}\Big)} \triangleq \bar{\Delta}_{t,1}.
\end{aligned}
\tag{42}
$$

This completes the proof. $\qquad\square$

**Step 2: Bounding the Adversarial Term $\Delta_{t,2}$.** Before discussing the bound on the adversarial noise term $\Delta_{t,2}$ under the asynchronous sampling model, we first fix some notations that will be used frequently in Lemmas 8 and 9. Denote by $\mathcal{N}_t(s, a)$ a random variable which represents the count of the number of times the state-action pair $(s, a)$ has been visited up to (and including) time $t$. Here, $\mathbb{1}_k(s, a)$ denotes the indicator variable that takes the value 1 if the state-action pair $(s_k, a_k)$ at iteration $k$ is equal to $(s, a)$, and 0 otherwise. Thus, we observe the fact that $\mathcal{N}_t(s, a) = \sum_{k\in[t]} \mathbb{1}_k(s, a)$. Under the i.i.d.

sampling model, the probability of visiting a particular $(s, a)$ pair at each time-step is given by $\lambda(s, a) = \pi(s)\mu(a|s)$. As a result, the following is true:

$$\mathbb{E}\left[\mathcal{N}_t(s, a)\right] = \lambda(s, a)t. \tag{43}$$

Building on the above fact, we now construct a "good event" of sufficient measure on which, after a burn-in time, the number of visits to each state-action pair will concentrate around its mean value. To that end, we have the following simple application of Bernstein's inequality (Lemma 2).

**Lemma 8.** *(**Constructing Good Event**) There exists an event $\mathcal{K}$ of measure at least $1 - \frac{\delta_1}{4}$, on which, the following holds simultaneously $\forall (s, a) \in \mathcal{S} \times \mathcal{A}$, $\forall t \geq \bar{T}$:*

$$\mathcal{N}_t(s, a) \geq \frac{3}{4}\lambda_{min} \cdot t,$$

*where $\bar{T} = \left\lceil \frac{104}{3\lambda_{min}} \log\left(\frac{8|\mathcal{S}||\mathcal{A}|T}{\delta_1}\right) \right\rceil$.*

*Proof.* We start by writing $\mathcal{N}_t(s, a) = \sum_{k \in [t]} \mathbb{1}_k(s, a)$, and observing the following basic facts: $\mathbb{E}[\mathbb{1}_k(s, a)] = \lambda(s, a)$, and $\text{Var}[\mathbb{1}_k(s, a))] \leq \lambda(s, a)$. For a fixed $(s, a) \in \mathcal{S} \times \mathcal{A}$ and fixed $t \in T$, the probability of the following event $\mathcal{K}_1^{\complement}(s, a, t) = \{\mathcal{N}_t(s, a) \leq \frac{3}{4}\lambda(s, a)t\}$ can be then bounded using Bernstein's inequality:

$$
\begin{aligned}
\mathbb{P}(\mathcal{K}_1^{\complement}(s, a, t)) &= \mathbb{P}\left(\left\{\mathcal{N}_t(s, a) \leq \frac{3}{4}\lambda(s, a)t\right\}\right) \\
&\leq \mathbb{P}\left(\left\{\left|\mathcal{N}_t(s, a) - \mathbb{E}\left[\mathcal{N}_t(s, a)\right]\right| \geq \frac{1}{4}\lambda(s, a)t\right\}\right) \leq 2e^{\left(-\frac{3}{104}\lambda(s,a)t\right)}.
\end{aligned}
\tag{44}
$$

Let us set $2e^{\left(-\frac{3}{104}\lambda(s,a)t\right)} \leq \hat{\delta}$. Thus, for a fixed state-action pair $(s, a) \in \mathcal{S} \times \mathcal{A}$, and a fixed $t \in T$:

$$\mathbb{P}(\mathcal{K}_1(s, a, t)) \geq 1 - \hat{\delta}, \quad \text{provided } t \geq \frac{104}{3\lambda(s, a)} \log\left(\frac{2}{\hat{\delta}}\right) \triangleq \bar{T}(s, a).$$

Union-bounding over all state-action pairs $(s, a) \in \mathcal{S} \times \mathcal{A}$ and all time-steps $t \geq \max_{(s,a) \in \mathcal{S} \times \mathcal{A}} \bar{T}(s, a)$, we conclude that there exists an event $\mathcal{K}$ of measure at least $1 - \hat{\delta}|\mathcal{S}||\mathcal{A}|T$, on which the following holds simultaneously for all state-action pairs $(s, a) \in \mathcal{S} \times \mathcal{A}$:

$$\mathcal{N}_t(s, a) \geq \frac{3}{4}\lambda(s, a)t \geq \frac{3}{4}\lambda_{\min}t,$$

provided $t \geq \bar{T}$, with $\bar{T}$ as defined in the statement of the lemma with $\hat{\delta} = \delta_1/(4|\mathcal{S}||\mathcal{A}|T)$. This concludes the proof. $\square$

**Lemma 9.** *(**Bounding Adversarial Corruption in Robust Async-Q**) With probability at least $1 - \frac{\delta}{2}$, the following bound holds simultaneously $\forall t \in [T]$:*

$$\left\|\sum_{k=0}^{t} \alpha(I - \alpha D)^{t-k} \zeta_{k,2}\right\|_{\infty} \leq \mathcal{O}\left(\alpha \mathcal{C}\tilde{\sigma}\right)\left(\sqrt{\frac{T}{\lambda_{min}} \log\left(\frac{32|\mathcal{S}||\mathcal{A}|T^2}{\delta}\right)}\right) + \frac{\mathcal{C}\bar{\sigma}}{\lambda_{min}}\sqrt{\varepsilon}, \tag{45}$$

*where $\zeta_{k,2}$ is defined in Eq. (34).*

*Proof.* We will split our analysis into two separate cases.
**Case I**: When $t \leq \bar{T}$, the term on the left-hand side of Eq. (45) deterministically simplifies to:

$$
\begin{aligned}
\left\|\sum_{k=0}^{t} \alpha(I - \alpha D)^{t-k} \zeta_{k,2}\right\|_{\infty} &\overset{(*)}{\leq} \alpha \bar{R}\bar{T} \overset{(**)}{\leq} \alpha\tilde{\sigma} \cdot \sqrt{\frac{104T}{3\lambda_{\min}} \log\left(\frac{8|\mathcal{S}||\mathcal{A}|T}{\delta_1}\right)}, \\
&\overset{(***)}{\leq} 6\alpha\mathcal{C}\tilde{\sigma} \cdot \sqrt{\frac{T}{\lambda_{\min}} \log\left(\frac{8|\mathcal{S}||\mathcal{A}|T}{\delta_1}\right)}.
\end{aligned}
\tag{46}
$$

In Eq. (46), we leveraged the threshold function described in Eq. (5) to derive the subsequent bound for the case where $k \leq \bar{T}$. It is evident that $\|I - \alpha D\|_{\infty} \leq 1$ and $\|\zeta_{k,2}\|_{\infty} \leq \bar{R} \leq \tilde{\sigma}$, since $\tilde{r}_t(s, a) = 0$ using Eq. (5) for $t \in [\bar{T}]$. Hence, the

bound in $(*)$ is satisfied. In $(**)$, we used $\bar{T} \leq \sqrt{\bar{T}}\sqrt{T}$. Finally, we substitute the value of $\bar{T}$ from Eq. (4) to arrive at the final form.[1]

**Case II**: Next, consider the case when $t > \bar{T}$. We start out by considering the following events $\mathcal{E}_k$, and $\mathcal{E}_{k,1}$ for a fixed $k \in [\bar{T}+1, T]$:

$$\mathcal{E}_k \triangleq \left\{ |\bar{r}_k(s_k, a_k) - R(s_k, a_k)| \leq C\bar{\sigma}\left(\sqrt{\frac{4}{3}\frac{\log\left(\frac{4}{\delta_1}\right)}{\lambda_{\min}k}} + \sqrt{\varepsilon}\right)\right\}. \tag{47}$$

$$\mathcal{E}_{k,1} \triangleq \left\{ |\bar{r}_k(s_k, a_k) - R(s_k, a_k)| \leq C\bar{\sigma}\left(\sqrt{\frac{\log\left(\frac{4}{\delta_1}\right)}{\mathcal{N}_k(s_k, a_k)}} + \sqrt{\varepsilon}\right)\right\}. \tag{48}$$

Next, let us borrow the good event $\mathcal{K}$ from Lemma 8, and decompose the complement of the event $\mathcal{E}_k$ described in Eq. (47) as follows:

$$\{\mathcal{E}_k^{\mathsf{C}}\} := \{\mathcal{E}_k^{\mathsf{C}}\} \cap \{\mathcal{K} \cup \mathcal{K}^{\mathsf{C}}\} = \{\mathcal{E}_k^{\mathsf{C}} \cap \mathcal{K}\} \cup \{\mathcal{E}_k^{\mathsf{C}} \cap \mathcal{K}^{\mathsf{C}}\}. \tag{49}$$

This immediately implies the following:

$$\begin{aligned}
\mathbb{P}(\mathcal{E}_k^{\mathsf{C}}) &= \mathbb{P}(\mathcal{E}_k^{\mathsf{C}} \cap \mathcal{K}) + \mathbb{P}(\mathcal{E}_k^{\mathsf{C}} \cap \mathcal{K}^{\mathsf{C}}), \\
&\leq \mathbb{P}(\mathcal{E}_k^{\mathsf{C}} \cap \mathcal{K}) + \mathbb{P}(\mathcal{K}^{\mathsf{C}}).
\end{aligned} \tag{50}$$

From Lemma 8, on the good event $\mathcal{K}$, we know that for $t \geq \bar{T}$, the following holds: $\mathcal{N}_t(s, a) \geq \frac{3}{4}\lambda_{\min}t$ for all state-action pairs $(s, a) \in \mathcal{S} \times \mathcal{A}$. Next, we establish a bound on $\mathbb{P}(\mathcal{E}_k^{\mathsf{C}} \cap \mathcal{K})$ in Eq. (50) as follows:

$$\begin{aligned}
\mathbb{P}(\mathcal{E}_k^{\mathsf{C}} \cap \mathcal{K}) &= \sum_{j=\frac{3}{4}\lambda_{\min}k}^{k} \mathbb{P}\left(\mathcal{E}_k^{\mathsf{C}} \cap \mathcal{K} \cap \{\mathcal{N}_k(s_k, a_k) = j\}\right), \\
&\leq \sum_{j=\frac{3}{4}\lambda_{\min}k}^{k} \mathbb{P}\left(\mathcal{E}_k^{\mathsf{C}} \cap \{\mathcal{N}_k(s_k, a_k) = j\}\right), \\
&\leq \sum_{j=\frac{3}{4}\lambda_{\min}k}^{k} \mathbb{P}\left(\mathcal{E}_k^{\mathsf{C}} | \{\mathcal{N}_k(s_k, a_k) = j\}\right) \cdot \mathbb{P}\left(\{\mathcal{N}_k(s_k, a_k) = j\}\right), \\
&\overset{(\bullet)}{\leq} \sum_{j=\frac{3}{4}\lambda_{\min}k}^{k} \mathbb{P}\left(\mathcal{E}_{k,1}^{\mathsf{C}} | \{\mathcal{N}_k(s_k, a_k) = j\}\right) \cdot \mathbb{P}\left(\{\mathcal{N}_k(s_k, a_k) = j\}\right), \\
&\overset{(\bullet\bullet)}{\leq} \delta_1 \cdot \sum_{j=\frac{3}{4}\lambda_{\min}k}^{k} \mathbb{P}\left(\{\mathcal{N}_k(s_k, a_k) = j\}\right), \\
&\overset{(\bullet\bullet\bullet)}{\leq} \delta_1 \cdot \sum_{j=0}^{k} \mathbb{P}\left(\{\mathcal{N}_k(s_k, a_k) = j\}\right) = \delta_1.
\end{aligned} \tag{51}$$

In $(\bullet)$, for any fixed $k \in [\bar{T}+1, T]$ and $j \in \left[\frac{3}{4}\lambda_{\min}k, k\right]$, the deviation bound specified by the event $\mathcal{E}_k$ in Eq. (47) is looser than that in $\mathcal{E}_{k,1}$ in Eq. (48) conditioned on $\mathcal{N}_k(s_k, a_k) = j$. Specifically, the following is true:

$$\{\mathcal{E}_k^{\mathsf{C}} | \mathcal{N}_k(s_k, a_k) = j\} \implies \{\mathcal{E}_{k,1}^{\mathsf{C}} | \mathcal{N}_k(s_k, a_k) = j\}. \tag{52}$$

In $(\bullet\bullet)$, by conditioning on $\mathcal{N}_k(s_k, a_k)$, we eliminate the randomness associated with asynchronous sampling. Since $j \geq \frac{3}{4}\lambda_{\min}k$, and $k \geq \bar{T} \geq T_{\lim} = \left\lceil \frac{8}{3\lambda_{\min}}\log\left(\frac{4}{\delta_1}\right)\right\rceil$ in Case II, it implies that $j \geq \frac{3}{4}\lambda_{\min}T_{\lim} \geq 2\log\left(\frac{4}{\delta_1}\right)$. Hence,

---

[1] For simplicity, we assume $\bar{T} = \frac{104}{3\lambda_{\min}}\log\left(\frac{8|\mathcal{S}||\mathcal{A}|T}{\delta_1}\right)$.

when we fix $\mathcal{N}_k(s_k, a_k) = j \in \left[\frac{3}{4}\lambda_{\min}k, k\right]$, we can leverage the robust mean guarantee in Theorem 1 as follows:

$$\mathbb{P}\left(\mathcal{E}_{k,1}^{\complement} | \{\mathcal{N}_k(s_k, a_k) = j\}\right) \leq \delta_1. \tag{53}$$

Lastly, in ($\bullet\bullet\bullet$), we used the fact that $\sum_{j=0}^{k} \mathbb{P}\left(\{\mathcal{N}_k(s_k, a_k) = j\}\right) = 1$. With Eq. (51), we can further simplify our decomposition in Eq. (50) as follows:

$$\mathbb{P}(\mathcal{E}_k^{\complement}) = \mathbb{P}(\mathcal{E}^{\complement} \cap \mathcal{K}) + \mathbb{P}(\mathcal{K}^{\complement}),$$
$$\overset{(*)}{\leq} \delta_1 + \frac{\delta_1}{4} \leq 2\delta_1. \tag{54}$$

In step ($*$), we applied the upper bound on the probability of the good event $\mathcal{K}$ established in Lemma 8. Combining these results, we conclude that the following holds for a fixed $k \in [\bar{T} + 1, T]$:

$$\mathbb{P}(\mathcal{E}_k) \geq 1 - 2\delta_1. \tag{55}$$

Union-bounding over all time-steps $k \in [\bar{T} + 1, T]$, we conclude that there exists an event $\mathcal{J}$ of measure at least $1 - 2\delta_1 T$, on which, the following holds simultaneously for all time steps $k \in [\bar{T} + 1, T]$:

$$|\bar{r}_k(s_k, a_k) - R(s_k, a_k)| \leq \mathcal{C}\bar{\sigma}\left(\sqrt{\frac{4}{3} \cdot \frac{\log\left(\frac{4}{\delta_1}\right)}{\lambda_{\min}k}} + \sqrt{\varepsilon}\right). \tag{56}$$

Now notice that on the good event $\mathcal{J}$ defined as above, when $k > \bar{T}$, the following is true:

$$|\bar{r}_k(s_k, a_k)| \leq \mathcal{C}\bar{\sigma}\left(\sqrt{\frac{4}{3} \cdot \frac{\log\left(\frac{4}{\delta_1}\right)}{\lambda_{\min}k}} + \sqrt{\varepsilon}\right) + |R(s_k, a_k)| \leq G_k, \tag{57}$$

where we used $|R(s_k, a_k)| \leq \bar{R} \leq \tilde{\sigma}$, and the definition of the threshold $G_k$ from (5). We conclude that on event $\mathcal{J}$, the thresholding step in line 7 of Algorithm 1 will get bypassed, ensuring that $\tilde{r}_k(s_k, a_k) = \bar{r}_k(s_k, a_k), \forall k > \bar{T}$. Crucially, based on (56), this implies that on the event $\mathcal{J}$, the following deviation bound on the reward proxy applies simultaneously for all time steps $k \in [\bar{T} + 1, T]$:

$$|\tilde{r}_k(s_k, a_k) - R(s_k, a_k)| \leq \mathcal{C}\bar{\sigma}\left(\sqrt{\frac{4}{3} \cdot \frac{\log\left(\frac{4}{\delta_1}\right)}{\lambda_{\min}k}} + \sqrt{\varepsilon}\right). \tag{58}$$

Now, we substitute $\delta_1 = \delta/4T$, ensuring that the event $\mathcal{J}$ takes place with probability at least $1 - \frac{\delta}{2}$. Before moving forward, we pause to note that the aforementioned arguments have already established Lemma 1 in the main text.

In the remainder of the proof, we will condition on the good event $\mathcal{J}$ on which (58) holds. On this event, it is easy to see that for $k > \bar{T}$,

$$\|\zeta_{k,2}\|_\infty = \left\| [\tilde{r}_k(s_k, a_k) - R(s_k, a_k)] \mathbb{1}_k \right\|_\infty$$
$$= |\tilde{r}_k(s_k, a_k) - R(s_k, a_k)| \leq \mathcal{C}\bar{\sigma}\left(\sqrt{\frac{4}{3} \cdot \frac{\log\left(\frac{4}{\delta_1}\right)}{\lambda_{\min}k}} + \sqrt{\varepsilon}\right). \tag{59}$$

Invoking Eq. (59), the following then holds on event $\mathcal{J}$:

$$\left\| \sum_{k=\bar{T}+1}^{t} \alpha(I - \alpha D)^{t-k} \zeta_{k,2} \right\|_{\infty} \leq \sum_{k=\bar{T}+1}^{t} \alpha \|(I - \alpha D)\|_{\infty}^{t-k} \cdot \|\zeta_{k,2}\|_{\infty}$$

$$\overset{(*)}{\leq} \alpha \mathcal{C} \bar{\sigma} \sum_{k=\bar{T}+1}^{t} (1 - \alpha \lambda_{\min})^{t-k} \left( \sqrt{\frac{4}{3} \cdot \frac{\log\left(\frac{4}{\delta_1}\right)}{\lambda_{\min} k}} + \sqrt{\varepsilon} \right)$$

$$\leq \alpha \mathcal{C} \bar{\sigma} \left( \sqrt{\frac{4}{3} \frac{\log\left(\frac{4}{\delta_1}\right)}{\lambda_{\min}}} \right) \sum_{k=\bar{T}+1}^{t} \left( \frac{1}{\sqrt{k}} \right) + \sum_{k=\bar{T}+1}^{t} \alpha(1 - \alpha \lambda_{\min})^{t-k} \mathcal{C} \bar{\sigma} \sqrt{\varepsilon} \qquad (60)$$

$$\overset{(**)}{\leq} \alpha \mathcal{C} \bar{\sigma} \left( \sqrt{\frac{4}{3} \frac{\log\left(\frac{4}{\delta_1}\right)}{\lambda_{\min}}} \right) \int_{k=\bar{T}+1}^{t} \left( \frac{1}{\sqrt{k}} \right) + \frac{\mathcal{C} \bar{\sigma}}{\lambda_{\min}} \sqrt{\varepsilon}$$

$$\overset{(***)}{\leq} \mathcal{O}(\alpha \mathcal{C} \bar{\sigma}) \left( \sqrt{\frac{4}{3} \frac{T}{\lambda_{\min}} \log\left(\frac{4}{\delta_1}\right)} \right) + \frac{\mathcal{C} \bar{\sigma}}{\lambda_{\min}} \sqrt{\varepsilon}.$$

Using the bound $\|I - \alpha D\|_{\infty} \leq (1 - \alpha \lambda_{\min})$ and the deviation bound on $\zeta_{k,2}$ from event $\mathcal{J}$, we obtain step $(*)$. The resulting summation is then separated into two terms—one involving $\frac{1}{\sqrt{k}}$ and another involving a constant $\sqrt{\varepsilon}$. The first term is further upper bounded via an integral approximation $(**)$, while the second term is bounded using the geometric sum of the decaying factor $(1 - \alpha \lambda_{\min})^{t-k}$, which sums to at most $1/(\alpha \lambda_{\min})$. Finally, evaluating the integral and using the upper bound $T$ on the total number of iterations yields the bound in step $(***)$.

Next, to obtain the final bound for Case II, we leverage the bound from Case I to obtain the following (on event $\mathcal{J}$) for all $t > \bar{T}$:

$$\left\| \sum_{k=0}^{t} \alpha(I - \alpha D)^{t-k} \zeta_{k,2} \right\|_{\infty} \leq \left\| \sum_{k=0}^{\bar{T}} \alpha(I - \alpha D)^{t-k} \zeta_{k,2} \right\|_{\infty} + \left\| \sum_{k=\bar{T}+1}^{t} \alpha(I - \alpha D)^{t-k} \zeta_{k,2} \right\|_{\infty}$$

$$\overset{(\dagger)}{\leq} \mathcal{O}(\alpha \mathcal{C} \tilde{\sigma}) \cdot \sqrt{\frac{T}{\lambda_{\min}} \log\left(\frac{8|\mathcal{S}||\mathcal{A}|T}{\delta_1}\right)} + \mathcal{O}(\alpha \mathcal{C} \tilde{\sigma}) \left( \sqrt{\frac{4}{3} \frac{T}{\lambda_{\min}} \log\left(\frac{16T}{\delta}\right)} \right) + \frac{\mathcal{C} \bar{\sigma}}{\lambda_{\min}} \sqrt{\varepsilon} \qquad (61)$$

$$\overset{(\dagger\dagger)}{\leq} \mathcal{O}(\alpha \mathcal{C} \tilde{\sigma}) \left( \sqrt{\frac{T}{\lambda_{\min}} \log\left(\frac{32|\mathcal{S}||\mathcal{A}|T^2}{\delta}\right)} \right) + \frac{\mathcal{C} \bar{\sigma}}{\lambda_{\min}} \sqrt{\varepsilon} \triangleq \bar{\Delta}_{t,2}.$$

In $(\dagger)$, we used the bounds obtained in Case I and Case II, and then used the fact that $\bar{\sigma} \leq \tilde{\sigma}$. In $(\dagger\dagger)$, we simply used the monotonicity of logarithms and substituted $\delta_1 = \delta/4T$. Lastly, combining our separate analyses for Case I and Case II leads to the claim of the lemma. $\qquad \square$

**Finite-Time Rates for Robust Async-Q (Proof of Theorem 2)**: Having established Lemmas 6, 7, 8, and 9, we are now ready to proceed with the proof of the bound stated in Theorem 2. First, to build intuition for the nature of the final bound, let us consider Eq. (29) in the absence of any contributions from noise or adversaries. In this case, the recursion simplifies to the idealized update rule: $Q_{t+1} = (I - \alpha D)Q_t + \alpha D(\mathcal{T} Q_t)$. Subtracting the fixed point $Q^*$, which satisfies $Q^* = \mathcal{T} Q^*$, we obtain the error recursion $Q_{t+1} - Q^* = (I - \alpha D)(Q_t - Q^*) + \alpha D(\mathcal{T} Q_t - \mathcal{T} Q^*)$. Defining $d_t(s, a) := |Q_t(s, a) - Q^*(s, a)|$, and applying the contractiveness of the Bellman optimality operator under the $\infty$-norm, we can then obtain the following for each state-action pair $(s, a) \in \mathcal{S} \times \mathcal{A}$:

$$\begin{aligned} d_{t+1}(s, a) &\leq (1 - \alpha \lambda(s, a)) d_t(s, a) + \alpha \gamma \lambda(s, a) \|d_t\|_{\infty}, \\ &\leq (1 - \alpha \lambda_{\min}(1 - \gamma)) \|d_t\|_{\infty}. \end{aligned} \qquad (62)$$

Since this upper bound holds uniformly over all $(s, a) \in \mathcal{S} \times \mathcal{A}$, we conclude:

$$\|d_{t+1}\|_{\infty} \leq (1 - \alpha \lambda_{\min}(1 - \gamma)) \|d_t\|_{\infty}. \qquad (63)$$

Unrolling this recursion yields the following for all $t \in [T]$:

$$\|d_t\|_\infty \le (1 - \alpha\lambda_{\min}(1 - \gamma))^t \|d_0\|_\infty. \tag{64}$$

The goal is to now establish a similar recursion for our setting, while accounting for noise and adversarial corruption. To do so, we note that based on Lemma 7 and Lemma 9, there exists an event - say $\mathcal{Y}$ - of measure at least $1 - \delta$, on which, $\|\Delta_{t,1}\|_\infty + \|\Delta_{t,2}\|_\infty \le \bar{\Delta}_{t,1} + \bar{\Delta}_{t,2} \triangleq \Delta, \forall t \in [T]$, where $\Delta_{t,1}$ and $\Delta_{t,2}$ are as defined in Eq. (33), $\bar{\Delta}_{t,1}$ is as defined in Eq. (42), and $\bar{\Delta}_{t,2}$ is as defined in Eq. (61). As our induction hypothesis, suppose that on the event $\mathcal{Y}$, the following bound holds for all $t \in [T]$:

$$\|d_t\|_\infty \le (1 - \alpha\lambda_{\min}(1 - \gamma))^t \|d_0\|_\infty + \frac{\Delta}{1 - \gamma}. \tag{65}$$

For $t = 0$, it is trivially true. Suppose the above bound holds for all time-steps up to time-step $t$. To show that it also applies to time-step $t + 1$, let us revisit Eq. (31) and analyze it component-wise. In order to simplify the notation for algebraic decompositions in the subsequent steps, for two given functions $\{Q_1, Q_2\}$ and their corresponding mappings $\{\mathcal{T}Q_1, \mathcal{T}Q_2\}$ under the influence of the Bellman operator, we denote their component-wise difference as:

$$\begin{aligned}
[Q_1 - Q_2](s, a) &\triangleq Q_1(s, a) - Q_2(s, a) \\
[\mathcal{T}Q_1 - \mathcal{T}Q_2](s, a) &\triangleq \mathcal{T}Q_1(s, a) - \mathcal{T}Q_2(s, a).
\end{aligned} \tag{66}$$

Similarly, we denote the $(s, a)$-th component of $\Delta_t$ defined in Eq. (33), as $\Delta_t(s, a)$. Now, we proceed component wise, where the $(s, a)$-th component of Eq. (31) gives us the following:

$$\begin{aligned}
[Q_{t+1} - Q^*](s, a) &= (1 - \alpha\lambda(s, a))^{t+1}[Q_0 - Q^*](s, a) \\
&+ \alpha\lambda(s, a) \sum_{k=0}^{t} (1 - \alpha\lambda(s, a))^{t-k}[\mathcal{T}Q_k - \mathcal{T}Q^*](s, a) + \Delta_t(s, a).
\end{aligned} \tag{67}$$

Taking absolute values on both sides of Eq. (67), and substituting $d_t(s, a) = |[Q_t - Q^*](s, a)|$, we get the following form:

$$d_{t+1}(s, a) \le (1 - \alpha\lambda(s, a))^{t+1}d_0(s, a) + \alpha\gamma\lambda(s, a) \sum_{k=0}^{t} (1 - \alpha\lambda(s, a))^{t-k}\|d_k\|_\infty + |\Delta_t(s, a)|. \tag{68}$$

Now, substituting $|\Delta_t(s, a)| \le |\Delta_{t,1}(s, a)| + |\Delta_{t,2}(s, a)| \le \|\Delta_{t,1}\|_\infty + \|\Delta_{t,2}\|_\infty \le \bar{\Delta}_{t,1} + \bar{\Delta}_{t,2} = \Delta$ and the claim from Eq. (65) into Eq. (68), we get:

$$\begin{aligned}
d_{t+1}(s, a) &\le \underbrace{(1 - \alpha\lambda(s, a))^{t+1}d_0(s, a) + \alpha\gamma\lambda(s, a) \sum_{k=0}^{t} (1 - \alpha\lambda(s, a))^{t-k} (1 - \alpha\lambda_{\min}(1 - \gamma))^k \|d_0\|_\infty}_{(\bullet)} \\
&+ \underbrace{\alpha\gamma\lambda(s, a) \sum_{k=0}^{t} (1 - \alpha\lambda(s, a))^{t-k} \frac{\Delta}{1 - \gamma} + \Delta}_{(\bullet\bullet)},
\end{aligned} \tag{69}$$

$$\overset{(a)}{\le} (1 - \alpha\lambda_{\min}(1 - \gamma))^{t+1}\|d_0\|_\infty + + \alpha\gamma\lambda(s, a) \sum_{r=0}^{\infty} (1 - \alpha\lambda(s, a))^r \frac{\Delta}{1 - \gamma} + \Delta,$$

$$\le (1 - \alpha\lambda_{\min}(1 - \gamma))^{t+1}\|d_0\|_\infty + \frac{\Delta}{1 - \gamma}.$$

In $(a)$, for bounding $(\bullet)$, we used the following argument:

$$\begin{aligned}
(\bullet) &\le \left[ (1 - \alpha\lambda(s, a))^{t+1} + \alpha\gamma\lambda(s, a)(1 - \alpha\lambda(s, a))^t \sum_{k=0}^{t} \left( \frac{1 - \alpha\lambda_{min}(1 - \gamma)}{1 - \alpha\lambda(s, a)} \right)^k \right] \|d_0\|_\infty, \\
&= \left[ (1 - \alpha\lambda(s, a))^{t+1} + \alpha\gamma\lambda(s, a) \frac{(1 - \alpha(1 - \gamma)\lambda_{\min})^{t+1} - (1 - \alpha\lambda(s, a))^{t+1}}{\alpha(\lambda(s, a) - (1 - \gamma)\lambda_{\min})} \right] \|d_0\|_\infty, \\
&\le \left[ (1 - \alpha\lambda(s, a))^{t+1} + \alpha\gamma\lambda(s, a) \frac{(1 - \alpha(1 - \gamma)\lambda_{\min})^{t+1} - (1 - \alpha\lambda(s, a))^{t+1}}{\alpha(\lambda(s, a) - (1 - \gamma)\lambda(s, a))} \right] \|d_0\|_\infty, \\
&\le (1 - \alpha\lambda_{\min}(1 - \gamma))^{t+1} \|d_0\|_\infty.
\end{aligned} \tag{70}$$

For ($\bullet\bullet$), we have upper bounded the finite-sum by an infinite-sum as follows:

$$
\begin{aligned}
(\bullet\bullet) &= \alpha\gamma\lambda(s,a) \sum_{k=0}^{t} (1 - \alpha\lambda(s,a))^{t-k} \frac{\Delta}{1-\gamma} + \Delta, \\
&\leq \alpha\gamma\lambda(s,a) \sum_{r=0}^{\infty} (1 - \alpha\lambda(s,a))^r \frac{\Delta}{1-\gamma} + \Delta \leq \frac{\Delta}{1-\gamma}.
\end{aligned}
\tag{71}
$$

This settles our claim made in Eq. (65). As a result, we conclude that the following holds on event $\mathcal{Y}$:

$$
\begin{aligned}
\|d_T\|_\infty &\leq (1 - \alpha\lambda_{\min}(1-\gamma))^T \|d_0\|_\infty + \frac{\Delta}{1-\gamma}, \\
&\leq e^{-\alpha\lambda_{\min}(1-\gamma)T} \|d_0\|_\infty + \frac{\Delta}{1-\gamma}.
\end{aligned}
\tag{72}
$$

Substituting $\alpha = \dfrac{\log T}{\lambda_{\min} T (1-\gamma)}$ in the above display, simplifying, and using the fact that $\mathcal{Y}$ has measure at least $1 - \delta$, we conclude that the following holds with probability $1 - \delta$:

$$
\|d_T\|_\infty \leq \frac{\|d_0\|_\infty}{T} + \mathcal{O}\left( \frac{\tilde{\sigma}}{(1-\gamma)^{\frac{5}{2}} \lambda_{\min}^{\frac{3}{2}} \sqrt{T}} \sqrt{\log\left( \frac{32|\mathcal{S}||\mathcal{A}|T^2}{\delta} \right)} + \frac{\bar{\sigma}\sqrt{\varepsilon}}{\lambda_{\min}(1-\gamma)} \right).
\tag{73}
$$

This completes our proof.

# D. Proof of Theorem 3

In this section, we prove the lower bound stated in Theorem 3. The proof is based on constructing two carefully designed observation models under a simple synchronous Huber contamination setting outlined in (Kearns & Singh, 1998; Even-Dar et al., 2003; Sidford et al., 2018), where at each round the learner receives corrupted or clean rewards for all state-action pairs simultaneously. We begin by outlining the core intuition before delving into the technical details. We carefully construct two MDPs that satisfy two crucial properties: (i) the optimal state-action value functions corresponding to the constructed MDPs differ by $\Omega(\sqrt{\varepsilon})$, and (ii) under the Huber contamination model, the observed reward distributions are identical across the two MDPs. This setup ensures that no estimator can reliably distinguish between the two MDPs based on the contaminated observations alone, thereby forcing any estimator to incur an error of at least $\Omega(\sqrt{\varepsilon})$ in the worst case. We now proceed to formalize the argument.

● `Step 1` **(MDP Construction)** To construct the lower bound instance, we consider two MDPs that have a single common state $s$ and a single common action $a$, such that the only source of randomness arises from the observed reward for the state-action pair $(s, a)$. Slightly departing from the notation introduced earlier in the prelude to Theorem 3, we use indices $i = 1$ and $i = 2$ to represent objects associated with MDP 1 and MDP 2, respectively. The true noisy reward distributions $\mathcal{R}_1(s, a)$ and $\mathcal{R}_2(s, a)$ associated with MDPs 1 and 2 are as follows:

$$\mathcal{R}_1(s, a) = \begin{cases} \frac{\bar{\sigma}}{\sqrt{\varepsilon}} & \text{with prob. } \frac{\varepsilon}{4(1-\varepsilon)}, \\ 0 & \text{with prob. } 1 - \frac{\varepsilon}{4(1-\varepsilon)} \end{cases}, \mathcal{R}_2(s, a) = \begin{cases} -\frac{\bar{\sigma}}{\sqrt{\varepsilon}} & \text{with prob. } \frac{\varepsilon}{4(1-\varepsilon)}, \\ 0 & \text{with prob. } 1 - \frac{\varepsilon}{4(1-\varepsilon)} \end{cases} \tag{74}$$

where $\bar{\sigma} > 0$ is a fixed constant. Let the expected rewards under distributions $\mathcal{R}_1(s, a)$ and $\mathcal{R}_2(s, a)$ be denoted by $R_1$ and $R_2$, respectively. It is straightforward to check that:

$$R_1 = \frac{\bar{\sigma}\sqrt{\varepsilon}}{4(1-\varepsilon)}, \qquad R_2 = -\frac{\bar{\sigma}\sqrt{\varepsilon}}{4(1-\varepsilon)}. \tag{75}$$

Additionally, if $r_1(s, a) \sim \mathcal{R}_1(s, a)$ and $r_2(s, a) \sim \mathcal{R}_2(s, a)$, then the variances of these random variables are as follows:

$$\text{Var}(r_1(s, a)) = \text{Var}(r_2(s, a)) \le \frac{\bar{\sigma}^2}{\varepsilon} \cdot \frac{\varepsilon}{4(1-\varepsilon)} = \frac{\bar{\sigma}^2}{4(1-\varepsilon)} < 0.5\bar{\sigma}^2, \tag{76}$$

where we have used the assumption that $\varepsilon < 0.5$. Thus, each reward model has a finite variance uniformly bounded above by $\bar{\sigma}^2$. Since there is only one state-action pair, the optimal $Q$-value in each MDP is given by:

$$Q_i^*(s, a) = \frac{R_i}{1 - \gamma}, \quad i \in \{1, 2\}. \tag{77}$$

● `Step 2` **(Construction of Corrupted Observation Models)** We now construct adversarial reward contaminations under the Huber contamination model. For each MDP $i \in \{1, 2\}$, the observed reward at $(s, a)$ is drawn from the true distribution $\mathcal{R}_i(s, a)$ with probability $1 - \varepsilon$, and from an adversarial distribution $\mathcal{Q}_i$ with probability $\varepsilon$. Here, $\mathcal{Q}_i$ is the corruption distribution for MDP $i \in \{1, 2\}$, defined as follows:

$$\mathcal{Q}_1 = \begin{cases} -\frac{\bar{\sigma}}{\sqrt{\varepsilon}} & \text{with probability } 0.5 \\ 0 & \text{with probability } 0.25 \\ \frac{\bar{\sigma}}{\sqrt{\varepsilon}} & \text{with probability } 0.25. \end{cases}, \quad \mathcal{Q}_2 = \begin{cases} -\frac{\bar{\sigma}}{\sqrt{\varepsilon}} & \text{with probability } 0.25 \\ 0 & \text{with probability } 0.25 \\ \frac{\bar{\sigma}}{\sqrt{\varepsilon}} & \text{with probability } 0.5. \end{cases} \tag{78}$$

The resulting Huber-contaminated reward distributions are $\tilde{R}_i = (1 - \varepsilon)\mathcal{R}_i(s, a) + \varepsilon\mathcal{Q}_i$, $i \in \{1, 2\}$. Crucially, based on our construction above, $\tilde{\mathcal{R}}_1 = \tilde{\mathcal{R}}_2 = -\bar{\sigma}/\sqrt{\varepsilon}$ with probability $\varepsilon/2$; $0$ with probability $1 - \varepsilon$; and $\bar{\sigma}/\sqrt{\varepsilon}$ with probability $\varepsilon/2$. As a result, *a learner cannot distinguish between the corrupted reward distributions of the two MDPs.* However, as established in `Step 1`, the true (uncorrupted) expected rewards under these MDPs differ. Thus, the corresponding true optimal $Q^*$-values also differ, with the following bound: $|Q_1^* - Q_2^*| = |R_1 - R_2|/(1 - \gamma) \ge 0.5\bar{\sigma}\sqrt{\varepsilon}/(1 - \gamma)$, where we will henceforth use the simpler notation $Q_i^*(s, a) \triangleq Q_i^*$ for $i \in \{1, 2\}$ since there is only one state-action pair. We now proceed to establish that *any* estimator of the optimal state-action value function must suffer an error of $\Omega\left(\bar{\sigma}\sqrt{\varepsilon}/(1 - \gamma)\right)$ on at least one of the two MDPs.

● `Step 3` **(Proof of the Lower Bound)** We construct two statistically indistinguishable instances. In `Instance 1`, the learner observes $T$ i.i.d. samples $X := \{X_i\}_{i \in [T]}$ drawn from the distribution $\tilde{\mathcal{R}}_1$. In `Instance 2`, the learner instead

observes $T$ i.i.d. samples $\mathbb{Y} := \{Y_i\}_{i \in [T]}$ drawn from the distribution $\tilde{\mathcal{R}}_2$. We denote $\tilde{\mathcal{R}}_i^{\otimes}$ as the $T$-fold product measure of $\tilde{\mathcal{R}}_i$, i.e., the joint law of $T$ i.i.d. samples from $\tilde{\mathcal{R}}_i$ for $i \in \{1, 2\}$. Now, suppose $\hat{R}_T$ and $\hat{Q}_T$ are estimators for the mean rewards and optimal state-action value functions, respectively, in the two MDPs. We will show that a lower bound on the performance of $\hat{R}_T$ directly implies a corresponding lower bound on the performance of $\hat{Q}_T$. To begin, observe that

$$2 \max \left\{ \mathbb{P}_{\tilde{\mathcal{R}}_1^{\otimes}} \left( |\hat{R}_T - R_1| > \frac{\bar{\sigma}\sqrt{\varepsilon}}{8(1-\varepsilon)} \right), \mathbb{P}_{\tilde{\mathcal{R}}_2^{\otimes}} \left( |\hat{R}_T - R_2| > \frac{\bar{\sigma}\sqrt{\varepsilon}}{8(1-\varepsilon)} \right) \right\}$$

$$\geq \mathbb{P}_{\tilde{\mathcal{R}}_1^{\otimes}} \underbrace{\left( |\hat{R}_T - R_1| > \frac{\bar{\sigma}\sqrt{\varepsilon}}{8(1-\varepsilon)} \right)}_{\mathcal{A}_1} + \mathbb{P}_{\tilde{\mathcal{R}}_2^{\otimes}} \underbrace{\left( |\hat{R}_T - R_2| > \frac{\bar{\sigma}\sqrt{\varepsilon}}{8(1-\varepsilon)} \right)}_{\mathcal{A}_2}$$

$$\overset{(\bullet)}{\geq} \mathbb{P}_{\tilde{\mathcal{R}}_1^{\otimes}}(\mathcal{A}_1) + \mathbb{P}_{\tilde{\mathcal{R}}_2^{\otimes}}(\mathcal{A}_1^c) \overset{(\bullet\bullet)}{\geq} (1/2) \cdot \exp\left( -\mathrm{KL}(\tilde{\mathcal{R}}_1^{\otimes} \| \tilde{\mathcal{R}}_2^{\otimes}) \right) = \frac{1}{2},$$

where we use $\mathrm{KL}(P \| Q)$ to denote the KL-divergence between two distributions $P$ and $Q$. From the expressions for $R_1$ and $R_2$, it follows that $\mathcal{A}_1^c \implies \mathcal{A}_2$, implying $\mathbb{P}_{\tilde{\mathcal{R}}_2^{\otimes}}(\mathcal{A}_1^c) \leq \mathbb{P}_{\tilde{\mathcal{R}}_2^{\otimes}}(\mathcal{A}_2)$; this explains $(\bullet)$. For $(\bullet\bullet)$, we first apply the Bretagnolle–Huber inequality as in Lemma 5, and then use the fact that $\mathrm{KL}(\tilde{\mathcal{R}}_1^{\otimes} \| \tilde{\mathcal{R}}_2^{\otimes}) = T \cdot \mathrm{KL}(\tilde{\mathcal{R}}_1 \| \tilde{\mathcal{R}}_2) = 0$, since $\tilde{\mathcal{R}}_1 = \tilde{\mathcal{R}}_2$ by construction. Using $1/(1-\varepsilon) > 1$, we then conclude that:

$$\max \left\{ \mathbb{P}_{\tilde{\mathcal{R}}_1^{\otimes}} \left( \left| \hat{R}_T - R_1 \right| > \frac{\bar{\sigma}\sqrt{\varepsilon}}{8} \right), \mathbb{P}_{\tilde{\mathcal{R}}_2^{\otimes}} \left( \left| \hat{R}_T - R_2 \right| > \frac{\bar{\sigma}\sqrt{\varepsilon}}{8} \right) \right\} \geq \frac{1}{4}. \tag{79}$$

In light of (79), we claim the following:

$$\max \left\{ \mathbb{P}_{\tilde{\mathcal{R}}_1^{\otimes}} \left( |\hat{Q}_T - Q_1^*| > \frac{\bar{\sigma}\sqrt{\varepsilon}}{8(1-\gamma)} \right), \mathbb{P}_{\tilde{\mathcal{R}}_2^{\otimes}} \left( |\hat{Q}_T - Q_2^*| > \frac{\bar{\sigma}\sqrt{\varepsilon}}{8(1-\gamma)} \right) \right\} \geq \frac{1}{4}. \tag{80}$$

The claim essentially follows from the simple observation that if an optimal state-action value-function estimator $\hat{Q}_T$ can accurately estimate both $Q_1^*$ and $Q_2^*$, then one can use such an estimator to construct accurate estimates of both $R_1$ and $R_2$, thereby violating Eq. (79). Formally, to see that Eq. (79) implies Eq. (80), suppose there exists an estimator $\hat{Q}_T$ such that

$$\max \left\{ \mathbb{P}_{\tilde{\mathcal{R}}_1^{\otimes}} \left( |\hat{Q}_T - Q_1^*| > \frac{\bar{\sigma}\sqrt{\varepsilon}}{8(1-\gamma)} \right), \ \mathbb{P}_{\tilde{\mathcal{R}}_2^{\otimes}} \left( |\hat{Q}_T - Q_2^*| > \frac{\bar{\sigma}\sqrt{\varepsilon}}{8(1-\gamma)} \right) \right\} < \frac{1}{4}. \tag{81}$$

Using $\hat{Q}_T$, construct a reward estimator $\hat{R}_T = (1-\gamma)\hat{Q}_T$. From Eq. (77), we then immediately have:

$$\max \left\{ \mathbb{P}_{\tilde{\mathcal{R}}_1^{\otimes}} \left( \left| \hat{R}_T - R_1 \right| > \frac{\bar{\sigma}\sqrt{\varepsilon}}{8} \right), \ \mathbb{P}_{\tilde{\mathcal{R}}_2^{\otimes}} \left( \left| \hat{R}_T - R_2 \right| > \frac{\bar{\sigma}\sqrt{\varepsilon}}{8} \right) \right\} < \frac{1}{4}. \tag{82}$$

This completes the claim and the proof.

# E. Proof of Theorem 4

The finite-time performance of `Robust Async-RAQ` is established in Theorem 4. The first step in the proof of this result is an error-decomposition that mirrors Eq. (31) in Section C. The structure of the rest of the proof is similar to that of Theorem 2 in Appendix C. However, there will be some departures that arise from the use of a reward-agnostic threshold function in Eq. (8). We will highlight these points of departure in our subsequent analysis.

`Step 1`: **Bound on the Adversarial Term** $\Delta_{t,2}$. We begin by analyzing the contribution of the adversarial corruption term, before turning to the non-adversarial noisy component. The latter necessitates a more refined and intricate analysis, as will become evident in the sequel.

**Lemma 10.** *(Bounding Adversarial Corruption in Robust Async-RAQ) Suppose $\delta_1 \leq \delta/4T$. Then, with probability at least $1 - \delta/2$, the following bound holds simultaneously for all $t \in [T]$:*

$$\left\| \sum_{k=0}^{t} \alpha (I - \alpha D)^{t-k} \zeta_{k,2} \right\|_{\infty} \leq \mathcal{O}(\alpha\tilde{\sigma}) \left( \tilde{\sigma}^{1/2p} \sqrt{T} + \sqrt{\frac{T}{\lambda_{\min}} \log\left(\frac{|\mathcal{S}||\mathcal{A}|T}{\delta_1}\right)} \right) + \mathcal{O}\left(\frac{\bar{\sigma}\sqrt{\varepsilon}}{\lambda_{\min}}\right),$$

*where $\zeta_{k,2}$ is defined in Eq. (34).*

*Proof.* Like in our proof of Lemma 9, we divide the analysis into two cases based on the value of $t$. Since the threshold function defined in Eq. (8) is agnostic to the underlying reward statistics, we introduce an auxiliary time-step $\tilde{T} := \max\{\tilde{\sigma}^{1/p}, \bar{T}\}$, where $\bar{T}$ was previously defined in Eq. (4), and recall that $p$ is the parameter in the function $m(t) = t^p$ that appears in the modified threshold (8).

**Case I:** Consider first the case where $t \leq \tilde{T}$. We further split up this case into two sub-cases: one where $\tilde{T} = \bar{T}$, and the other where $\tilde{T} = \tilde{\sigma}^{1/p}$. We separately analyze these sub-cases below.

- Suppose $\tilde{T} = \bar{T}$, which implies $t \leq \bar{T}$. Then, by the definition of the threshold function in Eq. (8), we have $\tilde{r}_t(s_t, a_t) = 0$. Consequently, just like in Case 1 of Lemma 9, in this case we have $\|\zeta_{t,2}\|_{\infty} \leq \tilde{\sigma}$.

- Next, when $\tilde{T} = \tilde{\sigma}^{1/p}$, and $t \in [\bar{T}, \tilde{T}]$, we can use the reward-agnostic threshold function defined in Eq. (8) to bound $\|\zeta_{t,2}\|_{\infty}$. To see how, start by noting that the following is always true deterministically: $|\tilde{r}_t(s_t, a_t)| \leq \tilde{G}_t, \forall t \geq 0$. Using $m(t) = t^p$ in Eq. (8), and the fact that $t \geq \bar{T}$, we note that for $t \in [\bar{T}, \tilde{T}]$, the following is true: $\tilde{G}_t \leq 3\mathcal{C}t^p \leq 3\mathcal{C}\tilde{T}^p = 3\mathcal{C}\tilde{\sigma}$, where in the last step, we used that in this case $\tilde{T} = \tilde{\sigma}^{1/p}$. Thus, for $t \in [\bar{T}, \tilde{T}]$, we have $|\tilde{r}_t(s_t, a_t)| \leq 3\mathcal{C}\tilde{\sigma}$. As a result, we have $\|\zeta_{t,2}\|_{\infty} = |\tilde{r}_t(s_t, a_t) - R(s_t, a_t)| \leq 3\mathcal{C}\tilde{\sigma} + \bar{R} \leq 4\mathcal{C}\tilde{\sigma}$, since $\mathcal{C} \geq 1$, and $\bar{R} \leq \tilde{\sigma}$.

From our analysis of the two sub-cases above, we conclude that for $t \leq \tilde{T}$, $\|\zeta_{t,2}\|_{\infty} \leq 4\mathcal{C}\tilde{\sigma}$. Next, we bound the adversarial corruption term $\Delta_{t,2}$ in the $\infty$-norm for all $t \in [\tilde{T}]$ as follows:

$$\begin{aligned}
\|\Delta_{t,2}\|_{\infty} &\leq \alpha \left\| \sum_{k=0}^{t} (I - \alpha D)^{t-k} \zeta_{k,2} \right\|_{\infty} \\
&\overset{(*)}{\leq} \alpha \sum_{k=0}^{\tilde{T}-1} \|(I - \alpha D)^{t-k}\|_{\infty} \cdot \|\zeta_{k,2}\|_{\infty} \\
&\overset{(**)}{\leq} 4\mathcal{C}\alpha\tilde{\sigma}\tilde{T}.
\end{aligned} \tag{83}$$

In $(*)$, we apply the triangle inequality, followed by the sub-multiplicative property of the $\infty$-norm. In $(**)$, we use the fact that $\|(I - \alpha D)^{t-k}\|_{\infty} \leq 1$, and that $\|\zeta_{k,2}\|_{\infty} \leq 4\mathcal{C}\tilde{\sigma}$, as established earlier for Case I. This completes Case I.

**Case II:** We now consider the case when $t > \tilde{T}$. Since $\tilde{T} := \max\{\tilde{\sigma}^{1/p}, \bar{T}\}$, it follows that $t > \tilde{T} \Rightarrow t > \bar{T}$. Now recall from the analysis of Lemma 9 that there exists an event $\mathcal{J}$ of measure at least $1 - 2\delta_1 T \geq 1 - \delta/2$, on which, the following holds simultaneously for all time steps $t \in [\bar{T} + 1, T]$:

$$|\bar{r}_t(s_t, a_t) - R(s_t, a_t)| \leq \mathcal{C}\bar{\sigma} \left( \sqrt{\frac{4}{3} \cdot \frac{\log\left(\frac{4}{\delta_1}\right)}{\lambda_{\min}t}} + \sqrt{\varepsilon} \right). \tag{84}$$

On this event, we further have that for $t > \bar{T}$: $|\bar{r}_t(s_t, a_t)| \leq G_t$, where $G_t$ is the original threshold defined in (5). While this condition was enough to prevent any thresholding on event $\mathcal{J}$ for $t > \bar{T}$ for Robust Async-Q, it does not immediately imply that thresholding will not take place for Robust Async-RAQ. The reason for this stems from the fact that in the new algorithm, the modified threshold $\tilde{G}_t$ in (8) can be an under-approximation of $G_t$ during the period $[\bar{T}, \tilde{T}]$. However, for $t > \tilde{T}$, we have $m(t) = t^p > \tilde{T}^p \geq \tilde{\sigma}$, since $\tilde{T} = \max\{\tilde{\sigma}^{1/p}, \bar{T}\}$. As a result, for $t > \tilde{T}$, we have $G_t \leq \tilde{G}_t$. Consequently, on the event $\mathcal{J}$, we have that for all $t > \tilde{T}$, $|\bar{r}_t(s_t, a_t)| \leq G_t < \tilde{G}_t$. Thus, the thresholding operation in line 7 will get bypassed, ensuring that $\tilde{r}_t(s_t, a_t) = \bar{r}_t(s_t, a_t)$, and, as a result, we conclude based on (84) that on event $\mathcal{J}$, for all $t > \tilde{T}$, the following is true:

$$|\tilde{r}_t(s_t, a_t) - R(s_t, a_t)| \leq \mathcal{C}\bar{\sigma}\left(\sqrt{\frac{4}{3} \cdot \frac{\log\left(\frac{4}{\delta_1}\right)}{\lambda_{\min}t}} + \sqrt{\varepsilon}\right). \tag{85}$$

Based on the above bound, we can proceed to control the adversarial term $\Delta_{t,2}$ as follows:

$$\left\|\sum_{k=0}^{t}\alpha(I-\alpha D)^{t-k}\zeta_{k,2}\right\|_{\infty} \leq \left\|\sum_{k=0}^{\tilde{T}}\alpha(I-\alpha D)^{t-k}\zeta_{k,2}\right\|_{\infty} + \left\|\sum_{k=\tilde{T}+1}^{t}\alpha(I-\alpha D)^{t-k}\zeta_{k,2}\right\|_{\infty},$$

$$\leq 4\mathcal{C}\alpha\tilde{\sigma}\tilde{T} + \mathcal{O}(\mathcal{C}\alpha\bar{\sigma})\sqrt{T\frac{\log(4/\delta_1)}{\lambda_{\min}}} + \mathcal{O}\left(\frac{\mathcal{C}\bar{\sigma}\sqrt{\varepsilon}}{\lambda_{\min}}\right),$$

$$\leq 4\mathcal{C}\alpha\tilde{\sigma}\sqrt{\tilde{T}}\cdot\sqrt{T} + \mathcal{O}(\mathcal{C}\alpha\bar{\sigma})\sqrt{T\frac{\log(4/\delta_1)}{\lambda_{\min}}} + \mathcal{O}\left(\frac{\mathcal{C}\bar{\sigma}\sqrt{\varepsilon}}{\lambda_{\min}}\right), \tag{86}$$

$$\leq 4\mathcal{C}\alpha\tilde{\sigma}\sqrt{\bar{T} + \tilde{\sigma}^{\frac{1}{p}}}\cdot\sqrt{T} + \mathcal{O}(\mathcal{C}\alpha\bar{\sigma})\sqrt{T\frac{\log(4/\delta_1)}{\lambda_{\min}}} + \mathcal{O}\left(\frac{\mathcal{C}\bar{\sigma}\sqrt{\varepsilon}}{\lambda_{\min}}\right),$$

$$\leq \mathcal{O}(\mathcal{C}\alpha\tilde{\sigma})\left(\tilde{\sigma}^{1/2p}\sqrt{T} + \sqrt{\frac{T}{\lambda_{\min}}\log\left(\frac{|\mathcal{S}||\mathcal{A}|T}{\delta_1}\right)}\right) + \mathcal{O}\left(\frac{\mathcal{C}\bar{\sigma}\sqrt{\varepsilon}}{\lambda_{\min}}\right) \triangleq \tilde{\Delta}_{t,2}.$$

For the first step, we stitched together the bounds for Cases I and II, and followed a similar reasoning as in the proof of Lemma 9. Under the assumption $T \geq \tilde{T}$, we further used $\tilde{T} \leq \sqrt{\tilde{T}}\cdot\sqrt{T}$. Finally, we leveraged the definition $\tilde{T} = \max\left\{\bar{T}, \tilde{\sigma}^{1/p}\right\}$, which implies $\tilde{T} \leq \bar{T} + \tilde{\sigma}^{1/p}$, and plugged in the expression for $\bar{T}$ from (4), followed by the fact $\bar{\sigma} \leq \tilde{\sigma}$. Hence, combining the bounds obtained in **Case I** and **Case II**, we conclude the proof of Lemma 10. $\qquad\square$

**Step 2: Bound the Non-Adversarial Noise Term** $\Delta_{t,1}$. We now proceed to the more delicate part of the analysis that involves controlling the effect of noise. Like before, to control the noise effect using a martingale-based argument, we will derive uniform bounds on the iterates generated by Robust Async-RAQ. However, as a departure from the analysis in Appendix C, we will derive two sets of bounds: crude bounds that hold deterministically, and finer bounds that hold with high probability. The rationale for this will become clearer soon. We start with the cruder bounds.

**Lemma 11.** *(Coarse Deterministic Bounds on Iterates for Robust Async-RAQ) The following bounds hold deterministically for all $t \in [T]$:*

$$|\eta_{t,1}(s_t, a_t)| \leq \frac{6\mathcal{C}T^p}{1-\gamma}, \quad \|\zeta_{t,1}\|_{\infty} \leq \frac{12\mathcal{C}T^p}{1-\gamma}, \tag{87}$$

*where $\mathcal{C}$ is the universal constant that appears in (5).*

*Proof.* The proof is nearly identical to that of Lemma 6, with the only difference arising from the modified threshold function. Let us start by noting that the following is always true deterministically: $|\tilde{r}_t(s_t, a_t)| \leq \tilde{G}_t, \forall t \geq 0$. Now based on the definition of the modified threshold $\tilde{G}_t$ in (8) and $\bar{T}$ in (4), we have that $\tilde{G}_t = 0, \forall t \leq \bar{T}$, and $\tilde{G}_t \leq 3\mathcal{C}t^p \leq 3\mathcal{C}T^p, \forall t > \bar{T}$. As a result, in Robust Async-RAQ, the reward proxy $\tilde{r}_t(s_t, a_t)$ is deterministically bounded at each time step as $|\tilde{r}_t(s_t, a_t)| \leq \tilde{G}_t \leq 3\mathcal{C}T^p, \forall t \in [T]$. Using this fact, and the exact same inductive reasoning as in the proof of Lemma 6, we can show that:

$$\|Q_t\|_{\infty} \leq \frac{3\mathcal{C}T^p}{1-\gamma}, \forall t \geq 0. \tag{88}$$

Following the same arguments as in Lemma 6, one can then also show that

$$|\eta_{t,1}(s_t, a_t)| \leq \frac{6\mathcal{C}T^p}{1 - \gamma}, \forall t \geq 0. \tag{89}$$

Now fix any state-action pair $(s, a)$, and observe that

$$
\begin{aligned}
|\mathcal{T}Q_t(s, a)| &= |R(s, a) + \gamma\mathbb{E}_{s' \sim \mathcal{P}(\cdot|s,a)}[\max_{a' \in \mathcal{A}} Q_t(s', a')]| \\
&\leq |R(s, a)| + \gamma\mathbb{E}_{s' \sim \mathcal{P}(\cdot|s,a)}[|\max_{a' \in \mathcal{A}} Q_t(s', a')|] \\
&\overset{(a)}{\leq} \tilde{\sigma} + \frac{3\gamma\mathcal{C}T^p}{1 - \gamma} \\
&\overset{(b)}{\leq} 3\mathcal{C}T^p + \frac{3\gamma\mathcal{C}T^p}{1 - \gamma} \\
&= \frac{3\mathcal{C}T^p}{1 - \gamma}.
\end{aligned}
\tag{90}
$$

For (a), we used $|R(s, a)| \leq \tilde{\sigma}$ and Eq. (88). For (b), we used the fact that $T \geq \tilde{T} \implies T^p \geq (\tilde{T})^p \geq \tilde{\sigma} \geq |R(s, a)|$. As a result, $|R(s, a)| \leq 3\mathcal{C}T^p$. Since our analysis above holds for *any* state-action pair, we conclude that $\|\mathcal{T}Q_t\|_\infty \leq 3\mathcal{C}T^p/(1 - \gamma)$. With these developments, we can proceed to bound $\zeta_{t,1}$ as follows:

$$
\begin{aligned}
\|\zeta_{t,1}\|_\infty &\leq |\eta_{t,1}(s_t, a_t)| + \|D_t - D\|_\infty (\|Q_t\|_\infty + \|\mathcal{T}Q_t\|_\infty) \\
&\overset{(a)}{\leq} \frac{6\mathcal{C}T^p}{1 - \gamma} + (\|Q_t\|_\infty + \|\mathcal{T}Q_t\|_\infty) \\
&\overset{(b)}{\leq} \frac{12\mathcal{C}T^p}{1 - \gamma},
\end{aligned}
$$

where (a) follows from (89) and (b) from (88) and the bound we derived on $\|\mathcal{T}Q_t\|_\infty$. This concludes the proof. $\qquad\square$

At this stage, it is instructive to compare the bound on $\|\zeta_{t,1}\|_\infty$ from Lemma 6 with that in Lemma 11 above. While in the former, this bound is on the order of $\mathcal{O}(1)$, it is on the order of $\mathcal{O}(T^p)$ in the latter. As a result, if one were to directly use the bound from Lemma 11 in the standard Azuma Hoeffding inequality (much like what we do in Lemma 7), the resulting final bounds would be vacuous. This calls for a more intricate analysis. In this context, our next result provides a finer bound on $\|\zeta_{t,1}\|_\infty$; however, the price of this finer bound is that it now only holds with high probability.

**Lemma 12.** *(**Finer Probabilistic Bounds on Iterates for Robust Async-RAQ**) The following bounds hold with probability at least $1 - 2\delta_1 T$ for all $t \in [T]$:*

$$|\eta_{t,1}(s_t, a_t)| \leq \frac{6\mathcal{C}\tilde{\sigma}}{1 - \gamma}, \quad \|\zeta_{t,1}\|_\infty \leq \frac{12\mathcal{C}\tilde{\sigma}}{1 - \gamma}, \tag{91}$$

*where $\mathcal{C}$ is the universal constant that appears in* (5).

*Proof.* Let us start by revisiting the bounds on the reward proxy $\tilde{r}_t(s_t, a_t)$ established in Lemma 10. In the proof of Lemma 10, we established that for $t \leq \tilde{T}$, $|\tilde{r}_t(s_t, a_t)| \leq 3\mathcal{C}\tilde{\sigma}$ *deterministically*. Furthermore, we also showed that for $t > \tilde{T}$, the following are true with probability at least $1 - 2\delta_1 T$: (i) $\tilde{r}_t(s_t, a_t) = \bar{r}_t(s_t, a_t)$, and (ii) $|\bar{r}_t(s_t, a_t)| \leq G_t$, where $G_t$ is as in (5). Since $G_t \leq 3\mathcal{C}\tilde{\sigma}, \forall t \geq \tilde{T}$, we conclude that there exists an event of measure at least $1 - 2\delta_1 T$, on which, $|\tilde{r}_t(s_t, a_t)| \leq 3\mathcal{C}\tilde{\sigma}, \forall t \geq 0$. Restricted to this good event, one can now perform the exact same analysis as in the proof of Lemma 6 to establish the claim of this lemma. $\qquad\square$

Based on the previous two results, we now have a martingale difference which exhibits a crude deterministic upper bound, and a finer bound that holds with a fixed high probability. We are in need of a refined version of the Azuma Hoeffding inequality that can exploit this structure. Thankfully, (Shamir & Spencer, 1987, Theorem 7) provides us with precisely the right tool. Our next result is a slight adaptation of this theorem; we provide its proof for completeness.

**Theorem.** *(Probabilistic Azuma-Hoeffding Inequality) (Shamir & Spencer, 1987) Let $X_0, \ldots, X_n$ be a martingale with $X_0$ constant, such that:*

*(i) With probability at least $1 - r$, $|X_{i+1} - X_i| \leq c_i$ for $0 \leq i < n$.*

*(ii) $|X_{i+1} - X_i| \leq b_i$, deterministically.*

*Assume $b_i \cdot r^{\frac{1}{2}} \leq c_i$. Then, the following holds:*

$$\mathbb{P}\left[|X_n - X_0| > \sqrt{\left(32 \sum_{i=1}^{n} c_i^2\right) \log\left(\frac{2}{\delta}\right)} + \sum_{i=0}^{n-1} b_i \cdot r^{1/2}\right] < \delta + 2nr^{1/2}. \tag{92}$$

*Proof.* The core idea behind the proof is to carefully construct a new martingale $\{Y_0, Y_1, \ldots, Y_n\}$ that satisfies the following two properties simultaneously: (i) the martingale differences are "well-behaved" in the sense that $|Y_{i+1} - Y_i| = \mathcal{O}(c_i), \forall i \geq 0$ *deterministically*, and (ii) $|Y_n - X_n|$ is "small" on a good event of sufficient measure. To achieve this, let us start by using $\mathcal{F}_i$ to denote the event $|X_{i+1} - X_i| > c_i$. Next, set $Y_0 = X_0$ and let $p = \mathbb{P}(\mathcal{F}_i | X_i)$. Assuming $Y_i$ has been already defined, we consider two cases:

(A) If $p \geq r^{\frac{1}{2}}$, terminate the martingale by setting $Y_j = Y_i$ for all $j \in [i + 1, n]$.

(B) If $p < r^{\frac{1}{2}}$, and the martingale has not been previously terminated, define:

$$\bar{X}_{i+1} = \begin{cases} X_i & \text{if } \mathcal{F}_i, \\ X_{i+1} & \text{otherwise.} \end{cases}$$

We now have:

$$\mathbb{E}[\bar{X}_{i+1} | X_i] = \mathbb{E}[X_{i+1} | X_i] + \mathbb{E}[\bar{X}_{i+1} - X_{i+1} | X_i] = X_i + A_i, \tag{93}$$

where $A_i \triangleq \mathbb{E}[\bar{X}_{i+1} - X_{i+1} | X_i]$. Then:

$$A_i = \mathbb{E}[\bar{X}_{i+1} - X_{i+1} | X_i, \mathcal{F}_i] \cdot \mathbb{P}(\mathcal{F}_i | X_i).$$

Using the crude bound $|X_{i+1} - X_i| \leq b_i$ and $p = \mathbb{P}(\mathcal{F}_i | X_i) < r^{\frac{1}{2}}$, we obtain:

$$A_i \leq b_i \cdot r^{\frac{1}{2}}, \tag{94}$$

where we used the condition for Case B. With this preparation, we define the sequence $\{Y_i\}$ recursively as follows:

$$Y_{i+1} = Y_i + (\bar{X}_{i+1} - X_i - A_i).$$

Our immediate goal is to establish that $\{Y_{i+1} - Y_i\}$ is a bounded martingale difference sequence. To establish the boundedness aspect, start by observing that

$$|\bar{X}_{i+1} - X_i| = |\bar{X}_{i+1} - X_i| \left(\mathbf{1}_{\mathcal{F}_i} + \mathbf{1}_{\mathcal{F}_i^c}\right) = |X_{i+1} - X_i| \mathbf{1}_{\mathcal{F}_i^c} \leq c_i,$$

where we used the definition of the event $\mathcal{F}_i$ in the last step. Appealing to (94) and using $b_i \cdot r^{\frac{1}{2}} \leq c_i$, we then obtain

$$|Y_{i+1} - Y_i| \leq c_i + b_i \cdot r^{\frac{1}{2}} \leq 2c_i.$$

Next, using (93) and the definition of $Y_{i+1}$, observe that $\mathbb{E}[Y_{i+1} - Y_i | Y_i] = 0$. Thus, $\{Y_n\}_{n \geq 1}$ is indeed a martingale with bounded martingale differences. To proceed, let $\mathcal{G}$ be the "good event" where Case A never occurs, and $\mathcal{F}_i$ never occurs. On this event, it follows from our construction that

$$Y_n = X_n - \sum_{i=0}^{n-1} A_i.$$

Therefore, we get the following deterministic bound on event $\mathcal{G}$:

$$|Y_n - X_n| = \left| \sum_{i=0}^{n-1} A_i \right| \leq r^{\frac{1}{2}} \sum_{i=0}^{n-1} b_i. \tag{95}$$

Thus, on the good event, the above display provides control over the difference between our martingale of interest $\{X_n\}$, and the martingale we constructed $\{Y_n\}$. To gain control over the bad event $\mathcal{G}^c$, our next task is to get a bound on $\mathbb{P}(\mathcal{G}^c)$. To that end, we will require the following estimate:

$$\begin{aligned}
\mathbb{P}\left( \mathbb{P}(\mathcal{F}_i|X_i) > r^{\frac{1}{2}} \right) &\leq \frac{\mathbb{E}[\mathbb{P}(\mathcal{F}_i|X_i)]}{r^{\frac{1}{2}}} \\
&= \frac{\mathbb{E}[\mathbb{E}[\mathbf{1}_{\mathcal{F}_i}|X_i]]}{r^{\frac{1}{2}}} \\
&= \frac{\mathbb{E}[\mathbf{1}_{\mathcal{F}_i}]}{r^{\frac{1}{2}}} \\
&= \frac{\mathbb{P}(\mathcal{F}_i)}{r^{\frac{1}{2}}} \\
&\leq r^{\frac{1}{2}},
\end{aligned} \tag{96}$$

where for the first step, we used Markov's inequality, and for the last step, we used the fact that $\mathbb{P}(\mathcal{F}_i) \leq r$. Using the above estimate, we then have using union-bounding:

$$\begin{aligned}
\mathbb{P}(\mathcal{G}^c) &= \mathbb{P}\left( \{\cup_i \mathcal{F}_i\} \bigcup \{\cup_i \mathbb{P}(\mathcal{F}_i|X_i) > r^{\frac{1}{2}}\} \right) \\
&\leq \mathbb{P}(\cup_i \mathcal{F}_i) + \mathbb{P}(\cup_i \mathbb{P}(\mathcal{F}_i|X_i) > r^{\frac{1}{2}}) \\
&\leq \sum_{i=1}^{n} \mathbb{P}(\mathcal{F}_i) + \sum_{i=1}^{n} \mathbb{P}\left( \mathbb{P}(\mathcal{F}_i|X_i) > r^{\frac{1}{2}} \right) \leq nr + nr^{\frac{1}{2}} \\
&\leq 2nr^{\frac{1}{2}}.
\end{aligned} \tag{97}$$

Now, we can finally arrive at the following bound:

$$\begin{aligned}
&\mathbb{P}\left[ |X_n - X_0| > \sqrt{\left( 32 \sum_{i=1}^{n} c_i^2 \right) \log\left(\frac{2}{\delta}\right)} + \sum_{i=0}^{n-1} b_i \cdot r^{1/2} \right] \\
&\overset{(*)}{\leq} \mathbb{P}\left[ |X_n - Y_n| + |Y_n - Y_0| > \sqrt{\left( 32 \sum_{i=1}^{n} c_i^2 \right) \log\left(\frac{2}{\delta}\right)} + \sum_{i=0}^{n-1} b_i \cdot r^{1/2} \right] \\
&\overset{(**)}{\leq} \mathbb{P}\left[ \left\{ |X_n - Y_n| > \sum_{i=0}^{n-1} b_i \cdot r^{1/2} \right\} \cup \left\{ |Y_n - Y_0| > \sqrt{\left( 32 \sum_{i=1}^{n} c_i^2 \right) \log\left(\frac{2}{\delta}\right)} \right\} \right] \\
&\overset{(***)}{\leq} \mathbb{P}(\mathcal{G}^c) + \mathbb{P}\left[ |Y_n - Y_0| > \sqrt{\left( 32 \sum_{i=1}^{n} c_i^2 \right) \log\left(\frac{2}{\delta}\right)} \right] \\
&\leq 2nr^{\frac{1}{2}} + \delta.
\end{aligned} \tag{98}$$

In step $(*)$, we apply the triangle inequality, which states that $|X_n - X_0| \leq |X_n - Y_n| + |Y_n - Y_0|$, allowing us to bound the original probability by replacing $|X_n - X_0|$ with $|X_n - Y_n| + |Y_n - Y_0|$. In step $(**)$, we use the union bound, which ensures that $\mathbb{P}(\mathcal{A} + \mathcal{B} > \mathcal{Q}) \leq \mathbb{P}(\mathcal{A} > \mathcal{Q}_1) + \mathbb{P}(\mathcal{B} > \mathcal{Q}_2)$, where $\mathcal{Q}_1 + \mathcal{Q}_2 = \mathcal{Q}$. Finally, in step $(***)$, we use the bound $\mathbb{P}(\mathcal{G}^c) \leq 2nr^{1/2}$ for the first term, as $|X_n - Y_n|$ is controlled by the good event $\mathcal{G}$, and the second term is bounded by $\delta$ via an application of Azuma-Hoeffding (Lemma 3) to the martingale $Y_n$. With this, our proof is complete. $\qquad\square$

Armed with the previous result, we are now in a position to control the noise term in `Robust Async-RAQ`.

**Lemma 13.** *(Bounding Non-Adversarial Noise in Robust Async-RAQ) Suppose $\delta_1 \leq \delta^2/128|\mathcal{S}|^2|\mathcal{A}|^2T^{2p+3}$. Then, with probability at least $1 - \delta/2$, the following bound holds simultaneously for all $t \in [T]$:*

$$\left\| \sum_{k=0}^{t} \alpha(I - \alpha D)^{t-k}\zeta_{k,1} \right\|_{\infty} \leq \mathcal{O}\left(\frac{\tilde{\sigma}}{1-\gamma}\right) \cdot \sqrt{\frac{\alpha}{\lambda_{\min}} \log\left(\frac{8|\mathcal{S}||\mathcal{A}|T}{\delta}\right)} + \mathcal{O}\left(\frac{\alpha}{1-\gamma}\right), \tag{99}$$

*where $\zeta_{k,1}$ is defined in Eq. (34).*

*Proof.* We follow the proof idea in Lemma 7, but invoke the finer variant of the Azuma–Hoeffding inequality in Theorem 5. To that end, for a fixed state-action pair $(s,a)$, recall that $\{\alpha(1 - \alpha\lambda(s,a))^{t-i}\zeta_{i,1}(s,a)\}_{i\in[t]}$ is a martingale difference sequence. As per the notation of Theorem 5, using Lemmas 11 and 12, $b_i$ and $c_i$ are the cruder deterministic and finer probabilistic bounds, respectively, on the $i$-th term of this sequence:

$$c_i = \frac{12C\tilde{\sigma}}{1-\gamma} \cdot \alpha(1 - \alpha\lambda(s,a))^{t-i}, \quad b_i = \frac{12CT^p}{1-\gamma} \cdot \alpha(1 - \alpha\lambda(s,a))^{t-i}, \quad r = 2\delta_1 T. \tag{100}$$

To satisfy the condition $b_i \cdot r^{1/2} \leq c_i$ that is required to apply Theorem 5, it suffices to ensure:

$$(2\delta_1 T)^{1/2} \cdot T^p \leq \tilde{\sigma}. \tag{101}$$

Since $\tilde{\sigma} \geq 1$, the above condition can be ensured by requiring

$$(2\delta_1 T)^{1/2} \cdot T^p \leq 1 \iff \delta_1 \leq 1/(2T^{2p+1}). \tag{102}$$

Assuming the above requirement holds, for a fixed $(s,a) \in \mathcal{S} \times \mathcal{A}$ and $t \in [T]$, Theorem 5, when applied with the parameter choices in Eq. (100), implies that with probability at least $1 - \bar{\delta} - 2T(2\delta_1 T)^{1/2}$, the following holds:

$$\left| \sum_{k=0}^{t} \alpha(1 - \alpha\lambda(s,a))^{t-k}\zeta_{k,1}(s,a) \right| \leq \mathcal{O}\left(\frac{\tilde{\sigma}}{1-\gamma}\right) \cdot \sqrt{\frac{\alpha}{\lambda_{\min}} \log\left(\frac{2}{\bar{\bar{\delta}}}\right)} + \mathcal{O}\left(\frac{\alpha T^{p+1}}{1-\gamma} \cdot (2\delta_1 T)^{1/2}\right). \tag{103}$$

As an immediate next step, applying an union bound over all $(s,a) \in \mathcal{S} \times \mathcal{A}$ and $t \in [T]$, the bound in Eq. (103) holds simultaneously for all state-action pairs and time steps with probability at least

$$1 - \underbrace{|\mathcal{S}||\mathcal{A}|T\bar{\delta}}_{(\bullet)} - \underbrace{2|\mathcal{S}||\mathcal{A}|T^2(2\delta_1 T)^{1/2}}_{(\bullet\bullet)}. \tag{104}$$

Next, we impose the following additional conditions on the failure probability $\delta_1$ to control the second term in Eq. (103), and to ensure that Eq. (103) holds with probability at least $1 - \delta/2$:

$$(2\delta_1 T)^{1/2} \cdot T^{p+1} \leq 1, \quad \underbrace{2|\mathcal{S}||\mathcal{A}|T^2(2\delta_1 T)^{1/2}}_{(\bullet\bullet)} \leq \delta/4. \tag{105}$$

Combining all the constraints on $\delta_1$ from Eq. (102) and Eq. (105), we arrive at the final condition on the failure probability $\delta_1$ as follows:

$$(2\delta_1 T)^{\frac{1}{2}} \leq \delta/(8|\mathcal{S}||\mathcal{A}|T^{p+1}) \implies \delta_1 \leq \delta/(128|\mathcal{S}|^2|\mathcal{A}|^2T^{2p+3}). \tag{106}$$

Now by ensuring that term $(\bullet) \leq \delta/4$ and applying the final requirement on the failure probability from Eq. (106), we conclude that the following bound holds for all state-action pairs $(s,a) \in \mathcal{S} \times \mathcal{A}$, and $t \in [T]$ with probability at least $1 - \delta/2$:

$$\left| \sum_{k=0}^{t} \alpha(1 - \alpha\lambda(s,a))^{t-k}\zeta_{k,1}(s,a) \right| \leq \mathcal{O}\left(\frac{\tilde{\sigma}}{1-\gamma}\right) \cdot \sqrt{\frac{\alpha}{\lambda_{\min}} \log\left(\frac{8|\mathcal{S}||\mathcal{A}|T}{\delta}\right)} + \mathcal{O}\left(\frac{\alpha}{1-\gamma}\right). \tag{107}$$

Hence, given $\delta_1 \leq \delta/(128|\mathcal{S}|^2|\mathcal{A}|^2T^{2p+3})$, the following also holds with probability at least $1 - \frac{\delta}{2}$:

$$\left\| \sum_{k=0}^{t} \alpha(I - \alpha D)^{t-k}\zeta_{k,1} \right\|_{\infty} = \max_{(s,a)\in\mathcal{S}\times\mathcal{A}} \left| \sum_{k=0}^{t} \alpha(1 - \alpha\lambda(s,a))^{t-k}\zeta_{k,1}(s,a) \right|$$

$$\leq \mathcal{O}\left(\frac{\tilde{\sigma}}{1-\gamma}\right) \cdot \sqrt{\frac{\alpha}{\lambda_{\min}} \log\left(\frac{8|\mathcal{S}||\mathcal{A}|T}{\delta}\right)} + \mathcal{O}\left(\frac{\alpha}{1-\gamma}\right) \triangleq \tilde{\Delta}_{t,1}. \tag{108}$$

$\square$

**Finite-Time Rates for Robust Async-RAQ (Proof of Theorem 4)**. Having established bounds on the non-adversarial and adversarial terms via Lemma 13 and Lemma 10, respectively, we proceed by adopting the exact same inductive strategy as in Section C for the proof of Theorem 2. Keeping the notation same, in Robust Async-RAQ, we define the total perturbation term as $\Delta = \tilde{\Delta}_{t,1} + \tilde{\Delta}_{t,2}$, and mimic the inductive proof of Theorem 2 to establish that the exact same bound as in (65) holds with probability at least $1 - \delta$. Finally, substituting $\alpha = \dfrac{\log T}{\lambda_{\min} T(1-\gamma)}$, and simplifying, we arrive at the following bound with probability at least $1 - \delta$:

$$\|d_T\|_\infty \leq \frac{\|d_0\|_\infty}{T} + \mathcal{O}\left( \frac{\tilde{\sigma}^{1+1/2p}}{(1-\gamma)^{\frac{5}{2}}} \frac{\log T}{\lambda_{\min}^{\frac{3}{2}} \sqrt{T}} \sqrt{\log\left(\frac{|\mathcal{S}||\mathcal{A}|T}{\delta}\right)} + \frac{\bar{\sigma}\sqrt{\varepsilon}}{\lambda_{\min}(1-\gamma)} \right). \tag{109}$$

With this, we complete the proof of the finite-time convergence rate for Robust Async-RAQ.

# F. Extension to the Markov Setting and Proof of Theorem 6

The goal of this section is to extend the analysis of `Robust Async-RAQ` from the asynchronous i.i.d. sampling setting to the Markovian data setting. The main difficulty in the Markovian setting is that consecutive samples are no longer independent: the state-action pairs observed along a single trajectory are correlated through the underlying Markov chain. To recover an approximately independent sampling structure, we follow the standard *blocking* idea (Dorfman & Levy, 2022): instead of using every sample, the algorithm retains only every $\tau$-th sample, where $\tau$ is chosen on the order of the mixing time of the Markov chain. This allows us to reduce the Markovian analysis to an i.i.d.-like analysis, at the cost of replacing the total sample size $t$ by the effective sample size $t/\tau$.

---

**Algorithm 3** Robust Asynchronous $Q$-learning Algorithm–Markovian (`Robust Async-RAQ-M`)

---

1: **Require:** Step-size $\alpha$, corruption fraction $\varepsilon$, confidence level $\delta$, mixing time $\bar{\tau}$, iteration count $T$.
2: Initialize datasets $\mathcal{D}_0(s,a) = \emptyset$, for all $(s,a) \in \mathcal{S} \times \mathcal{A}$, and $Q$-table $Q_0 \leftarrow 0$.
3: Set block size $\tau = \lceil \lceil \log(2T/\delta)/\log 2 \rceil \cdot \bar{\tau} \rceil$.
4: **for** $t = 0, \dots, T-1$ **do**
5:     Observe data tuple $\{s_t, a_t, s_{t+1}\}$, and reward $y_t(s_t, a_t)$.
6:     **if** $t \mod \tau = 0$ **then**
7:         Append $y_t(s_t, a_t)$ to $\mathcal{D}_t(s_t, a_t)$, and compute $\bar{r}_t(s_t, a_t) \leftarrow \text{TRIM}[\mathcal{D}_t(s_t, a_t), \varepsilon, \delta_1]$.
8:         **if** $|\bar{r}_t(s_t, a_t)| > \widetilde{G}_t^{\text{M}}$ in Eq. (110) **then**
9:             Set $\tilde{r}_t(s_t, a_t) \leftarrow 0$.
10:         **else**
11:             Set $\tilde{r}_t(s_t, a_t) \leftarrow \bar{r}_t(s_t, a_t)$.
12:         **end if**
13:         Update $Q_{t+1}$ using Eq. (6).
14:     **else**
15:         Continue to `Line 4`.
16:     **end if**
17: **end for**

---

• **Modified Threshold in the Markovian Setting.** In the Markovian case, the threshold must account for the loss in effective sample size caused by temporal dependence. Accordingly, we modify the threshold in (8) by incorporating the sub-sampling parameter $\tau$ in the concentration term, while keeping the truncation envelope $m(t)$ as before. The resulting Markovian threshold is given by

$$\widetilde{G}_t^{\text{M}} = \begin{cases} 0, & \text{if } t \leq \tau\bar{T}, \\ \mathcal{C}\, m(t) \left( \sqrt{\frac{4\tau \log(8/\delta_1)}{3\lambda_{\min}t}} + \sqrt{\varepsilon} \right) + m(t), & \text{if } t > \tau\bar{T}. \end{cases} \tag{110}$$

The Markovian modification enters through the effective sample-size calculation: the burn-in time is inflated from $\bar{T}$ to $\tau\bar{T}$, and the concentration term uses only $\mathcal{O}(\lambda_{\min}t/\tau)$ effectively independent samples. On the good event $\mathcal{K}$ from Lemma 8, for every $(s,a)$ and every $t \geq \bar{T}$, $\mathcal{N}_t(s,a) \geq (3/4)\lambda_{\min}t$. In the Markovian setting, however, these visits are temporally dependent. After accounting for mixing, only a $1/\tau$-fraction of them can be treated as effectively independent, so the concentration step uses an effective sample size of order $\lambda_{\min}t/\tau$. This produces the $\sqrt{\tau}$-inflation in the threshold shown in (110). The truncation envelope $m(t)$ is not affected by sub-sampling, and therefore is left unchanged.

Next, to keep the paper self-contained, we review other key ingredients needed for the Markovian setting, drawing primarily on Dorfman & Levy (2022).

• **Background.** Let $\{Z_t\}$ be an ergodic time-homogeneous Markov chain over a finite-state space $\Omega$ with stationary distribution $\rho$. Define

$$d_{mix}(t) := \sup_{Z \in \Omega} D_{TV}\left( \mathbb{P}(Z_t \in \cdot \mid Z_0 = Z), \rho \right). \tag{111}$$

Then, $d_{mix}(t)$ is a non-increasing function of $t$. We define the *mixing time* as

$$\bar{\tau} := \inf\{t \mid d_{mix}(t) \leq 1/4\}. \tag{112}$$

Intuitively, the mixing time measures *how fast the state distribution approaches stationarity.* We then have the following key fact (Dorfman & Levy, 2022):

$$d_{mix}(\ell\bar{\tau}) \leq 2^{-\ell}, \quad \forall \ell \in \mathbb{N}. \tag{113}$$

With the notations specified above, we then introduce the following theorem that will play a crucial role in our extension to the Markov setting.

**Theorem 8.** *Let $Z_0, Z_1, \cdots$ be a stationary finite-state Markov chain with stationary distribution $\rho$, and let $K, n \in \mathbb{N}$. Then, we can couple $Z_{K,n} := (Z_0, Z_K, \cdots, Z_{(n-1)K})$ and $\tilde{Z}_{K,n} := (\tilde{Z}_0, \tilde{Z}_K, \cdots, \tilde{Z}_{(n-1)K}) \sim \rho^{\otimes n}$, such that*

$$\mathbb{P}\left(Z_{K,n} \neq \tilde{Z}_{K,n}\right) \leq (n-1)d_{mix}(K). \tag{114}$$

The proof of this theorem can be found in Nagaraj et al. (2020). Intuitively, Theorem 8 states that if we subsample a sequence from an ergodic Markov chain with sufficiently large sampling interval, then with high probability, the sub-sampled sequence is identical to its i.i.d. counterpart sampled from the stationary distribution of that Markov chain. Let us now see how these ideas can be exploited for our setting.

**Extension to the Markov Setting.** Recall that $\mu$ is the behavior policy that generates data in our problem. Let the trajectory generated by this policy be $\{s_0, a_0, s_1, a_1, \cdots\}$. Note that $\{Z_t\} := \{(s_t, a_t, s_{t+1})\}$ is also a Markov chain, and that it is ergodic in light of Assumption 1; see Chen et al. (2019). Suppose this chain is initialized from its stationary distribution $\rho$. Let $\bar{\tau}$ be the mixing time of this Markov chain.

We now propose a simple modification to `Robust Async-RAQ` that is based on dropping certain data points. To see how this can be done, we define a *block* parameter $\tau := \lceil \ell\bar{\tau} \rceil$, where $\ell = \lceil \log(2T/\delta)/\log 2 \rceil$. The only modification to `Robust Async-RAQ` is that the agent now uses every $\tau$-th sample, and drops the rest; this variant is formally described in Algorithm 3.

To analyze Algorithm 3, we note that it essentially runs on $n = T/\tau$ samples; for simplicity, we assume that $n$ is an integer. Specifically, the learner only uses the data set $\{Z_0, Z_\tau, \cdots, Z_{(n-1)\tau}\}$. Let $\{\tilde{Z}_0, \tilde{Z}_\tau, \cdots, \tilde{Z}_{(n-1)\tau}\} \sim \rho^{\otimes n}$ be i.i.d. samples drawn from the stationary distribution $\rho$. From the coupling theorem, namely Theorem 8, given any $\delta \in (0, 1)$, we then have

$$\begin{aligned}
\mathbb{P}\left(\{Z_0, Z_\tau, \cdots, Z_{(n-1)\tau}\} \neq \{\tilde{Z}_0, \tilde{Z}_\tau, \cdots, \tilde{Z}_{(n-1)\tau}\}\right) &\leq nd_{mix}(\tau) \\
&= nd_{mix}(\lceil \ell\bar{\tau} \rceil) \\
&\leq \frac{T}{\tau} \cdot 2^{-\ell} \\
&\leq T \cdot 2^{-\ell} \\
&\leq T \cdot \frac{\delta}{2T} \\
&= \frac{\delta}{2},
\end{aligned} \tag{115}$$

where we used the key fact (113), the definition of $\ell$, and the fact that $d_{mix}(t)$ is non-increasing in $t$.

Thus, there exists a "good event", say $\mathcal{B}$, of measure at least $1 - \delta/2$, on which

$$\{Z_0, Z_\tau, \cdots, Z_{(n-1)\tau}\} = \{\tilde{Z}_0, \tilde{Z}_\tau, \cdots, \tilde{Z}_{(n-1)\tau}\}. \tag{116}$$

Equation (116) states that on the good event $\mathcal{B}$, the sub-sampled Markovian data is identical to its i.i.d. counterpart. To see how this result can be exploited, let us recall the guarantee from Theorem 4 when `Robust Async-RAQ` is run on $n = (T/\tau)$ i.i.d. samples with

$$T > \max\{\tau\bar{T}, \tau\log(T)/(\lambda_{\min}(1-\gamma))\} \quad \text{and} \quad \alpha = \frac{\tau\log T}{\lambda_{\min}(1-\gamma)T}.$$

In this setting, the following holds with probability $1 - \delta/2$:

$$\|d_n\|_\infty \leq \frac{\|d_0\|_\infty}{T} + c_1 \underbrace{\left(\frac{\tilde{\sigma}^{1+1/2p}}{(1-\gamma)^{\frac{5}{2}}} \frac{\log T}{\lambda_{\min}^{\frac{3}{2}}\sqrt{T}} \sqrt{\tau\log\left(\frac{|\mathcal{S}||\mathcal{A}|T}{\delta}\right)}\right)}_{\Psi} + c_2\left(\frac{\tilde{\sigma}\sqrt{\varepsilon}}{\lambda_{\min}(1-\gamma)}\right), \tag{117}$$

where $c_1$ and $c_2$ are suitable universal constants.

Now consider running Algorithm 3, which we denote by $\mathcal{A}$ for convenience, on the $n$ subsampled Markov tuples $\mathcal{D} := (Z_0, Z_\tau, \ldots, Z_{(n-1)\tau})$. Let the output of $\mathcal{A}$ in this case be

$$Q_n := \mathcal{A}(\mathcal{D}; \mathcal{U}), \quad \text{where} \quad \mathcal{U} := \underbrace{\{(Y_{k\tau}, n_{k\tau})\}_{k=0}^{(n-1)}}_{\mathcal{U}_1}, \underbrace{\{(z_{k\tau})\}_{k=0}^{(n-1)}}_{\mathcal{U}_2} \tag{118}$$

collects the auxiliary randomness associated with our problem. All of these components are formally defined in Eq. (2).

Next, let $\tilde{Q}_n := \mathcal{A}(\tilde{\mathcal{D}}; \mathcal{U})$ be the output of the algorithm $\mathcal{A}$ when it is fed with the same auxiliary randomness $\mathcal{U}$, but with the i.i.d. subsampled data set $\tilde{\mathcal{D}} := (\tilde{Z}_0, \tilde{Z}_\tau, \ldots, \tilde{Z}_{(n-1)\tau}) \sim \rho^{\otimes n}$. On the coupling event $\mathcal{B}$, we have $\mathcal{D} = \tilde{\mathcal{D}}$, and hence $Q_n = \tilde{Q}_n$ on event $\mathcal{B}$. In simple words, the event $\mathcal{B}$ ensures that the sub-sampled Markov dataset $\mathcal{D}$ and the i.i.d. dataset $\tilde{\mathcal{D}}$ coincide, so that given the same $\mathcal{U}$, both executions of the algorithm $\mathcal{A}$ produce identical outputs. We then have:

$$\begin{aligned}
\mathbb{P}(\{\|Q_n - Q^*\|_\infty > \Psi\}) &= \mathbb{P}(\{\|Q_n - Q^*\|_\infty > \Psi\} \cap \mathcal{B}) + \mathbb{P}(\{\|Q_n - Q^*\|_\infty > \Psi\} \cap \mathcal{B}^c) \\
&\leq \mathbb{P}(\{\|Q_n - Q^*\|_\infty > \Psi\} \cap \mathcal{B}) + \mathbb{P}(\mathcal{B}^c) \\
&\overset{(a)}{\leq} \mathbb{P}(\{\|\tilde{Q}_n - Q^*\|_\infty > \Psi\} \cap \mathcal{B}) + \mathbb{P}(\mathcal{B}^c) \\
&\overset{(b)}{\leq} \mathbb{P}(\{\|\tilde{Q}_n - Q^*\|_\infty > \Psi\}) + \delta/2 \\
&\overset{(c)}{\leq} \delta.
\end{aligned} \tag{119}$$

In the above steps, for (a), we used the fact that $Q_n = \tilde{Q}_n$ on event $\mathcal{B}$. For (b), we appealed to (115), and for (c), we used (117). Thus, via the coupling argument above, we have established that with probability at least $1 - \delta$, the following is true:

$$\|Q_n - Q^*\|_\infty \leq \Psi,$$

with $\Psi$ as in (117). This is precisely what was needed to be shown to establish Theorem 6.

## G. Experimental Results

We evaluate the performance of our proposed algorithms in a synthetic grid-world environment. The underlying Markov Decision Process (MDP) consists of $|\mathcal{S}| = 25$ states and $|\mathcal{A}| = 10$ actions, with discount factor $\gamma = 0.7$. The true mean rewards are bounded within the interval $[0, 10]$, and the reward variance $\sigma^2$ is upper bounded by 10. To assess robustness, we consider an adversarial corruption model in which, at each corrupted time step, the adversary injects a fixed bias of $-10^4$. Each plot in Figure 1 reports the average over 100 independent runs. Our simulations reveal that: (i) vanilla asynchronous $Q$-learning incurs large errors under the corruption model in (2); (ii) `Robust Async-Q` continues to converge to a neighborhood of $Q^*$ despite adversarial influences. For `Robust Async-RAQ`, our simulations also illustrate the effect of the reward-agnostic parameter $p$ in the threshold function $m(t) = t^p$ in (8).

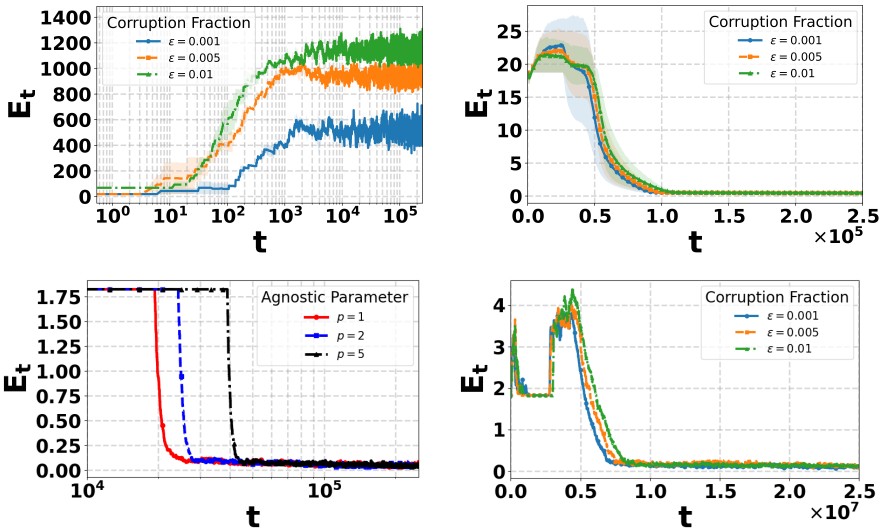

*Figure 1.* **(Top Left)** $\ell_\infty$ error $E_t = \|Q_t - Q^*\|_\infty$ for `Vanilla-Q` under the Huber-contaminated reward model in Eq. (2), with $\varepsilon \in \{0.001, 0.005, 0.01\}$, variance $\sigma^2 = 5$, and a $-10^4$ biasing attack. **(Top Right)** $E_t$ for `Robust Async-Q` under the same corruption levels, noise statistics, and attack. **(Bottom Left)** $E_t$ for `Robust Async-RAQ` with $\varepsilon = 0.001$ and reward-agnostic parameter $p \in \{1, 2, 5\}$. **(Bottom Right)** $E_t$ for `Robust Async-Q` **under Markovian sampling**, with $\varepsilon \in \{0.001, 0.005, 0.01\}$.

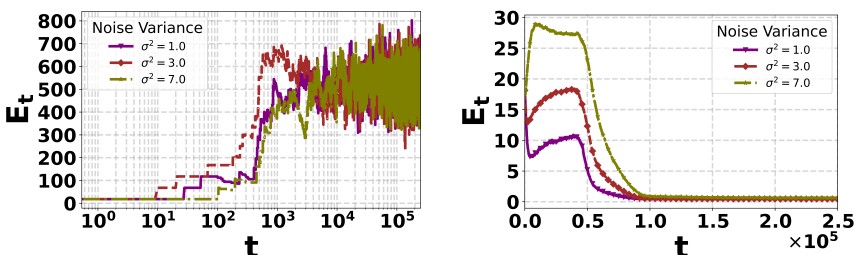

*Figure 2.* **(Left)** $\ell_\infty$ error $E_t = \|Q_t - Q^*\|_\infty$ for `Vanilla-Q` under the Huber-contaminated reward model in Eq. (2), with $\varepsilon \in \{0.002\}$, variance $\sigma^2 = \{1, 3, 7\}$, and a $-10^3$ biasing attack. **(Right)** $E_t$ for `Robust Async-Q` under the same corruption level, noise variances, and biasing attack of $-10^3$. Each plot in Figure 2 reports the average over 100 independent runs.

### G.1. Additional Experiments

We now evaluate the performance of `Robust Async-Q/RAQ` on some additional grid-world tasks and Gymnasium environments.

**MDP 1.** The environment is modeled as an MDP with $|\mathcal{S}| = 50$ states, $|\mathcal{A}| = 20$ actions, and discount factor $\gamma = 0.9$. The true mean rewards lie in $[0, 20]$, and the reward variance satisfies $\sigma^2 \leq 10$. To test robustness, we use an adversarial corruption model defined in Eq. (2) where, at each corrupted time step, the reward is shifted by a bias of $-10^5$. We report the plots in Figure 4.

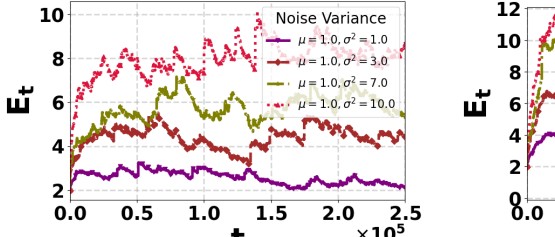

*Figure 3.* **(Left)** The $\ell_\infty$ error $E_t = \|Q_t - Q^*\|_\infty$ for `Vanilla-Q` under heavy-tailed reward noise with no corruption. The noise has mean $\mu = 1$ and variance $\sigma^2 \in \{1, 3, 7, 10\}$, while higher moments may be infinite. We model the heavy-tailed rewards using a scaled Student-$t$ distribution. **(Right)** The corresponding $E_t$ for `Robust Async-Q` under the same noise model. Even in the absence of corruption, heavy-tailed noise with only finite mean and variance can substantially degrade `Vanilla-Q`, whereas our robust variant remains stable. Each curve in Figure 3 is averaged over 100 independent runs.

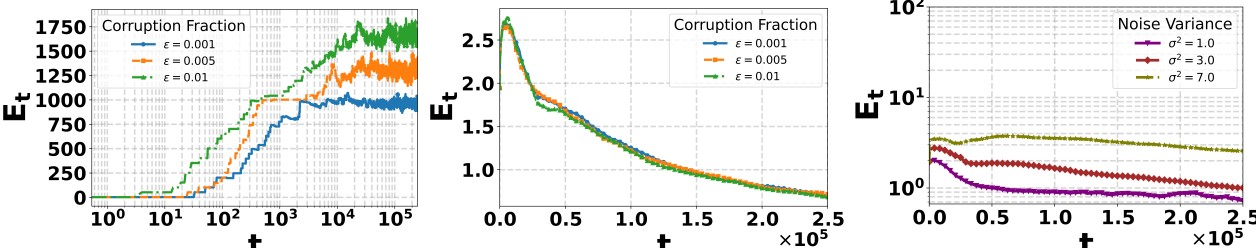

*Figure 4.* **(Left)** $\ell_\infty$ error $E_t = \|Q_t - Q^*\|_\infty$ for `Vanilla-Q` under the Huber-contaminated reward model in Eq. (2), with $\varepsilon \in \{0.001, 0.005, 0.01\}$, variance $\sigma^2 = 0.5$, and a $-10^5$ biasing attack. **(Middle)** $E_t$ for `Robust-Async-Q` with $\varepsilon \in \{0.001, 0.005, 0.01\}$, variance $\sigma^2 = 0.5$, and a $-10^5$ biasing attack. **(Right)** $\ell_\infty$ error $E_t = \|Q_t - Q^*\|_\infty$ for `Robust-Async-Q` under the Huber-contaminated reward model in Eq. (2), with $\varepsilon \in \{0.01\}$, variance $\sigma^2 = \{1, 3, 7\}$, and a $-10^5$ biasing attack. Each plot in Figure 4 reports the average over 100 independent runs.

**MDP 2.** We model the environment as an `MDP` with $|\mathcal{S}| = 100$ states, $|\mathcal{A}| = 50$ actions, and discount factor $\gamma = 0.9$. The true mean rewards lie in $[0, 5]$, and the reward noise has variance $\sigma^2 \leq 10$. To evaluate robustness, we consider the adversarial corruption model in Eq. (2) which, on corrupted iterations $t \in \{1, \ldots, T\}$, the observed reward is shifted by a negative bias of $-10^t$. We report the plots in Figure 5.

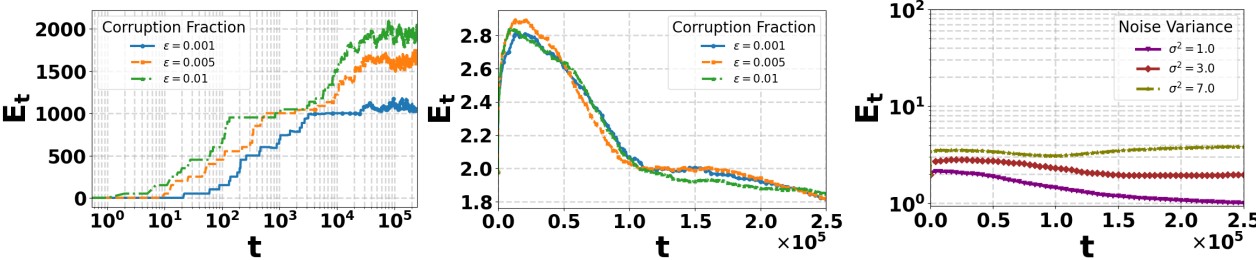

*Figure 5.* **(Left)** $\ell_\infty$ error $E_t = \|Q_t - Q^*\|_\infty$ for `Vanilla-Q` under the Huber-contaminated reward model in Eq. (2), with $\varepsilon \in \{0.001, 0.005, 0.01\}$, variance $\sigma^2 = 1$, reward-agnostic parameter $p = 3$ and a $-10^t$ biasing attack. **(Middle)** $E_t$ for `Robust-Async-RAQ` with $\varepsilon \in \{0.001, 0.005, 0.01\}$, variance $\sigma^2 = 1$, reward-agnostic parameter $p = 3$ and a $-10^t$ biasing attack. **(Right)** $\ell_\infty$ error $E_t = \|Q_t - Q^*\|_\infty$ for `Robust-Async-RAQ` under the Huber-contaminated reward model in Eq. (2), with $\varepsilon \in \{0.01\}$, variance $\sigma^2 = \{1, 3, 7\}$, reward-agnostic parameter $p = 3$ and a $-10^t$ biasing attack. Each plot in Figure 5 reports the average over 100 independent runs.

■ **`FrozenLake-v1`.** We now evaluate `Robust Async-Q` in the `FrozenLake-v1` (Towers et al., 2024) ($8 \times 8$) environment under non-slippery conditions, and under the adversarial reward-corruption model defined in Eq. (2). The agent interacts with the underlying MDP using the true transition dynamics. The simulation results for `FrozenLake-v1` are presented in Figures 6 and 7.

● `Goal`: In `FrozenLake-v1`, the agent aims to navigate from the start to the goal across the frozen grid while avoiding holes that terminate the episode; it receives a reward of 1 upon reaching the goal and 0 otherwise.

• `Adversarial Reward Model for` **`FrozenLake-v1`**: We consider a reward corruptive adversary following a Huber contamination model defined in Eq. (2) that acts *only* upon successful termination. Specifically, whenever an episode terminates by reaching the goal state, an adversary independently corrupts the observed terminal reward with probability $\varepsilon$ by replacing it with a large negative outlier of the form $-10^{\beta}$, where the exponent $\beta \in \{1, \dots, 9\}$ may be chosen arbitrarily. Otherwise, with probability $1 - \varepsilon$, the agent observes an uncorrupted but noisy terminal reward $r_{\texttt{Terminal}}$ with mean $\mu_{\texttt{Terminal}} = 1$ and variance $\sigma^2_{\texttt{Terminal}} \leq 2$. All non-terminal rewards remain uncorrupted and noisy, with mean $\mu_{\sim \texttt{Terminal}} = 0$ and variance $\sigma^2_{\sim \texttt{Terminal}} \leq 3$. Thus, the attack preserves the MDP dynamics and the episode termination condition, and perturbs learning solely through occasional adversarial spikes on goal-reaching transitions. The $Q$-table is updated using these training rewards under an $\varepsilon$-greedy policy with a decaying exploration schedule, and we compare cumulative training rewards and learned greedy policies across varying corruption levels.

• `Training:` In sparse-reward tasks such as `FrozenLake-v1`, `Vanilla-Q` can be highly sensitive to rare but extreme corrupted terminal rewards: a single outlier can create a large TD update and, via bootstrapping, propagate to many predecessor state–action pairs, leading to unstable value estimates and a degraded greedy policy. A robust variant, such as `Robust Async-Q` that robustly aggregates multiple samples per state-action pair $(s, a) \in \mathcal{S} \times \mathcal{A}$ using a robust estimator outlined in Appendix B and rejects overly large updates via thresholding following Eq. 5 should largely contain these outliers, yielding smoother learning dynamics and a policy closer to the clean-optimal behavior even under nontrivial corruption. We ran the training for $t = 90000$ episodes.

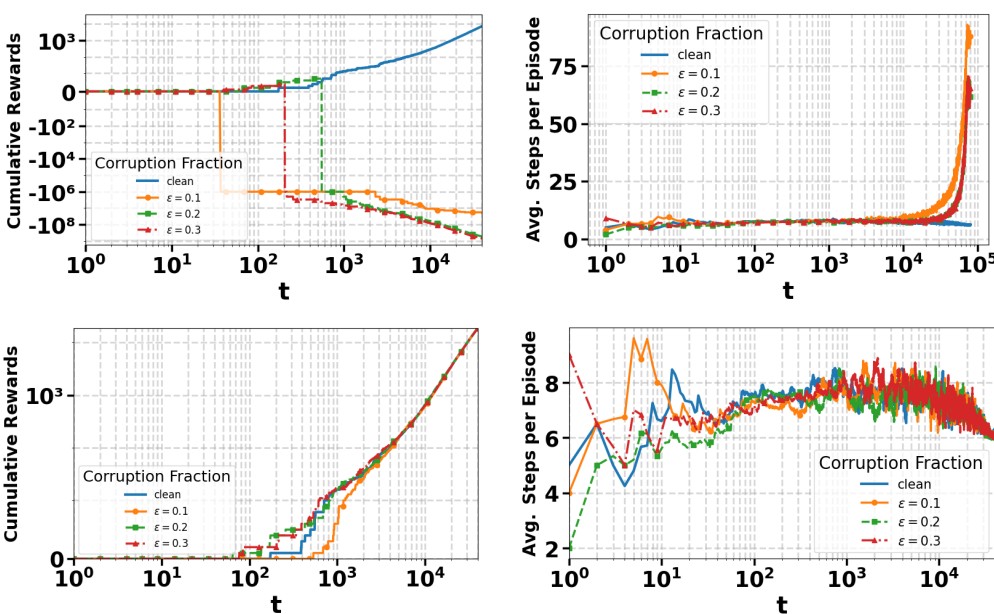

*Figure 6.* **(Top Left)** Cumulative training reward for `Vanilla-Q` in `FrozenLake-v1` under the Huber reward-corruption model in Eq. (2) with $\varepsilon \in \{0.1, 0.2, 0.3\}$. The rewards are perturbed according to the `Adversarial Reward Model for` **`FrozenLake-v1`**. **(Top Right)** Rolling average episode length for `Vanilla-Q` under the same setting. **(Bottom Left)** Cumulative training reward for `Robust Async-Q` under the same corruption levels and outlier model. **(Bottom Right)** Rolling average episode length for `Robust Async-Q`. Each plot in Figure 6 reports the average over 100 independent runs.

• `Evaluation:` In evaluation, we compare the robust greedy policy implied by the learned $Q$-table to the clean optimal policy. Let $Q^*$ denote the optimal action-value function of the *clean* MDP, computed once via value iteration using the true transition kernel $P$. At training time $t$, let $\hat{\pi}_t$ be the greedy policy induced by $Q_t$, i.e., $\hat{\pi}_t(s) \in \arg\max_{a \in \mathcal{A}} Q_t(s, a)$, and let $\hat{Q}_t := Q^{\hat{\pi}_t}$ be the action-value function of $\hat{\pi}_t$ under the clean dynamics, obtained by policy evaluation. We then report the evaluation error: $E_t := \|Q^{\hat{\pi}_t} - Q^*\|_{\infty}$, which measures how close the greedy policy's state-action value function is to the clean-optimal state-action value function.

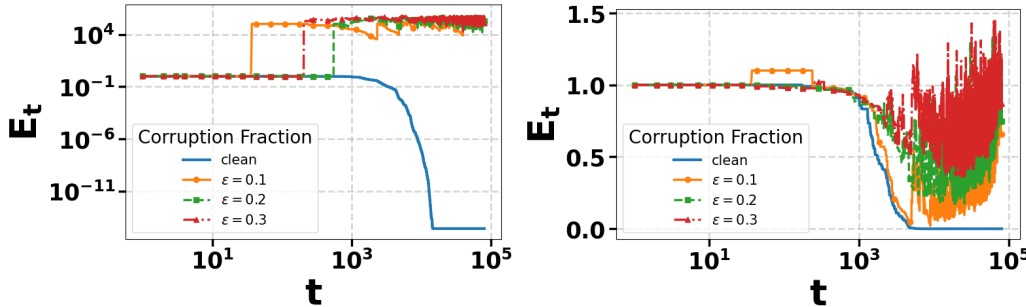

*Figure 7.* **(Left)** $\ell_\infty$-error $E_t = \|Q^{\hat{\pi}_t} - Q^*\|_\infty$ for `Vanilla-Q` under the adversarial reward-corruption model, where goal-reaching terminal rewards are replaced by large negative outliers with probability $\varepsilon$. **(Right)** The same error metric for `Robust Async-Q`, which mitigates these outliers and tracks the clean optimum more closely. Each plot in Figure 7 reports the average over 100 independent runs.

■ **`CliffWalking-v1`.** We now evaluate `Robust Async-RAQ` in the `CliffWalking-v1` environment (Towers et al., 2024) under the adversarial reward-corruption model defined in Eq. (2). The agent interacts with the underlying MDP using the true transition dynamics. The simulation results for `CliffWalking-v1` are presented in Figures 8 and 9.

● `Goal`. In `CliffWalking-v1`, the agent aims to navigate from the start to the goal on the grid while avoiding cliff cells that terminate the episode; it receives a reward of $-1$ per step, a large negative penalty of $-100$ upon stepping into the cliff, and 0 upon reaching the goal.

● `Adversarial Reward Model for `**`CliffWalking-v1`**: We adopt a Huber-style reward corruption model that targets cliff events in `CliffWalking-v1`. Whenever the agent steps into the cliff (i.e., the environment emits the cliff penalty), an adversary independently corrupts the observed reward with probability $\varepsilon$ by replacing it with a large *positive* outlier $r = +10^\beta$, where the exponent $\beta \in \{1, \dots, 5\}$ is chosen adversarially. With probability $1 - \varepsilon$, the agent observes the uncorrupted cliff penalty plus additive noise with bounded second moment; no assumptions are imposed on higher moments. All non-cliff rewards are left unchanged. This effectively makes the cliff *appear beneficial*: with probability $\varepsilon$, **stepping into the cliff** is reported as a **large positive reward**, which can incentivize the learner to move toward (or repeatedly fall into) the cliff.

● `Training`: In dense-penalty tasks such as `CliffWalking-v1`, `Vanilla-Q` can be highly sensitive to corrupted cliff penalties: when the agent steps into the cliff, the environment produces a large negative reward (e.g., $-100$), and an adversary occasionally replaces this signal with an extreme positive outlier following the `Adversarial model` described above. We ran the training for $t = 80000$ episodes.

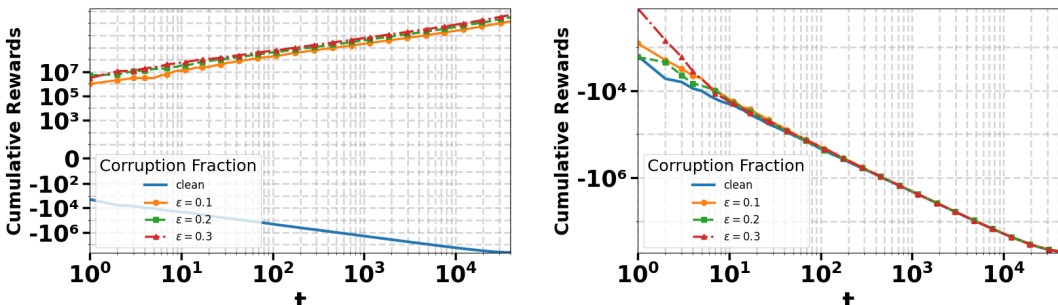

*Figure 8.* **(Left)** Cumulative training reward for `Vanilla-Q` in `CliffWalking-v1` under the Huber reward-corruption model in Eq. (2) with $\varepsilon \in \{0.1, 0.2, 0.3\}$. The rewards are perturbed according to the `Adversarial Reward Model for `**`CliffWalking-v1`**. **(Right)** Cumulative training reward for `Robust Async-RAQ`, with reward-agnostic parameter $p = 3$ under the same corruption levels and outlier model. Each plot in Figure 8 reports the average over 100 independent runs.

● `Motivation behind the Adversarial Model`: In `CliffWalking-v1`, clean rewards are negative (mean step cost $\approx -1$ and mean cliff penalty $\approx -100$), so the clean-optimal policy still has a negative return. Under cliff-poisoning, some cliff hits are replaced by a large *positive* outlier during training, which can make cumulative *training* rewards look deceptively high even when behavior degrades. Hence we judge performance using greedy evaluation on the *clean* environment, as done in Figure 9.

● `Evaluation`: In the `CliffWalking-v1` environment, the optimal policy on the *clean* MDP follows the shortest safe route along the cliff and reaches the goal in 13 steps, incurring a per-step reward of $-1$ and hence an optimal return of approximately $-13$. Under our corruption model, the adversary can inject large *positive* reward outliers, so the

*training* cumulative reward can increase even when the agent is learning a poor policy; training returns are therefore not a reliable indicator of true performance. To obtain an unambiguous performance measure, we periodically evaluated the learned policy by freezing the current $Q$-table and executing the *greedy* policy $\pi_Q(s) = \arg\max_a Q(s, a)$ on a *clean* `CliffWalking-v1` environment for $N_{\text{eval}} = 200$ independent episodes, once every 400 training episodes. This clean, greedy evaluation isolates the quality of the learned action-values from training-time corruption and noise: a near-optimal policy yields evaluation returns tightly concentrated near the optimal level (around $-13$), whereas substantial deviations indicate that learning has been misled.

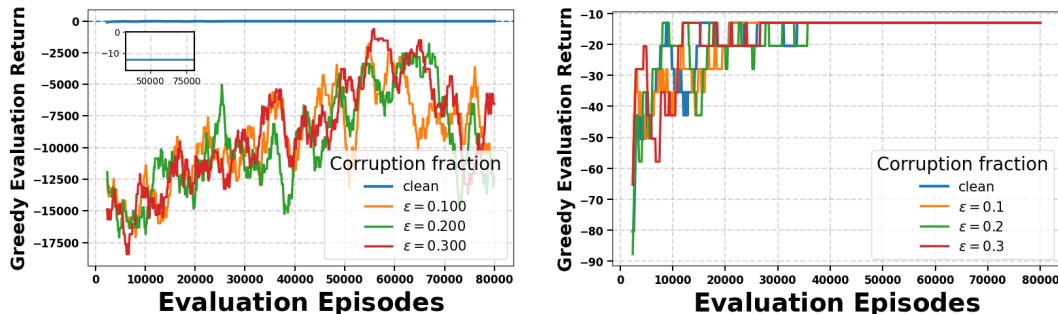

*Figure 9.* **(Left)** Greedy evaluation return for `Vanilla-Q` in `CliffWalking-v1` under the Huber reward-corruption model in Eq. (2) with $\varepsilon \in \{0.1, 0.2, 0.3\}$. The rewards are perturbed according to the `Adversarial Reward Model for` **CliffWalking-v1**. **(Right)** Greedy evaluation return for `Robust Async-RAQ` under the same corruption levels and outlier model. Each plot in Figure 9 reports the average over 100 independent runs.

■ **`Taxi-v3`.** We now evaluate `Robust Async-Q` in the `Taxi-v3` environment (Towers et al., 2024) under the adversarial reward-corruption model defined in Eq. (2). The transition dynamics are unaltered. The simulation results for `Taxi-v3` are presented in Figures 10 and 11.

● `Goal:` In `Taxi-v3`, the agent aims to navigate the grid to pick up the passenger at the correct location and drop them off at the designated destination; it receives $-1$ per time step, $+20$ for a successful drop-off, and $-10$ for an illegal pickup or drop-off action.

● `Adversarial Reward Model for` **`Taxi-v3`**: We use a reward-only Huber-style corruption model targeted to *successful drop-off* events in `Taxi-v3`. Whenever an episode terminates by completing the correct passenger drop-off (i.e., the environment emits the terminal success reward), an adversary independently corrupts the observed terminal reward with probability $\varepsilon$ by replacing it with a large *negative* outlier $r = -10^\beta$, where the exponent $\beta \in \{1, \dots, 6\}$ is chosen arbitrarily. With probability $1 - \varepsilon$, the agent observes the uncorrupted terminal reward plus additive noise with bounded second moment; no assumptions are imposed on higher moments. All non-terminal rewards are left unchanged. This effectively makes *success appear catastrophic*: with probability $\varepsilon$, **finishing the task** is reported as a **large negative reward**, which can discourage the learner from completing the drop-off.

● `Training:` In sparse-terminal-reward tasks such as `Taxi-v3`, `Vanilla-Q` can be highly sensitive to corrupted terminal success rewards: when the agent completes the correct drop-off, the environment produces a positive terminal reward (e.g., $+20$), and an adversary that occasionally replaces this signal with an extreme negative outlier following the `Adversarial model` described above. We ran the training for $t = 80000$ episodes.

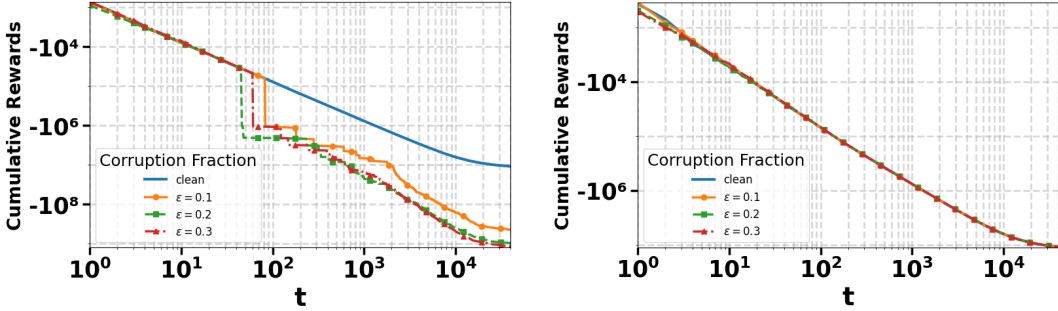

*Figure 10.* **(Left)** Cumulative training reward for `Vanilla-Q` in `Taxi-v3` under the Huber reward-corruption model in Eq. (2) with $\varepsilon \in \{0.1, 0.2, 0.3\}$. The rewards are perturbed according to the `Adversarial Reward Model for` **`Taxi-v3`**. **(Right)** Cumulative training reward for `Robust Async-Q`, under the same corruption levels and outlier model. Each plot in Figure 10 reports the average over 100 independent runs.

• `Motivation behind the Adversarial Model:` In `Taxi-v3`, clean rewards include a per-step penalty (typically mean $\approx -1$) and a sparse terminal success reward (typically mean $\approx +20$), so early in training the episode return is often negative until the agent reliably completes the correct drop-off. Under terminal-success poisoning, a fraction of successful drop-offs are replaced by a large *negative* outlier during training, which can make cumulative *training* rewards look deceptively low even when the learned behavior is improving. Hence we judge performance using greedy evaluation on the *clean* environment.

• `Evaluation:` We periodically evaluated the learned policy by freezing the current $Q$-table and executing the *greedy* policy $\pi_Q(s) = \arg\max_a Q(s, a)$ on a *clean* `Taxi-v3` environment for $N_{\text{eval}} = 200$ independent episodes, once every 100 training episodes. We plot the resulting greedy evaluation returns as an unambiguous behavioral measure of performance.

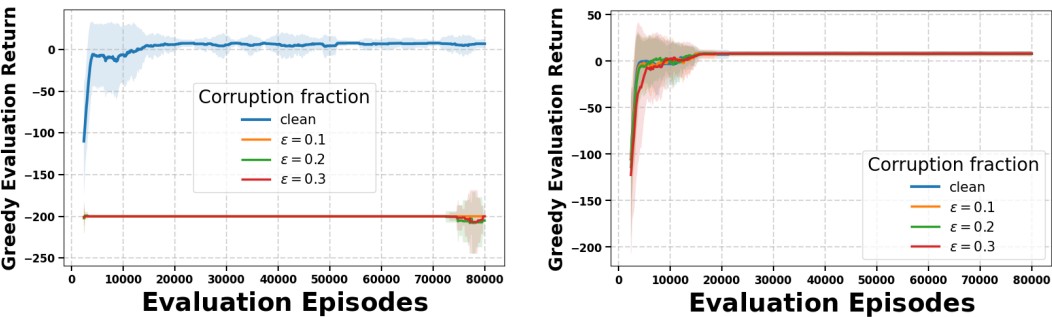

*Figure 11.* **(Left)** Greedy evaluation return for `Vanilla-Q` in `Taxi-v3` under the Huber reward-corruption model in Eq. (2) with $\varepsilon \in \{0.1, 0.2, 0.3\}$. The rewards are perturbed according to the `Adversarial Reward Model for Taxi-v3`. **(Right)** Greedy evaluation return for `Robust Async-Q` under the same corruption levels and outlier model. Each plot in Figure 11 shows the envelope spanning the individual runs, along with the average over 100 independent trials.

**Remark.** All the simulations in Appendix G are performed on an `Victus HP Gaming Laptop` with 12th Gen Intel(R) Core(TM) i7-12650H Processor.

