# OpenReview forum: "Corruption-Tolerant Asynchronous Q-Learning with Near-Optimal Rates"
_ICML.cc/2026/Conference — ICML 2026 regular_

### Official Review · Reviewer_Sopz · 2026-02-23

**Soundness:** 3
**Presentation:** 2
**Significance:** 3
**Originality:** 3
**Overall Recommendation:** 4
**Confidence:** 3

**Summary:**

This paper proposes a novel robust Q-learning algorithm to address online reinforcement learning under adversarially corrupted rewards in the infinite-horizon setting. The contributions are clearly articulated on both the algorithmic and theoretical sides. To the best of my understanding, this is a novel and rigorous work that systematically studies robust Q-learning under adversarial corruption with near-optimal guarantees.

**Compliance With Llm Reviewing Policy:**

Affirmed.

**Final Justification:**

Thank the authors for their detailed response. I would like to keep my score.

**Key Questions For Authors:**

1. In Assumption 1, the behavior policy $\mu$ affects both action and state. Could you clarify the “Markov chain” in Assumption 1? Is it the state process $s_t$, the action process $a_t$}, the joint process $(s_t,a_t)$, or $(s_t,a_t,r_t)$?
2. Do Theorems 1–5 all share the same data assumptions, particularly Assumption 1? How do the i.i.d. sampling assumptions in earlier sections align with the Markovian Assumption 1?
3. Is the state space assumed to be finite? It would improve clarity to explicitly list the conditions on: state/action space, moment/tail assumptions, dependence structure, data generation mechanism, algorithms, tuning parameters, and specify which theorems correspond to which settings.
4. The methodology requires several predetermined parameters. Is there any sensitivity analysis for these parameter choices? Are there data-driven rules for selecting them? For example, if the mixing time of a Markov chain is typically unknown, how should practitioners handle this in practice?
5. The paper proposes several modified algorithmic versions for different scenarios. Is there a unified or recommended implementation strategy that can handle all scenarios robustly? Alternatively, how should a practitioner decide which modification to use before starting online Q-learning?

**Limitations:**

yes

**Strengths And Weaknesses:**

Strengths:
1. Near-optimal finite-time guarantees under heavy-tailed rewards and adversarial corruption.
2. A novel robust Q-learning algorithm with adaptive thresholding, along with modified versions designed for different prior-knowledge settings.
3. Strong theoretical development, including careful analysis and new proof techniques to handle corruption and heavy-tailed data simultaneously.

Weaknesses:
1. Assumption structure is not clearly centralized. It would be more reader-friendly to explicitly summarize all assumptions that differ from traditional online RL settings. Currently, assumptions appear scattered across sections. A centralized and clearly structured presentation would help readers understand precisely under which conditions each theorem holds.
2. The methodology, including algorithmic modifications and tuning parameter choices, relies on quantities that may not be available in practice (e.g., mixing time). Since the data-generating scenario may be unknown to the agent before running online learning under corruption, this raises concerns about practical implementation.

---

> ### Author Rebuttal · Authors · 2026-03-30
>
> Dear Reviewer Sopz,
>
> Thank you for your thoughtful review of our paper and your encouraging comments; we are glad to know that you find our work novel and rigorous.
>
> Below, we respond to each of your main comments. We use 'W' to denote weaknesses and 'Q' to denote questions.
>
> - **[Q1]**: In Assumption 1, we are referring to the state process $\{s_t\}$ being aperiodic and irreducible. Specifically, any fixed policy $\mu$, upon interacting with the underlying MDP, induces a Markov chain *on the states* where the probability of transition from state $s$ to state $s'$ is given by $\sum_{a} \mu(a|s) P(s'|s,a)$. It is this Markov chain we are referring to in Assumption 1.
>
> That said, as we note explicitly in the Markovian extension (Section 4.1), the augmented process $\{Z_t:=(s_t,a_t,s_{t+1})\}$ is itself also a Markov chain, and this chain is ergodic as well under Assumption 1; we refer to [R1] below in our paper for a proof of this fact. We use this in our blocking and coupling argument for proving Theorem 5.
>
> In the revised manuscript, we will now explicitly specify that we are talking about the "state" Markov chain in Assumption 1.
>
> [R1]  Chen et al., Performance of Q-learning with linear function approximation: Stability and finite-time analysis, arXiv 2019.
>
> - **[Q2]**: Other than Theorem 3 which provides lower bounds, all our convergence results in Theorems 2, 4, and 5 do rely on Assumption 1. As we mention right before stating Assumption 1 in our paper, almost every RL paper we know of makes this assumption to ensure visitation to all states, needed for estimating the optimal state-action value function.
>
> For Theorems 2 and 4, we further make an  i.i.d. sampling assumption to help build intuition: at each time $t$, the state $s_t$ is sampled independently from the stationary distribution $\pi$ of the Markov chain induced by $\mu$, and then $a_t \sim \mu(\cdot\mid s_t)$. This i.i.d. model is introduced to isolate the technical challenges arising from asynchronous sampling, heavy-tailed rewards, and data corruption. *Assumption 1 is still relevant here* because it guarantees that the induced chain has a well-defined stationary distribution $\pi$ with strictly positive visitation probabilities.
>
> In Theorem 5, we consider the challenging single-trajectory Markovian regime, where $s_t$ evolves according to the induced Markov chain itself. Thus, to summarize, the i.i.d. results and the Markovian result *both* rely on Assumption 1. While the former assumes that data is generated from the stationary distribution of the policy-induced Markov chain, the latter considers data generated by the chain itself. Nonetheless, both require Assumption 1 for the chain to be well-behaved.
>
> We will clarify this relationship better in our revised paper.
>
> - **[Q3 and W1]**: Yes, the state and action spaces are both assumed to be finite, and we explicitly state this at the beginning of Section 2.
>
> We completely agree with the Reviewer's pertinent suggestion of having a centralized assumption structure. In the revised paper, in Section 2 itself, we plan to compile the following assumptions: (A1) Finite-state-action space; (A2) Bounded mean and variance of reward distributions; (A3) Ergodicity of Markov chain induced by behavior policy; (A4) I.I.D. data generation from stationary distribution. We will clarify that assumptions 1-3 pertain to Theorems 2, 4, and 5, and Theorems 2 and 4 further require assumption 4. We believe this will indeed improve the clarity of our presentation.
>
> - **[Q4 and W2]**: Consistent with the current finite-time RL theory papers (see, for example, Qu et al., COLT 2020), we assumed knowledge of the mixing time $\tau$ and minimum visitation probability $\lambda_{min}$ to focus on the challenge of adversarial corruption. Each of these objects can indeed be estimated from data. The mixing time can be estimated as in [R2] below, and $\lambda_{min}$ can be estimated by simply maintaining empirical visitation frequencies which converge to their stationary values under Assumption 1. Please also see our response to Reviewer YrUj in this regard where we provide more details. With suitable modifications, we believe our framework can accommodate such on-the-fly adaptive estimation of parameters.
>
> [R2] Wolfer and Kontorovich, Estimating the mixing time of ergodic markov chains, COLT 2019.
>
> - **[Q5]** Since Section 4.1. covers the most general setting with single-trajectory Markovian data and requires no prior knowledge of reward statistics, we would recommend using the algorithm prescribed in this section (formally stated as Algorithm 3 in Appendix F), with guarantees stated in Theorem 5. This algorithm and corresponding result subsumes all prior results in the paper. For practical implementation, Algorithm 3 can be further combined with data-driven estimates of $\tau$ and $\lambda_{min}$.
>
> We hope our rebuttal above has addressed your main comments. We would be happy to address any remaining concerns.

---

> > ### Author Rebuttal · Reviewer_Sopz · 2026-04-03
> >
> > Thank the authors for their detailed response. I would like to keep my score.

---

> > > ### Author Response · Authors · 2026-04-03
> > >
> > > We are glad to know that our responses have adequately addressed your concerns.
> > >
> > > Thank you once again for your review and your overall positive assessment. We will incorporate all your suggestions to the revised paper.

---

### Official Review · Reviewer_JXvK · 2026-03-09

**Soundness:** 2
**Presentation:** 1
**Significance:** 4
**Originality:** 3
**Overall Recommendation:** 4
**Confidence:** 2

**Summary:**

In this work, the authors investigate an online RL problem with infinite horizon and discounted cumulative reward objective, where the reward samples are Huber-contaminated by an arbitrary adversarial distribution with a small probability $\varepsilon$ independently at each time-step. The authors propose **Robust Async-Q**, which is an algorithm that accurately estimates the optimal Q function by using univariate trimmed mean estimator. With knowledge of the second moment of rewards and the upper bound on the absolute value of rewards, this algorithm achieves nearly optimal performance under i.i.d. sampling with high probability in theory not only when the rewards are contaminated, but also when the true rewards follow some heavy-tailed distribution. The authors then generalize Robust Async-Q to cases without any knowledge about the rewards, under which the generalized algorithm also attains nearly optimal performance under both i.i.d. sampling and Markovian sampling. It is also demonstrated that the extra $O(\sqrt{\varepsilon})$ term in the performance bound is unavoidable, indicating the effect of contamination. Some extra experiments are carried out to validate the performance of Robust Async-Q in practice.

**Compliance With Llm Reviewing Policy:**

Affirmed.

**Final Justification:**

My questions have all been addressed by the authors, and I will stick with my original score.

**Key Questions For Authors:**

1. Following the first weakness, is the performance nearly optimal with respect to $|\mathcal{S}|$ and $|\mathcal{A}|$ when $\lambda_{\min} = O(1/|\mathcal{S}||\mathcal{A}|)$?
2. Following the third weakness, what is the best choice of $p$ given a fixed $T$?
3. What is the value of $\mathcal{C}$ that you used in your experiment?

I will reevaluate my score based on how the authors addressed each question and weakness.

**Limitations:**

No. It would be helpful for the authors to address the case where Assumption 1 fails, and to mention that their results are asymptotic, indicating its limitation when $T$ is not large enough.

**Strengths And Weaknesses:**

In general, this is an interesting paper that tackles a highly challenging problem with solid results and some flaws mainly in the presentation.

**Strength**
1. In theory, the Robust Async-Q algorithm achieves near optimal performance under i.i.d. sampling, and it only requires knowledge of the second moment of rewards and the upper bound on the absolute value of rewards.
2. A new information-theoretic lower bound is derived, demonstrating that the Huber contamination incurs at least an extra $\Omega(\sqrt{\varepsilon})$ error term.
3. The generalized Robust Async-RAQ algorithm also achieves near optimal performance under both i.i.d. sampling and Markovian sampling without require any statistical knowledge about the rewards.
4. Simulations indicate that the algorithms also performs reasonably well in practice.
5. There is a refined version of Azuma-Hoeffding inequality that can possibly be leveraged in other context.

**Weakness**
1. The minimum visitation probability $\lambda_{\min}$ is smaller than $|\mathcal{S}||\mathcal{A}|$. Therefore, when writing the upper bounds in Theorems 2, 4, & 5, the author should consider the growth of $\lambda_{\min}^{-3/2}$ with respect to $|\mathcal{S}|$ and $|\mathcal{A}|$ and discuss when the performance is nearly optimal with respect to $|\mathcal{S}|$ and $|\mathcal{A}|$. However, no such discussion is found in the paper.
2. It would be preferable to have an explicit discussion about how to choose a behavioral policy satisfying Assumption 1.
3. When discussing the choice of $p$ in Theorem 4, the authors omit the fact that the choice of $p$ also affects the burn-in time. The larger the $p$ is, the longer the burn-in time is, making the "waiting time" to achieve the near optimal bound longer. Hence, when $T$ is not extremely large, it is not desirable to make $p$ as large as possible. It would be helpful if the authors take burn-in time into account.
4. The exact value of $\mathcal{C}$ in Theorem 1 is not specified, which is required when constructing the threshold $G_{t}$.
5. It would also be desirable to mention that Robust Async-Q is different from the behavioral policy, as readers might get confused about whether the action chosen is the one with the largest Q function estimate, or the one chosen by another different Markovian policy.
6. The bold texts and the italic tex are overused. Please only use it when the sentences/phrases are actually important.
7. Please move the sketches of proofs to be right after the theorems, which should save much space.
8. The **Simulation** paragraph should be a separate section, and it would be desirable to elaborate more on your experiments.
9. Please explain to the readers what $\tilde{O}$ stands for and how it differs from $O$ in the bullet point **Finite-Time Rates under I.I.D. Sampling**.
10. There is no need to have a **Main Takeaway** paragraph after each theorem, as the importance is emphasized in the introduction and some content in these paragraphs are redundant.

**Minor Typos/Suggestions**
1. On line 139, it should be "revealing that when Q-learning runs for T iterations,..." instead of "revealing that when runs for T iterations,...".
2. On line 163, it should be "Here, it is assumed..." instead of "Here, we assume...", as the authors of prior works instead of those of this paper make this assumption.
3. On line 298, the "recall that" can be discarded.
4. On line 383, remove the "of this result".

---

> ### Author Rebuttal · Authors · 2026-03-30
>
> Dear Reviewer JXvK,
>
> Thank you for your thoughtful review of our paper and your positive comments; we are glad to know that you consider our work to be an interesting paper that tackles a highly challenging problem with solid results.
>
> Below, we respond to each of your main comments. We use 'W' to denote weaknesses and 'Q' to denote questions.
>
> - **[W1 + Q1]**.  The object $\lambda_{min}$ does, in general, depend on $|S|$ and $|A|$. However, the exact functional dependence is unclear when the stationary distribution is not uniform. Thus, SOTA rates for Q-learning are typically stated in terms of $\lambda_{min}$; for instance, Theorem 7 in [R1] below provides a rate $\tilde{O}(1/(\lambda_{min} \sqrt{T})),$ where dependencies on $|S|$, $|A|$ are hidden within log factors. One can further upper bound this result by replacing $1/(\lambda_{min})$ with $|S| \times |A|$. To be consistent with such results in the literature, we also stated our main convergence results in Theorems 2, 4, and 5 in terms of $\lambda_{min}$.
>
> [R1] Qu and Wierman, Finite-time analysis of asynchronous stochastic approximation and Q-learning, COLT 2020.
>
> **On Optimality**: While the standard dependence on the minimum visitation probability is $1/(\lambda_{min})$, our dependence of $(\lambda_{min})^{-1.5}$ is slightly worse. As we explain in detail to Reviewer Fknz, the extra $(\lambda_{min})^{-0.5}$ factor can be traced to robust reward estimation under asynchronous sampling in Lemma 1, and is hence unique to our setting. We kindly request this Reviewer to look at our response to Fknz for understanding the source of the sup-optimal dependence on $\lambda_{min}$. In short, our bound is off from the dependence in [R1] by a factor of $(|S| \times |A|)^{0.5}$. Whether this is fundamental or an artifact of our analysis is unclear. We will explicitly point this out in our revised paper.
>
> - **[W2]**: We thank the Reviewer for this interesting question. Before we proceed, we should mention that the existence of a behavior policy ensuring Assumption 1 is extremely standard in RL, and appears in almost all papers we are aware of; see, for instance, the references in our paper right before Assumption 1.
>
> We now explain that given **any** stochastic policy $\mu$ such that $\mu(a|s) >0, \forall (s,a)$, Assumption 1 will hold under fairly mild conditions on the underlying MDP. To see this, we note that under policy $\mu$, the probability of going from a state $s$ to another state $s'$ is given by $P_\mu(s,s') = \sum_{a\in\mathcal A}\mu(a\mid s) P(s'\mid s,a).$ Now, consider the classical one-to-one correspondence between a Markov transition matrix $P_{\mu}$ and a directed graph  $G_\mu=(\mathcal S,E_\mu)$; here, the nodes of the graph are the states of the Markov chain, and an edge $(s,s') \in E_{\mu}$ exists if and only if $P_\mu(s,s')>0.$ For each action $a$, let the graph $G_a$ be defined similarly.
>
> With this interpretation, **irreducibility** boils down to $G_{\mu}$ being strongly connected, i.e., a directed path should exist between every pair of nodes/states.  It is easy to then see that as long as there exists even one action $a$ for which $G_a$ is strongly connected, $G_{\mu}$ will also be strongly connected. An even weaker sufficient condition is for the **union** of the graphs corresponding to the actions to be strongly connected.
>
> Similarly, for **aperiodicity**, a standard sufficient condition is for $G_{\mu}$ to have self-loops, i.e., we want $P_{\mu}(s,s) > 0, \forall s$. For this, a simple sufficient condition is that for each $s$, there exists an action $a(s)$ such that $P(s \mid s, a(s)) > 0$.
>
> In the revised manuscript, we will add a detailed discussion of the above point.
>
> - **[W3 + Q2]**. Indeed, as we also explain to Reviewer Fknz, the choice of $p$ not only affects the final bound, but also the burn-in time. We kindly ask the Reviewer to look at our response to Fknz where we outline how an optimal $p$ can be derived. However, the issue is that such a $p$ will depend on the *unknown* $\tilde{\sigma}$. Thankfully, our results hold for *any* $p \geq 1$, and as we explain to Fknz, a value of $p$ in the range 5-10 suffices for practical purposes.
>
> In the revised paper, we will clearly articulate the trade-offs imposed by the choice of $p$.
>
> - **[W4 + Q3]**: The constant $C$ in Theorem 1 is a universal constant from the trimmed-mean guarantee in [R2] below. A value of $C= 20\sqrt{3}$ suffices in [R2]. After the extra steps in our analysis (see Appendix B), a value of $C \approx 100$ suffices, which we used in our experiments. We will specify this constant in our revised paper.
>
> [R2] Lugosi et al., Robust multivariate mean estimation: the optimality of trimmed mean, The Annals of Statistics, 2021.
>
> We sincerely appreciate all the other pertinent suggestions in [W5] - [W10]; we will revise our paper accordingly.
>
> We hope our rebuttal above has addressed your main comments. We would be happy to address any remaining concerns.

---

> > ### Author Rebuttal · Reviewer_JXvK · 2026-04-03
> >
> > I appreciate the author's detailed response. I will stick with the original score.

---

> > > ### Author Response · Authors · 2026-04-04
> > >
> > > Thank you once again for your valuable review and your overall positive assessment of our work.
> > >
> > > We will incorporate all your suggestions in our revised paper.

---

### Official Review · Reviewer_Fknz · 2026-03-12

**Soundness:** 4
**Presentation:** 4
**Significance:** 3
**Originality:** 3
**Overall Recommendation:** 5
**Confidence:** 3

**Summary:**

This paper investigates the problem of finding an optimal policy in a discounted, infinite-horizon Markov Decision Process (MDP) when the reward signals are subject to $\epsilon$-adversarial corruption and heavy-tailed noise. The authors propose a robust variant of the asynchronous Q-learning algorithm that couples a univariate trimmed-mean estimator for robust reward estimation with a dynamically shrinking adaptive thresholding mechanism to filter out adversarial outliers. Theoretically, the authors prove that their algorithm achieves a finite-time convergence rate of $\tilde{\mathcal{O}}(1/\sqrt{T}) + \mathcal{O}(\sqrt{\epsilon})$ under both i.i.d. and Markovian asynchronous sampling models, matching the standard Q-learning rate up to an additive corruption term. Furthermore, they establish an information-theoretic lower bound of $\Omega(\sqrt{\epsilon})$, demonstrating that their achieved rate is near-optimal in its dependence on the corruption fraction. Finally, the authors extend their approach to a reward-agnostic setting where prior knowledge of the reward statistics is unavailable, relying on a refined variant of the Azuma-Hoeffding inequality to preserve the convergence guarantees.

**Compliance With Llm Reviewing Policy:**

Affirmed.

**Final Justification:**

See Rebuttal Acknowledgement.

**Key Questions For Authors:**

1. In Theorem 2, the sample complexity scales with $1/\lambda_{\min}^{3/2}$. Is this exponent strictly necessary due to the variance of the asynchronous robust estimation, or could a tighter analysis improve this dependence to the minimax optimal $1/\lambda_{\min}$ typically seen in synchronous theoretical settings? Such as utilizing variance-reduction techniques or reference-advantage learning.
2. The Markovian extension in Algorithm 3 requires discarding significant portions of the trajectory, using only every $\tau$-th sample. Can your analysis be mathematically extended to utilize the full trajectory without sub-sampling, potentially via constructing a Lyapunov drift argument over the full temporal sequence?
3. Regarding the reward-agnostic threshold $m(t) = t^p$ in Eq. (8) , the analysis fundamentally requires $p \ge 1$. How does the explicit choice of $p$ influence the burn-in time $\tilde{T} = \max\{\tilde{\sigma}^{1/p}, \bar{T}\}$  in the non-asymptotic regime? Is there a theoretical optimum for $p$ that explicitly balances the $\tilde{\sigma}$ inflation and the failure probability scaling $\delta_1$?

**Limitations:**

yes

**Strengths And Weaknesses:**

**Strengths:**
- The paper provides a highly comprehensive theoretical treatment of the problem by establishing not only upper bounds for both known-parameter and reward-agnostic settings but also proving a matching information-theoretic lower bound of $\Omega(\sqrt{\epsilon})$. The upper and lower bounds are tight with respect to the corruption fraction $\epsilon$.
- The introduction of the adaptive thresholding mechanism represents an elegant algorithmic choice. Standard robust mean estimators hold only with high probability, but the adaptive thresholding ensures uniform boundedness of the Q-learning iterates, which is a critical necessity for martingale-based concentration arguments.
- To analyze the reward-agnostic variant (Robust Async-RAQ), the authors innovatively deploy a refined variant of the Azuma-Hoeffding inequality for almost-martingales. This allows them to bypass the vacuous bounds that would normally arise from the naive deterministic $\mathcal{O}(T^p)$ bounds on the iterates, representing a strong technical contribution to the RL theory toolkit.

**Weaknesses:**
- The convergence bounds exhibit an inverse dependence on the minimum state-action visitation probability $\lambda_{\min}$ and Theorem 2 scales with $1/\lambda_{\min}^{3/2}$. While standard in basic asynchronous Q-learning analysis, this dependence can be problematic in large state spaces where $\lambda_{\min}$ becomes exponentially small, raising questions about the theoretical scalability of the sample complexity in environments requiring deep exploration.
- The extension to Markovian data (Theorem 5) relies on a sub-sampling trick, dropping data points to create approximately i.i.d. blocks based on the mixing time $\tau$. This is a somewhat loose theoretical mechanism that mathematically wastes data and inflates the sample complexity, rather than analyzing the true single-trajectory temporal difference updates directly using generalized state-space mixing-time arguments.
- In the reward-agnostic setting (Theorem 4), the use of a polynomially growing threshold $m(t) = t^p$ mathematically inflates the dominant term's dependence on $\tilde{\sigma}$ to $\tilde{\sigma}^{1+1/(2p)}$. While asymptotically negligible, this introduces a direct trade-off between the failure probability parameter and the effective horizon bounds, making the non-asymptotic bounds highly sensitive to the arbitrary choice of the hyperparameter $p$.

---

> ### Author Rebuttal · Authors · 2026-03-30
>
> Dear Reviewer Fknz,
>
> Thank you for your thoughtful review of our paper and your positive comments; we are glad to know that you find our treatment highly comprehensive, and our algorithmic choice elegant.
>
> Below, we respond to each of your main comments. We use 'W' to denote weaknesses and 'Q' to denote questions.
>
> - **[W1+Q1]**: The Reviewer has correctly noted that while the standard dependence on the minimum visitation probability is $1/(\lambda_{min})$, our dependence of $(\lambda_{min})^{-1.5}$ is slightly worse. As we explain below, the source of the additional $(\lambda_{min})^{-0.5}$ factor can be traced back to robust reward estimation under asynchronous sampling, and is hence unique to our setting.
>
> Consider Lemma 1 in our paper that pertains to robust reward estimation, and notice that after $t$ time-steps, the error scales as $\tilde{O}(1/\sqrt{\lambda_{min} t})$ up to an additive $O(\sqrt{\epsilon})$ term. Recalling the meaning of $\lambda_{min}$, note that at time-step $t$, the number of visits/samples for the least-visited state-action pair is precisely on the order $\lambda_{min} t$, and not $t$. Now, when one invokes the robust mean estimation guarantee from Theorem 1 with $M=\lambda_{min} t$ samples, the error which scales as $O(1/\sqrt{M})$ naturally picks up the extra $(\lambda_{min})^{-0.5}$ factor. Given the nature of our update rule, the reward estimation error gets hit by the step-size $\alpha$ (see pages 16 and 17 in Appendix C), and $\alpha$ in asynchronous Q-learning typically scales inversely with $\lambda_{min}$.
>
> The main takeaway from this discussion is that it is the robust estimation part under asynchronous sampling that leads to the sub-optimal dependence on $\lambda_{min}$. The Reviewer's suggestion that variance-reduction might improve this dependence is well taken, and we will investigate this as future work.
>
> In our revised paper, we will add a detailed discussion of the above points.
>
> - **[W2 + Q2]**: The Reviewer makes an excellent point that our approach of handling Markov data via sub-sampling does, indeed, waste data. Nonetheless, we adopted this proof technique for three reasons. First, it allowed us to build concrete intuition by focusing on the simpler i.i.d. setting first, and tackling the complex interplay between asynchronous sampling and corrupted rewards. Second, even after using the crude "blocking" approach, our bounds seems to be no worse (theoretically) compared to SOTA finite-time rates that instead rely on mixing-time arguments. For instance, the bounds in [R1]-[R2] below are also inflated by a mixing time factor $\tau$, much like our bound in Theorem 5.
>
> [R1] Gu and Wierman, Finite-time analysis of asynchronous stochastic approximation and Q-learning, COLT 2020.
>
> [R2] Chen et al., Performance of Q-learning with linear function approximation: Stability and finite-time analysis, arXiv 2019.
>
> Third, and most importantly, the "i.i.d. data assumption" is crucial for us to be able to leverage the refined Azuma-Hoeffding bound needed for the reward agnostic setting. From Appendix C, our error decomposition step leads to a "noise term" that is a martingale only under i.i.d. noise. Under Markov noise, the martingale structure is lost, and one needs to further control a bias/drift term that captures the gap to stationarity of the underlying Markov chain. In the absence of such an analysis for the refined Azuma-Hoeffding inequality, we pursued the "i.i.d. data sampling approach", followed by a "blocking + coupling" argument to port the i.i.d. guarantees to the Markov setting.
>
> Under Markov data, it might be possible to treat the "noise term" as a martingale + drift effect, and then devise ways of controlling the latter. We will add a discussion of this potential strategy in our revised paper.
>
> - **[W3 + Q3]**: Indeed, the choice of $p$ not only affects the burn-in time, but also the main convergence bound in Theorem 4. From the dependence of $\delta_1$ on $p$, the burn-in time trade-off is reflected in $\max \{p, \tilde{\sigma}^{1/p} \}$, while the trade-off in the final convergence bound shows up as $ \sqrt{p} \tilde{\sigma}^{1/p}$. One can choose a *theoretically optimal p* to balance either of these metrics (burn-in or final bound) by using the above explicit expressions. That said, such an optimal $p$ would depend on the *unknown* $\tilde{\sigma}$. Fortunately, *any* $p \geq 1$ works in our analysis; a $p$ in the range $5-10$ typically suffices for practical purposes. To see why, even if $\tilde{\sigma}$ is as large as $1000$, $\max \{p, \tilde{\sigma}^{1/p} \} = 5$, with $p=5$. Such a $p$ only inflates the main bound by a factor of $\sqrt{5}$, and worsens the exponent of dependence on $\tilde{\sigma}$ by $0.2$ relative to when $\tilde{\sigma}$ is known.
>
> We will add a clearer description of the above points in the revised paper.
>
> We hope our rebuttal above has addressed your main comments. We would be happy to address any remaining concerns.

---

> > ### Author Rebuttal · Reviewer_Fknz · 2026-04-04
> >
> > Thanks for the response. I will keep my positive score.

---

> > > ### Author Response · Authors · 2026-04-04
> > >
> > > Thank you once again for your valuable review, and your positive assessment of our work.
> > >
> > > We will incorporate all your suggestions in our revised paper.

---

### Official Review · Reviewer_YrUj · 2026-03-13

**Soundness:** 3
**Presentation:** 3
**Significance:** 2
**Originality:** 3
**Overall Recommendation:** 4
**Confidence:** 3

**Summary:**

This paper studies infinite-horizon discounted RL with asynchronous Q-learning under a Huber-style reward corruption model, where, at each step, with probability $1-eps$ the agent observes a reward from the true distribution, while with probability $eps$ it observes an arbitrary adversarial sample). To solve this problem, the authors propose Robust Async-Q, which adaptively discards “extreme” estimator outputs to stabilize the process. They provide theoretical guarantee under that setting and they also provides a lower bound suggesting near-optimality.

**Compliance With Llm Reviewing Policy:**

Affirmed.

**Final Justification:**

The concern was addressed. I am on the positive side.

**Key Questions For Authors:**

Please see the weakness.

Q1. The results depend on prior knowledge about $eps$; it would be better to discuss if the algorithm only has noisy feedback about $eps$.

**Strengths And Weaknesses:**

Strengths
+ The topic is important and realistic, which might facilitate its use in many real-world settings, and the method is both heuristic and theoretically sound. And the authors provide sufficient detail on the proof and design. For the reward-agnostic case, the authors identify a refined Azuma–Hoeffding inequality that appears to be novel. They also provide empirical evaluation.

Weakness
- It seems that their algorithm must know the corruption fraction, which is often impractical. And the thresholding $G_t$ also requires $\lambda_{min}$.
- The algorithm also requires maintaining a dynamic history of state-action, which inherently limits its application to large applications.
- The empirical results are limited, t is unclear whether the paper compares against other robust alternatives

---

> ### Author Rebuttal · Authors · 2026-03-30
>
> Dear Reviewer YrUj,
>
> Thank you for your thoughtful review of our paper and your encouraging comments; we are glad to know that you find our problem important and realistic, and our approach theoretically sound.
>
> Below, we respond to each of your main comments. We use 'W' to denote weaknesses and 'Q' to denote questions.
>
> - **[W1 + Q1]**: Indeed, our current approach does require knowledge of both $\epsilon$ and $\lambda_{min}$. That said, as we discuss below, almost every paper we are aware of requires such information. Moreover, we also explain that an upper-bound on $\epsilon$ will suffice, and $\lambda_{min}$ can be readily estimated from data.
>
> The knowledge of $\epsilon$ is standard in robust estimation under corrupted data: **all the robust procedures** we know of require either the corruption fraction itself or a valid upper bound on it in order to calibrate trimming or filtering operations. For instance, [R1-R3] below on robust statistics,  [R4-R5] on corrupted bandits, and [R6] in offline RL under corruption, all make this assumption. Since our goal in this work was to establish the theoretical foundations of corruption-robust asynchronous $Q$-learning, assuming knowledge of $\epsilon$ provided us with a clean starting point, allowing us to isolate and address the main technical difficulties created by asynchronous sampling, heavy-tailed rewards, and adversarial contamination.
>
>  We further note that exact knowledge of $\epsilon$ is not essential for implementing our method. If $\epsilon$ is unknown but one has access to a valid upper bound $\bar{\epsilon} \geq \epsilon$, then our algorithms can be run using $\bar{\epsilon}$ in place of $\epsilon$. All our guarantees would then simply scale with $\bar{\epsilon}$.
>
> [R1] Lugosi et al., Robust multivariate mean estimation: the optimality of trimmed mean, The Annals of Statistics, 2021.
>
> [R2] Lai et al., Agnostic estimation of mean and covariance, FOCS, 2016.
>
> [R3] Dalalyan et al., All-in-one robust estimator of the gaussian mean, The Annals of Statistics, 2022.
>
> [R4] Kapoor et al., Corruption-tolerant bandit learning, Machine Learning, 2019.
>
> [R5]  Agrawal et al., Crimed: Lower and upper bounds on regret for bandits with unbounded stochastic corruption, ALT 2024.
>
> [R6] Mandal et al., Corruption robust offline reinforcement learning with human feedback, arXiv 2024.
>
> -- *On knowledge of $\lambda_{min}$* : The knowledge of  $\lambda_{min}$ is also not unique to our setting: even in the absence of reward corruption, state-of-the-art finite-time analyses of asynchronous \(Q\)-learning (see [R7] and [R8] below) assume knowledge of  $\lambda_{min}$ in order to properly design the step-size. In this sense, our assumption is consistent with the current theory for asynchronous \(Q\)-learning.
>
> In principle, one can estimate $\lambda_{min}$ from data by tracking empirical visitation frequencies, which converge to their stationary values under Assumption 1 (via strong law of large numbers for ergodic Markov chains). Such empirical frequencies could therefore serve as proxies for the stationary probabilities, including $\lambda_{min}$. Exploring such a strategy is a ripe direction for future work.
>
> [R7] Qu and Wierman, Finite-time analysis of asynchronous stochastic approximation and Q-learning, COLT 2020.
>
> [R8] Li et al., Is Q-learning minimax optimal? A tight sample complexity analysis, Operations Research, 2024.
>
> In the revised paper, we will add a detailed discussion of the above points.
>
> - **[W2]**  We agree that the current algorithm maintains a dynamic history of rewards for each state-action pair. This is needed because our robust reward-estimation step is based on the trimmed-mean estimator, which is inherently a batch estimator and does not admit a simple recursive update of the kind available for ordinary sample averages. We believe that our core conceptual ideas can be extended to address this issue by using more recent online robust mean estimation methods (see [R9] below), which are designed to be more memory-efficient while retaining robustness guarantees. We will add a discussion of this in the revised paper.
>
> [R9] Kane et al., Online Robust Mean Estimation, SODA 2024.
>
> - **[W3]** We appreciate your comment on our empirical part. While our paper's main contribution is theoretical, Appendix G already includes a reasonably broad set of simulations on several standard benchmark environments, including FrozenLake, CliffWalking, and Taxi. Across these experiments, we vary the size of the MDP, the corruption fraction, the noise level, and attack magnitudes to empirically corroborate our theory. Since there are no other robust Q-learning methods we are aware of for our setting, we only compared with the vanilla Q-learning algorithm. If the Reviewer would like us to conduct some specific experiments, we would be happy to do so.
>
> We hope our rebuttal above has addressed your main comments. We would be happy to address any remaining concerns.

---

> > ### Author Rebuttal · Reviewer_YrUj · 2026-04-03
> >
> > Thanks for the rebuttal. It addressed my concern, please add these discussions to the revision.

---

> > > ### Author Response · Authors · 2026-04-03
> > >
> > > We are glad to know that our rebuttal has addressed your concerns.
> > >
> > > Thank you once again for your positive comments overall. We will make sure to add all the relevant discussions to the revised paper.

---

### Decision · Program_Chairs · 2026-04-30

**Decision:**

Accept (regular)

**Comment:**

The paper studies infinite-horizon discounted RL with corrupted rewards and provides a comprehensive theoretical treatment. Reviewers have generally appreciated the effort in proving matching lower and upper bounds and the overall presentation of the paper. The work makes an interesting contribution to the literature on robustness in RL and I recommend acceptance.